# Early Neuron Alignment in Two-layer ReLU Networks with Small Initialization

**Hancheng Min**
University of Pennsylvania
hanchmin@seas.upenn.edu

**Enrique Mallada**
Johns Hopkins University
mallada@jhu.edu

**René Vidal**
University of Pennsylvania
vidalr@seas.upenn.edu

## Abstract

This paper studies the problem of training a two-layer ReLU network for binary classification using gradient flow with small initialization. We consider a training dataset with well-separated input vectors: Any pair of input data with the same label are positively correlated, and any pair with different labels are negatively correlated. Our analysis shows that, during the early phase of training, neurons in the first layer try to align with either the positive data or the negative data, depending on its corresponding weight on the second layer. A careful analysis of the neurons' directional dynamics allows us to provide an $\mathcal{O}(\frac{\log n}{\sqrt{\mu}})$ upper bound on the time it takes for all neurons to achieve good alignment with the input data, where $n$ is the number of data points and $\mu$ measures how well the data are separated. After the early alignment phase, the loss converges to zero at a $\mathcal{O}(\frac{1}{t})$ rate, and the weight matrix on the first layer is approximately low-rank. Numerical experiments on the MNIST dataset illustrate our theoretical findings.

## 1 Introduction

Neural networks have shown excellent empirical performance in many application domains such as vision (Krizhevsky et al., 2012), speech (Hinton et al., 2012) and video games (Silver et al., 2016). Despite being highly overparametrized, networks trained by gradient descent with random initialization and without explicit regularization enjoy good generalization performance. One possible explanation for this phenomenon is the implicit bias or regularization induced by first-order algorithms under certain initialization assumptions. For example, first-order methods applied to (deep) matrix factorization models may produce solutions that have low nuclear norm (Gunasekar et al., 2017) and/or low rank (Arora et al., 2019), and similar phenomena have been observed for deep tensor factorization (Razin et al., 2022). Moreover, prior work such as (Saxe et al., 2014; Stöger & Soltanolkotabi, 2021) has found that deep linear networks sequentially learn the dominant singular values of the input-output correlation matrix.

It is widely known that these sparsity-inducing biases can often be achieved by small initialization. This has motivated a series of works that theoretically analyze the training dynamics of first-order methods for neural networks with small initialization. For linear networks, the implicit bias of small initialization has been studied in the context of linear regression (Saxe et al., 2014; Gidel et al., 2019; Min et al., 2021; Varre et al., 2023) and matrix factorization (Gunasekar et al., 2017; Arora et al., 2019; Li et al., 2018; 2021; Stöger & Soltanolkotabi, 2021; Yaras et al., 2023; Soltanolkotabi et al., 2023). Recently, the effect of small initialization has been studied for two-layer ReLU networks (Maennel et al., 2018; Lyu et al., 2021; Phuong & Lampert, 2021; Boursier et al., 2022). For example, Maennel et al. (2018) observes that during the early stage of training, neurons in the first layer converge to one out of finitely many directions determined by the dataset. Based on this observation, Phuong & Lampert (2021) shows that in the case of well-separated data, where any pair of input data with the same label are positively correlated and any pair with different labels are negatively correlated, there are only two directions the neurons tend to converge to: the positive data center and the negative one. Moreover, Phuong & Lampert (2021) shows that if such directional convergence holds, then the loss converges, and the resulting first-layer weight matrix is low-rank. However, directional convergence is assumed in their analysis; there is no explicit characterization of how long it takes to achieve directional convergence and how the time to convergence depends on the initialization scale.

**Paper contributions**: In this paper, we provide a complete analysis of the dynamics of gradient flow for training a two-layer ReLU network on well-separated data with small initialization. Specifically, we show that if the initialization is sufficiently small, during the early phase of training the neurons in the first layer try to align with either the positive data or the negative data, depending on its corresponding weight on the second layer. Moreover, through a careful analysis of the neuron's directional dynamics we show that the time it takes for all neurons to achieve good alignment with the input data is upper bounded by $\mathcal{O}(\frac{\log n}{\sqrt{\mu}})$, where $n$ is the number of data points and $\mu$ measures how well the data are separated. We also show that after the early alignment phase the loss converges to zero at a $\mathcal{O}(\frac{1}{t})$ rate and that the weight matrix on the first layer is approximately low-rank.

**Notation**: We denote the Euclidean norm of a vector $x$ by $\|x\|$, the inner product between the vectors $x$ and $y$ by $\langle x, y \rangle = x^\top y$, and the cosine of the angle between them as $\cos(x, y) = \langle \frac{x}{\|x\|}, \frac{y}{\|y\|} \rangle$. For an $n \times m$ matrix $A$, we let $A^\top$ denote its transpose. We also let $\|A\|_2$ and $\|A\|_F$ denote the spectral norm and Frobenius norm of $A$, respectively. For a scalar-valued or matrix-valued function of time, $F(t)$, we let $\dot{F} = \dot{F}(t) = \frac{d}{dt}F(t)$ denote its time derivative. Furthermore, we define $\mathbb{1}_A$ to be the indicator for a statement $A$: $\mathbb{1}_A = 1$ if $A$ is true and $\mathbb{1}_A = 0$ otherwise. We also let $I$ denote the identity matrix, and $\mathcal{N}(\mu, \sigma^2)$ denote the normal distribution with mean $\mu$ and variance $\sigma^2$ .

## 2 PRELIMINARIES

In this section, we first discuss problem setting. We then present some key ingredients for analyzing the training dynamics of ReLU networks under small initialization, and discuss some of the weaknesses/issues from prior work.

### 2.1 PROBLEM SETTING

We are interested in a binary classification problem with dataset $[x_1, \cdots, x_n] \in \mathbb{R}^{D \times n}$ (input data) and $[y_1, \cdots, y_n]^\top \in \{-1, +1\}^n$ (labels). For the classifier, $f : \mathbb{R}^D \to \mathbb{R}$, we consider a two-layer ReLU network:

$$f(x; W, v) = v^\top \sigma(W^\top x) = \sum_{j=1}^{h} v_j \sigma(w_j^\top x), \qquad (1)$$

parametrized by network weights $W := [w_1, \cdots, w_h] \in \mathbb{R}^{D \times h}, v := [v_1, \cdots, v_h]^\top \in \mathbb{R}^{h \times 1}$, where $\sigma(\cdot) = \max\{\cdot, 0\}$ is the ReLU activation function. We aim to find the network weights that minimize the training loss $\mathcal{L}(W, v) = \sum_{i=1}^{n} \ell(y_i, f(x_i; W, v))$, where $\ell : \mathbb{R} \times \mathbb{R} \to \mathbb{R}_{\geq 0}$ is either the exponential loss $\ell(y, \hat{y}) = \exp(-y\hat{y})$ or the logistic loss $\ell(y, \hat{y}) = \log(1 + \exp(-y\hat{y}))$. The network is trained via the gradient flow (GF) dynamics

$$\dot{W} \in \partial_W \mathcal{L}(W, v), \ \dot{v} \in \partial_v \mathcal{L}(W, v), \qquad (2)$$

where $\partial_W \mathcal{L}, \partial_v \mathcal{L}$ are Clark sub-differentials of $\mathcal{L}$. Therefore, (2) is a differential inclusion (Bolte et al., 2010). For simplicity of presentation, instead of directly working on this differential inclusion, our theoretical results will be stated for the Caratheodory solution (Reid, 1971) of (2) when the ReLU subgradient is fixed as $\sigma'(x) = \mathbb{1}_{x>0}$[1]. In Appendix E, we show that under the data assumption of our interest (to be introduced later), the Caratheodory solution (s) $\{W(t), v(t)\}$ exists globally for all $t \in [0, \infty)$, which we call the solution (s) of (2) throughout this paper.

To initialize the weights, we consider the following initialization scheme. First, we start from a weight matrix $W_0 \in \mathbb{R}^{D \times h}$ , and then and then initialize the weights as

$$W(0) = \epsilon W_0, \quad v_j(0) \in \{\|w_j(0)\|, -\|w_j(0)\|\}, \forall j \in [h]. \qquad (3)$$

That is, the weight matrix $W_0$ determines the initial shape of the first-layer weights $W(0)$ and we use $\epsilon$ to control the initialization scale and we are interested in the regime where $\epsilon$ is sufficiently small. For the second layer weights $v(0)$, each $v_j(0)$ has magnitude $\|w_j(0)\|$ and we only need to decide its sign. Our results in later sections are stated for a deterministic choice of $\epsilon, W_0$, and $v(0)$, then we comment on the case where $W_0$ is chosen randomly via some distribution.

The resulting weights in (3) are always "balanced", i.e., $v_j^2(0) - \|w_j(0)\|^2 = 0, \forall j \in [h]$, because $v_j(0)$ can only take two values: either $\|w_j(0)\|$ or $-\|w_j(0)\|$. More importantly, under GF (2), this

---

[1]In Appendix F, we discuss how our results can be extended to the solution to differential inclusion.

balancedness is preserved (Du et al., 2018): $v_j^2(t) - \|w_j(t)\|^2 = 0, \forall t \geq 0, \forall j \in [h]$. In addition, it is shown in Boursier et al. (2022) that $\text{sign}(v_j(t)) = \text{sign}(v_j(0)), \forall t \geq 0, \forall j \in [h]$, and the dynamical behaviors of neurons will be divided into two types, depending on $\text{sign}(v_j(0))$.

**Remark 1.** *For our theoretical results, the balancedness condition is assumed for technical purposes: it simplifies the dynamics of GF and thus the analysis. It is a common assumption for many existing works on both linear (Arora et al., 2018b) and nonlinear (Phuong & Lampert, 2021; Boursier et al., 2022) neural networks. For the experiments in Section 4, we use a standard Gaussian initialization (not balanced) with a small variance to validate our theoretical findings.*

**Remark 2.** *Without loss of generality, we consider the case where all columns of $W_0$ are nonzero, i.e., $\|w_j(0)\| > 0, \forall j \in [h]$. We make this assumption because whenever $w_j(0) = 0$, we also have $v_j(0) = 0$ from the balancedness, which together would imply $\dot{v}_j \equiv 0, \dot{w}_j \equiv 0$ under gradient flow. As a result, $w_j$ and $v_j$ would remain zero and thus they could be ignored in the convergence analysis.*

**Remark 3.** *Our main results will depend on both $\max_j \|w_j(0)\|$ and $\min_j \|w_j(0)\|$, as shown in our proofs in Appendices C and D. Therefore, whenever we speak of small initialization, we will say that $\epsilon$ is small without worrying about the scale of $W_0$, which is already considered in our results.*

### 2.2 NEURAL ALIGNMENT WITH SMALL INITIALIZATION: AN OVERVIEW

Prior work argues that the gradient flow dynamics (2) under small initialization (3), i.e., when $\epsilon$ is sufficiently small, can be roughly described as "align then fit" (Maennel et al., 2018; Boursier et al., 2022) : During the early phase of training, every neuron $w_j, j \in [h]$ keeps a small norm $\|w_j\|^2 \ll 1$ while changing their directions $\frac{w_j}{\|w_j\|}$ significantly in order to locally maximize a "signed coverage" (Maennel et al., 2018) of itself w.r.t. the training data. After the alignment phase, part of (potentially all) the neurons grow their norms in order to fit the training data, and the loss decreases significantly. The analysis for the fitting phase generally depends on the resulting neuron directions at the end of the alignment phase (Phuong & Lampert, 2021; Boursier et al., 2022). However, prior analysis of the alignment phase either is based on a vanishing initialization argument that can not be directly translated into the case finite but small initialization (Maennel et al., 2018) or assumes some stringent assumption on the data (Boursier et al., 2022). In this section, we provide a brief overview of the existing analysis for neural alignment and then point out several weaknesses in prior work.

**Prior analysis of the alignment phase:** Since during the alignment phase all neurons have small norm, prior work mainly focuses on the directional dynamics, i.e., $\frac{d}{dt} \frac{w_j}{\|w_j\|}$, of the neurons. The analysis relies on the following approximation of the dynamics of every neuron $w_j, j \in [h]$:

$$\frac{d}{dt} \frac{w_j}{\|w_j\|} \simeq \text{sign}(v_j(0)) \mathcal{P}_{w_j(t)} x_a(w_j), \quad (4)$$

where $\mathcal{P}_w = I - \frac{ww^\top}{\|w\|^2}$ is the projection onto the subspace orthogonal to $w$ and

$$x_a(w) := \sum\nolimits_{i:\langle x_i, w \rangle > 0} y_i x_i \quad (5)$$

denotes the signed combination of the data points activated by $w$. First of all, (4) implies that the dynamics $\frac{w_j}{\|w_j\|}$ are approximately decoupled, and thus one can study each $\frac{w_j}{\|w_j\|}$ separately. Moreover, as illustrated in Figure 1, if $\text{sign}(v_j(0)) > 0$, the

Figure 1: Illustration of $\frac{d}{dt} \frac{w_j}{\|w_j\|}$ during the early alignment phase. $x_1$ has $+1$ label, and $x_2, x_3$ have $-1$ labels, $x_1, x_2$ lie inside the halfspace $\langle x, w_j \rangle > 0$ (gray shaded), thus $x_a(w_j) = x_1 - x_2$. Since $\text{sign}(v_j(0)) > 0$, GF pushes $w_j$ towards $x_a(w_j)$.

flow (4) pushes $w_j$ towards $x_a(w_j)$, since $w_j$ is attracted by its currently activated positive data and repelled by its currently activated negative data. Intuitively, during the alignment phase, a neuron $w_j$ with $\text{sign}(v_j(0)) > 0$ would try to find a direction where it can activate as much positive data and as less negative data as possible. If $\text{sign}(v_j(0)) < 0$, the opposite holds.

Indeed, Maennel et al. (2018) claims that the neuron $w_j$ would be aligned with some "extreme vectors", defined as vector $w \in \mathbb{S}^{D-1}$ that locally maximizes $\sum_{i \in [n]} y_i \sigma(\langle x_i, w \rangle)$ (similarly, $w_j$ with $\text{sign}(v_j(0)) < 0$ would be aligned with the local minimizer), and there are only finitely many such vectors. The analysis is done under the limit $\epsilon \to 0$, where the approximation in (4) is exact.

**Weakness in prior analyses**: Although Maennel et al. (2018) provides great insights into the dynamical behavior of the neurons in the alignment phase, the validity of the aforementioned approximation for finite but small $\epsilon$ remains in question. First, one needs to make sure that the error $\left\| \frac{d}{dt} \frac{w_j}{\|w_j\|} - \text{sign}(v_j(0)) \mathcal{P}_{w_j} x_a(w_j) \right\|$ is sufficiently small when $\epsilon$ is finite in order to justify (4) as a good approximation. Second, the error bound needs to hold for the entire alignment phase. Maennel et al. (2018) assumes $\epsilon \to 0$; hence there is no formal error bound. In addition, prior analyses on small initialization (Stöger & Soltanolkotabi, 2021; Boursier et al., 2022) suggest the alignment phase only holds for $\Theta(\log \frac{1}{\epsilon})$ time. Thus, the claim in Maennel et al. (2018) would only hold if good alignment is achieved before the alignment phase ends. However, Maennel et al. (2018) provides no upper bound on the time it takes to achieve good alignment. Therefore, without a finite $\epsilon$ analysis, Maennel et al. (2018) fails to fully explain the training dynamics under small initialization. Understanding the alignment phase with finite $\epsilon$ requires additional quantitative analysis. To the best of our knowledge, this has only been studied under a stringent assumption that all data points are orthogonal to each other (Boursier et al., 2022), or that there are effectively two data points Wang & Ma (2023).

**Goal of this paper**: In this paper, we want to address some of the aforementioned issues by developing a formal analysis for the early alignment phase with a finite but small initialization scale $\epsilon$. We first discuss our main theorem that shows that a directional convergence can be achieved within bounded time under data assumptions that are less restrictive and have more practical relevance. Then, we discuss the error bound for justifying (4) in the proof sketch of the main theorem.

## 3 CONVERGENCE OF TWO-LAYER RELU NETWORKS WITH SMALL INITIALIZATION

Our main results require the following data assumption:

**Assumption 1.** *Any pair of data with the same (different) label is positively (negatively) correlated, i.e.,* $\min_{i,j} \frac{\langle x_i y_i, x_j y_j \rangle}{\|x_i\| \|x_j\|} := \mu > 0$.

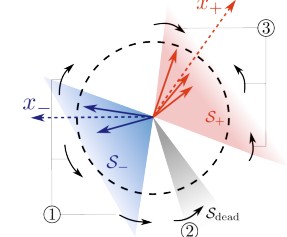

Given a training dataset, we define $\mathcal{S}_+ := \{ z \in \mathbb{R}^D : \mathbb{1}_{\langle x_i, z \rangle > 0} = \mathbb{1}_{y_i > 0}, \forall i \}$ to be the cone in $\mathbb{R}^n$ such that whenever neuron $w \in \mathcal{S}_+$, $w$ is activated exclusively by every $x_i$ with a positive label (see Figure 2). Similarly, for $x_i$ with negative labels, we define $\mathcal{S}_- := \{ z \in \mathbb{R}^D : \mathbb{1}_{\langle x_i, z \rangle > 0} = \mathbb{1}_{y_i < 0}, \forall i \}$. Finally, we define $\mathcal{S}_{\text{dead}} := \{ z \in \mathbb{R}^D : \langle z, x_i \rangle \leq 0, \forall i \}$ to be the cone such that whenever $w \in \mathcal{S}_{\text{dead}}$, no data activates $w$. Given Assumption 1, it can be shown (see Appendix C) that $\mathcal{S}_+$ ($\mathcal{S}_-$) is a non-empty, convex cone that contains all positive data $x_i, i \in \mathcal{I}_+$ (negative data $x_i, i \in \mathcal{I}_-$). $\mathcal{S}_{\text{dead}}$ is a convex cone as well, but not necessarily non-empty. We illustrate these cones in Figure 2 given some training data (red solid arrow denotes positive data and blue denotes negative ones).

Figure 2: Neuron alignment under Assumption 1. For neurons in $\mathcal{V}_+$, ① if it lies inside $\mathcal{S}_-$, then it gets repelled by $x_-$ and escapes $\mathcal{S}_-$; Once outside $\mathcal{S}_-$, it may ② get repelled by some negative data and eventually enters $\mathcal{S}_{\text{dead}}$, or may ③ gain some activation on positive data and eventually enter $\mathcal{S}_+$, then get constantly attracted by $x_+$.

Moreover, given some initialization from (3), we define $\mathcal{I}_+ := \{ i \in [n] : y_i > 0 \}$ to be the set of indices of positive data, and $\mathcal{I}_- := \{ i \in [n] : y_i < 0 \}$ for negative data. We also define $\mathcal{V}_+ := \{ j \in [h] : \text{sign}(v_j(t)) > 0 \}$ to be the set of indices of neurons with positive second-layer entry and $\mathcal{V}_- := \{ j \in [h] : \text{sign}(v_j(t)) < 0 \}$ for neurons with negative second-layer entry. Note that, as discussed in Section 2.1, $\text{sign}(v_j(t))$ does not change under balanced initialization, thus $\mathcal{V}_+, \mathcal{V}_-$ are time invariant. Further, as we discussed in Section 2.2 about the early alignment phase, we expect that every neuron in $\mathcal{V}_+$ will drift toward the region where positive data concentrate and thus eventually reach $\mathcal{S}_+$ or $\mathcal{S}_{\text{dead}}$, as visualized in Figure 2 ($x_+, x_-$ shown in the figure are defined in Assumption 2). Similarly, all neurons in $\mathcal{V}_-$ would chase after negative data and thus reach $\mathcal{S}_-$ or $\mathcal{S}_{\text{dead}}$. Our theorem precisely characterizes this behavior.

### 3.1 MAIN RESULTS

Our main results are stated for solutions to the GF dynamics (2). However, in rare cases, solutions to (2) could be non-unique and there are potentially "irregular solutions" (please refer to Appendix E.4

for details) that allow some neurons to regain activation even after becoming completely deactivated in $\mathcal{S}_{\text{dead}}$. We deem such irregular solutions of little practical relevance since when implementing gradient descent algorithm in practice, neurons in $\mathcal{S}_{\text{dead}}$ would receive zero update and thus stay in $\mathcal{S}_{\text{dead}}$. Therefore, our main theorem concerns some *regular* solutions to (2) (the existence of such solutions is shown in Appendix E.2), as defined below.

**Definition 1.** *A solution $\{W(t), v(t)\}$ to (2) is regular if it satisfy that $w_j(t_0) \in \mathcal{S}_{\text{dead}}$ for some $j \in [h]$ and some $t_0 \geq 0$ implies $w_j(t) \in \mathcal{S}_{\text{dead}}, \forall t \geq t_0$.*

Before we present our main theorem, we also need the following assumption on the initialization, for technical reasons, essentially asking the neuron $w_j(0), j \in \mathcal{V}_+$ (or $w_j(0), j \in \mathcal{V}_-$, resp.) to not be completely aligned with $x_+$ (or $x_-$, resp.).

**Assumption 2.** *The initialization from (3) satisfies that $\max_{j \in \mathcal{V}_+} \langle \frac{w_j(0)}{\|w_j(0)\|}, \frac{x_-}{\|x_-\|} \rangle < 1$, and $\max_{j \in \mathcal{V}_-} \langle \frac{w_j(0)}{\|w_j(0)\|}, \frac{x_+}{\|x_+\|} \rangle < 1$, where $x_+ = \sum_{i \in \mathcal{I}_+} x_i$ and $x_- = \sum_{i \in \mathcal{I}_-} x_i$.*

We are now ready to present our main result (given Assumption 1 and Assumption 2):

**Theorem 1.** *Given some initialization from (3), if $\epsilon = \mathcal{O}(\frac{1}{\sqrt{h}} \exp(-\frac{n}{\sqrt{\mu}} \log n))$, then for any regular solution to the gradient flow dynamics (2), we have*

1. *(Directional convergence in early alignment phase) $\exists t_1 = \mathcal{O}(\frac{\log n}{\sqrt{\mu}})$, such that*

   - *$\forall j \in \mathcal{V}_+$, either $w_j(t_1) \in \mathcal{S}_+$ or $w_j(t_1) \in \mathcal{S}_{dead}$. Moreover, if $\max_{i \in \mathcal{I}_+} \langle w_j(0), x_i \rangle > 0$, then $w_j(t_1) \in \mathcal{S}_+$.*
   - *$\forall j \in \mathcal{V}_-$, either $w_j(t_1) \in \mathcal{S}_-$ or $w_j(t_1) \in \mathcal{S}_{dead}$. Moreover, if $\max_{i \in \mathcal{I}_-} \langle w_j(0), x_i \rangle > 0$, then $w_j(t_1) \in \mathcal{S}_-$.*

2. *(Final convergence and low-rank bias) $\forall t \geq t_1$ and $\forall j \in [h]$, neuron $w_j(t)$ stays within $\mathcal{S}_+$ ($\mathcal{S}_-$, or $\mathcal{S}_{dead}$) if $w_j(t_1) \in \mathcal{S}_+$ ($\mathcal{S}_-$, or $\mathcal{S}_{dead}$ resp.). Moreover, if both $\mathcal{S}_+$ and $\mathcal{S}_-$ contains at least one neuron at time $t_1$, then*

   - *$\exists \alpha > 0$ and $\exists t_2$ with $t_1 \leq t_2 = \Theta(\frac{1}{n} \log \frac{1}{\sqrt{h}\epsilon})$, such that $\mathcal{L}(t) \leq \frac{\mathcal{L}(t_2)}{\mathcal{L}(t_2)\alpha(t-t_2)+1}$, $\forall t \geq t_2$.*
   - *As $t \to \infty$, $\|W(t)\| \to \infty$ and $\|W(t)\|_F^2 \leq 2\|W(t)\|_2^2 + \mathcal{O}(\epsilon)$. Thus, the stable rank of $W(t)$ satisfies $\limsup_{t \to \infty} \|W(t)\|_F^2 / \|W(t)\|_2^2 \leq 2$.*

We provide a proof sketch that highlights the technical novelty of our results in Section 3.3. Our $\mathcal{O}(\cdot)$ notations hide additional constants that depend on the data and initialization, for which we refer readers to the complete proof of Theorem 1 in Appendix C and D. We make the following remarks:

**Early neuron alignment**: The first part of the Theorem 1 describes the configuration of *all* neurons at the end of the alignment phase. Every neuron in $\mathcal{V}_+$ reaches either $\mathcal{S}_+$ or $\mathcal{S}_{\text{dead}}$ by $t_1$, and stays there for the remainder of training. Obviously, we care about those neurons reaching $\mathcal{S}_+$ as any neuron in $\mathcal{S}_{\text{dead}}$ does not contribute to the final convergence at all. Luckily, Theorem 1 suggests that any neuron in $\mathcal{V}_+$ that starts with some activation on the positive data, i.e., it is initialized in the union of halfspaces $\cup_{i \in \mathcal{I}_+} \{w : \langle w, x_i \rangle > 0\}$, will eventually reach $\mathcal{S}_+$. A similar discussion holds for neurons in $\mathcal{V}_-$. We argue that randomly initializing $W_0$ ensures that with high probability, there will be at least a pair of neurons reaching $\mathcal{S}_+$ and $\mathcal{S}_-$ by time $t_1$ (please see the next remark). Lastly, we note that it is possible that $\mathcal{S}_{\text{dead}} = \emptyset$, in which case every neuron reaches either $\mathcal{S}_+$ or $\mathcal{S}_-$.

**Merits of random initialization**: Our theorem is stated for a deterministic initialization (3) given an initial shape $W_0$. In practice, one would use random initialization to find a $W_0$, for example, $[W_0]_{ij} \overset{i.i.d.}{\sim} \mathcal{N}(0, 1/D)$. First, our Theorem 1 applies to this Gaussian initialization: Assumption 2 is satisfied with probability one because the events $\left\{ \langle \frac{w_j(0)}{\|w_j(0)\|}, \frac{x_-}{\|x_-\|} \rangle = 1 \right\}$ and $\left\{ \langle \frac{w_j(0)}{\|w_j(0)\|}, \frac{x_+}{\|x_+\|} \rangle = 1 \right\}$ have probability zero. Moreover, any neuron in $\mathcal{V}_+$ has at least probability $1/2$ of being initialized within the union of halfspaces $\cup_{i \in \mathcal{I}_+} \{w : \langle w, x_i \rangle > 0\}$, which ensures that this neuron reaches $\mathcal{S}_+$. Thus when there are $m$ neurons in $\mathcal{V}_+$, the probability that $\mathcal{S}_+$ has at least one neuron at time $t_1$ is lower bounded by $1 - 2^{-m}$ (same argument holds for $\mathcal{S}_-$), Therefore, with only very mild overparametrization on the network width $h$, one can make sure that with high probability there is at least one neuron in both $\mathcal{S}_+$ and $\mathcal{S}_-$, leading to final convergence.

**Importance of a quantitative bound on** $t_1$: The analysis for neural alignment relies on the approximation in (4), which, through our analysis (see Lemma 1), is shown to only hold before $T = \Theta(\frac{1}{n} \log \frac{1}{\sqrt{h}\epsilon})$, thus if one proves, through the approximation in (4), that good alignment is achieved within $t_1$ time, then the initialization scale $\epsilon$ must be chosen to be $\mathcal{O}(\frac{1}{\sqrt{h}} \exp(-nt_1))$ so that $t_1 \leq T$, i.e. the proved alignment should finish before the approximation (4) fails. Therefore, without an explicit bound on $t_1$, one does not know a prior how small $\epsilon$ should be. Our quantitative analysis shows that under (4), directional convergence is achieved within $t_1 = \mathcal{O}(\frac{\log n}{\sqrt{\mu}})$ time. This bound, in return, determines the bound for initialization scale $\epsilon$. Moreover, our bound quantitatively reveals the non-trivial dependency on the "data separation" $\mu$ for such directional convergence to occur. Indeed, through a numerical illustration in Appendix A.2, we show that the dependence on the data separability $\mu > 0$ is crucial in determining the scale of the initialization: As $\mu$ approaches zero, the time needed for the desired alignment increases, necessitate the use of a smaller $\epsilon$.

**Refined alignment within** $\mathcal{S}_+, \mathcal{S}_-$: Once a neuron in $\mathcal{V}_+$ reaches $\mathcal{S}_+$, it never leaves $\mathcal{S}_+$. Moreover, it always gets attracted by $x_+$. Therefore, every neuron gets well aligned with $x_+$, i.e., $\cos(w_j, x_+) \simeq 1, \forall w_j \in \mathcal{S}_+$. A similar argument shows neurons in $\mathcal{S}_-$ get attracted by $x_-$. We opt not to formally state it in Theorem 1 as the result would be similar to that in (Boursier et al., 2022), and alignment with $x_+, x_-$ is not necessary to guarantee convergence. Instead, we show this refined alignment through our numerical experiment in Section 4.

**Final convergence and low-rank bias**: We present the final convergence results mostly for the completeness of the analysis. GF after $t_1$ can be viewed as fitting positive data $x_i, i \in \mathcal{I}_+$, with a subnetwork consisting of neurons in $\mathcal{S}_+$, and fitting negative data with neurons in $\mathcal{S}_-$. By the fact that all neurons in $\mathcal{S}_+$ activate all $x_i, i \in \mathcal{I}_+$, the resulting subnetwork is linear, and so is the subnetwork for fitting $x_i, i \in \mathcal{I}_-$. The convergence analysis reduces to establishing $\mathcal{O}(1/t)$ convergence for two linear networks (Arora et al., 2018a; Min et al., 2021; Yun et al., 2020). The non-trivial and novel part is to show that right after the alignment phase ends, one can expect a substantial decrease of the loss (starting from time $t_2 = \Theta(\frac{1}{n} \log \frac{1}{\sqrt{h}\epsilon})$). An alternative way of proving convergence is by observing that at $t_1$, all data has been correctly classified (w.r.t. sign of $f$), which is sufficient for showing $\mathcal{O}(\frac{1}{t \log t})$ convergence (Lyu & Li, 2019; Ji & Telgarsky, 2020) of the loss, but this asymptotic rate does not suggest a time after which the loss start to decrease significantly. As for the stable rank, our result follows the analysis in Le & Jegelka (2022), but in a simpler form since ours is for linear networks. Although convergence is established partially by existing results, we note that these analyses are all possible because we have quantitatively bound $t_1$ in the alignment phase.

## 3.2 COMPARISON WITH PRIOR WORK

Our results provide a complete (from alignment to convergence), non-asymptotic (finite $\epsilon$), quantitative (bounds on $t_1, t_2$) analysis for the GF in (2) under small initialization. Similar neural alignment has been studied in prior work for *orthogonally separable* data (same as ours) and for *orthogonal* data, and we shall discuss them separately.

**Alignment under orthogonally separable data**: Phuong & Lampert (2021) assumes that there exists a time $t_1$ such that at $t_1$, the neurons are in either $\mathcal{S}_+, \mathcal{S}_-$ or $\mathcal{S}_{\text{dead}}$ and their main contribution is the analysis of the implicit bias for the later stage of the training. they justify their assumption by the analysis in Maennel et al. (2018), which does not necessarily apply to the case of finite $\epsilon$, as we discussed in Section 2.2. Later Wang & Pilanci (2022) shows $t_1$ exists, provided that the initialization scale $\epsilon$ is sufficiently small, but still with no explicit analysis showing how $t_1$ depends on the data separability $\mu$ and the size of the training data $n$. Moreover, there is no quantification on how small $\epsilon$ should be. In our work, all the results are non-asymptotic and quantitative: we show that good alignment is achieved within $t_1 = \mathcal{O}(\frac{\log n}{\sqrt{\mu}})$ time and provide an explicit upper bound on $\epsilon$. Moreover, our results highlight the dependence on the separability $\mu > 0$, (Further illustrated in Appendix A.2) which is not studied in Phuong & Lampert (2021); Wang & Pilanci (2022).

**Alignment under orthogonal data**: In Boursier et al. (2022), the neuron alignment is carefully analyzed for the case all data points are orthogonal to each other, i.e., $\langle x_i, x_j \rangle = 0, \forall i \neq j \in [n]$. We point out that neuron behavior is different under orthogonal data (illustrated in Appendix A.3): only the positive (negative) neurons initially activate all the positive (negative) data will end up in $\mathcal{S}_+$ ($\mathcal{S}_-$). In our case, all positive (negative) neurons will arrive at $\mathcal{S}_+$ ($\mathcal{S}_-$), unless they become a dead

neuron. Moreover, due to such distinction, the analysis is different: Boursier et al. (2022) restrict their results to positive (negative) neurons $w_j$ that initially activate all the positive (negative) data, and there is no need for analyzing neuron activation. However, since our analysis is on all positive neurons, regardless of their initial activation pattern, it utilizes novel techniques to track the evolution of the activation pattern (see Section 3.3).

**Other related work**: Convergence of two-layer (leaky-)ReLU networks are also studied under non-small initialization settings, mainly for gradient descent Wang & Ma (2022) and for training only the first-layer weights Frei et al. (2022); Kou et al. (2023). There is no direct comparison to them as they study the convergence in other regimes. Nonetheless, the analyses of neural alignment remain essential in these works but are done through different tools (one no longer has an approximation in (4)). We note that such analyses also require certain restrictive data assumptions. For example, Wang & Ma (2022) assumes orthogonal separability, together with some geometric constraint on the data; Frei et al. (2022); Kou et al. (2023) assumes high-dimensional near-orthogonal data.

### 3.3 PROOF SKETCH FOR THE ALIGNMENT PHASE

In this section, we sketch the proof for our Theorem 1. First of all, it can be shown that $\mathcal{S}_+, \mathcal{S}_{\text{dead}}$ are trapping regions for all $w_j(t), j \in \mathcal{V}_+$, that is, whenever $w_j(t)$ gets inside $\mathcal{S}_+$ (or $\mathcal{S}_{\text{dead}}$), it never leaves $\mathcal{S}_+$ (or $\mathcal{S}_{\text{dead}}$). Similarly, $\mathcal{S}_-, \mathcal{S}_{\text{dead}}$ are trapping regions for all $w_j(t), j \in \mathcal{V}_-$. The alignment phase analysis concerns how long it takes for all neurons to reach one of the trapping regions, followed by the final convergence analysis on fitting data with $+1$ label by neurons in $\mathcal{S}_+$ and fitting data with $-1$ label by those in $\mathcal{S}_-$. We have discussed the final convergence analysis in the remark "Final convergence and low-rank bias", thus we focus on the proof sketch for the early alignment phase here, which is considered as our main technical contribution.

**Approximating $\frac{d}{dt}\frac{w_j}{\|w_j\|}$**: Our analysis for the neural alignment is rooted in the following Lemma:

**Lemma 1.** *Given some initialization from (3), if $\epsilon = \mathcal{O}(\frac{1}{\sqrt{h}})$, then there exists $T = \Theta(\frac{1}{n}\log\frac{1}{\sqrt{h}\epsilon})$ such that any solution to the gradient flow dynamics (2) satisfies that $\forall t \leq T$,*

$$\max_j \left\| \frac{d}{dt}\frac{w_j(t)}{\|w_j(t)\|} - \text{sign}(v_j(0))\mathcal{P}_{w_j(t)}x_a(w_j(t)) \right\| = \mathcal{O}\left(\epsilon n\sqrt{h}\right). \tag{6}$$

This Lemma shows that the error between $\frac{d}{dt}\frac{w_j(t)}{\|w_j(t)\|}$ and $\text{sign}(v_j(0))\mathcal{P}_{w_j(t)}x_a(w_j(t))$ can be arbitrarily small with some appropriate choice of $\epsilon$ (to be determined later). This allows one to analyze the true directional dynamics $\frac{w_j(t)}{\|w_j(t)\|}$ using some property of $\mathcal{P}_{w_j(t)}x_a(w_j(t))$, which leads to a $t_1 = \mathcal{O}(\frac{\log n}{\sqrt{\mu}})$ upper bound on the time it takes for the neuron direction to converge to the sets $\mathcal{S}_+$, $\mathcal{S}_-$, or $\mathcal{S}_{\text{dead}}$. Moreover, it also suggests $\epsilon$ can be made sufficiently small so that the error bound holds until the directional convergence is achieved, i.e. $T \geq t_1$. We will first illustrate the analysis for directional convergence, then close the proof sketch with the choice of a sufficiently small $\epsilon$.

**Activation pattern evolution:** Given a sufficiently small $\epsilon$, one can show that under Assumption 1, for every neuron $w_j$ that is not in $\mathcal{S}_{\text{dead}}$ we have:

$$\frac{d}{dt}\left\langle \frac{w_j}{\|w_j\|}, \frac{x_i y_i}{\|x_i\|} \right\rangle \bigg|_{\langle w_i, x_i \rangle = 0} > 0, \forall i \in [n], \text{if } j \in \mathcal{V}_+, \tag{7}$$

$$\frac{d}{dt}\left\langle \frac{w_j}{\|w_j\|}, \frac{x_i y_i}{\|x_i\|} \right\rangle \bigg|_{\langle w_i, x_i \rangle = 0} < 0, \forall i \in [n], \text{if } j \in \mathcal{V}_-. \tag{8}$$

This is because if a neuron satisfies $\langle x_i, w_j \rangle = 0$ for some $i$, and is not in $\mathcal{S}_{\text{dead}}$, GF moves $w_j$ towards $x_a(w_j) = \sum_{i:\langle x_i, w_j \rangle > 0} x_i y_i$. Interestingly, Assumption 1 implies $\langle x_i y_i, x_a(w_j) \rangle > 0, \forall i \in [n]$, which makes $\frac{d}{dt}\frac{w_j}{\|w_j\|} \simeq \text{sign}(v_j(0))\mathcal{P}_{w_j}x_a(w_j)$ point inward (or outward) the halfspace $\langle x_i y_i, w_j \rangle > 0$, if $\text{sign}(v_j(0)) > 0$ (or $\text{sign}(v_j(0)) < 0$, respectively). See Figure 3 for illustration.

As a consequence, a neuron can only change its activation pattern in a particular manner: a neuron in $\mathcal{V}_+$, whenever it is activated by some $x_i$ with $y_i = +1$, never loses the activation on $x_i$ thereafter, because (7) implies that GF pushes $\frac{w_j}{\|w_j\|}$ towards $x_i$ at the boundary $\langle w_j, x_i \rangle = 0$. Moreover, (7)

also shows that a neuron in $\mathcal{V}_+$ will never regain activation on a $x_i$ with $y_i = -1$ once it loses the activation because GF pushes $\frac{w_j}{\|w_j\|}$ against $x_i$ at the boundary $\langle w_i, x_i \rangle = 0$. Similarly, a neuron in $\mathcal{V}_-$ never loses activation on negative data and never gains activation on positive data.

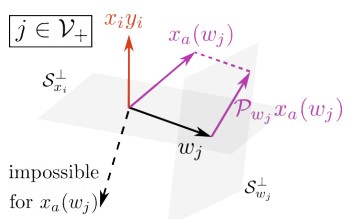

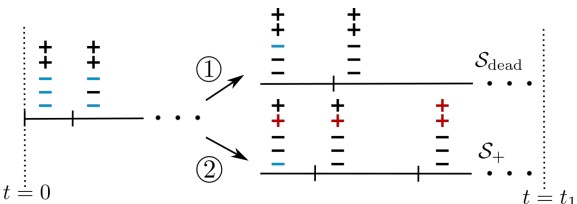

Figure 3: For $j \in \mathcal{V}_+$, Assumption 1 enforces $\langle x_i y_i, x_a(w_j) \rangle > 0$, thus GF pushes $w_j$ inward the half-space $\langle x_i y_i, w_j \rangle > 0$ at $\langle x_i, w_j \rangle = 0$ (i.e. towards gaining activation on $x_i$, if $y_i = +1$, or losing activation on $x_i$, if $y_i = -1$.). $\mathcal{S}_{x_i}^\perp$ and $\mathcal{S}_{w_j}^\perp$ denotes the subspace orthogonal to $x_i$ and $w_j$, respectively.

Figure 4: Illustration of the activation pattern evolution. The epochs on the time axis denote the time $w_j$ changes its activation pattern by either losing one negative data (denoted by "+") or gaining one positive data (denoted by "−"). The markers are colored if it currently activates $w_j$. During the alignment phase $0 \leq t \leq t_1$, a neuron $w_j, j \in \mathcal{V}_+$ starts with activation on all negative data and no positive data, every $\mathcal{O}(1/n_a)$ time, it must change its activation, unless either ① it reaches $\mathcal{S}_{\text{dead}}$, or ② it activates some positive data at some epoch then eventually reaches $\mathcal{S}_+$.

**Bound on activation transitions and duration:** Equations (7) and (8) are key in the analysis of alignment because they limit how many times a neuron can change its activation pattern: a neuron in $\mathcal{V}_+$ can only gain activation on positive data and lose activation on negative data, thus at maximum, a neuron $w_j$, $j \in \mathcal{V}_+$, can start with full activation on all negative data and no activation on any positive one (which implies $w_j(0) \in \mathcal{S}_-$) then lose activation on every negative data and gain activation on every positive data as GF training proceeds (which implies $w_j(t_1) \in \mathcal{S}_+$), taking at most $n$ changes on its activation pattern. See Figure 4 for an illustration. Then, since it is possible to show that a neuron $w_j$ with $j \in \mathcal{V}_+$ that has $\cos(w_j, x_-) < 1$ (guaranteed by Assumption 2) and is not in $\mathcal{S}_+$ or $\mathcal{S}_{\text{dead}}$, must change its activation pattern after $\mathcal{O}(\frac{1}{n_a \sqrt{\mu}})$ time (that does not depend on $\epsilon$), where $n_a$ is the number of data that currently activates $w_j$, one can upper bound the time for $w_j$ to reach $\mathcal{S}_+$ or $\mathcal{S}_{\text{dead}}$ by some $t_1 = \mathcal{O}(\frac{\log n}{\sqrt{\mu}})$ constant independent of $\epsilon$. Moreover, $w_j$ must reach $\mathcal{S}_+$ if it initially has activation on at least one positive data, i.e., $\max_{i \in \mathcal{I}_+} \langle w_j(0), x_i \rangle > 0$ since it cannot lose this activation. A similar argument holds for $w_j, j \in \mathcal{V}_-$ that they reaches either $\mathcal{S}_-$ or $\mathcal{S}_{\text{dead}}$ before $t_1$.

**Choice of $\epsilon$:** All the aforementioned analyses rely on the assumption that the approximation in equation (4) holds with some specific error bound. We show in Appendix C that the desired bound is $\left\| \frac{d}{dt} \frac{w_j(t)}{\|w_j(t)\|} - \text{sign}(v_j(0)) \mathcal{P}_{w_j(t)} x_a(w_j(t)) \right\| \leq \mathcal{O}(\sqrt{\mu})$, which, by Lemma 1, can be achieved by a sufficiently small initialization scale $\epsilon_1 = \mathcal{O}(\frac{\sqrt{\mu}}{\sqrt{h}n})$. Moreover, the directional convergence (which takes $\mathcal{O}(\frac{\log n}{\sqrt{\mu}})$ time) should be achieved before the alignment phase ends, which happens at $T = \Theta(\frac{1}{n} \log \frac{1}{\sqrt{h}\epsilon})$. This is ensured by choosing another sufficiently small initialization scale $\epsilon_2 = \mathcal{O}(\frac{1}{\sqrt{h}} \exp(-\frac{n}{\sqrt{\mu}} \log n))$. Overall, the initialization scale should satisfy $\epsilon \leq \min\{\epsilon_1, \epsilon_2\}$. We opt to present $\epsilon_2$ in our main theorem because $\epsilon_2$ beats $\epsilon_1$ when $n$ is large.

## 4 NUMERICAL EXPERIMENTS

We use a toy example in Appendix A.1 to clearly visualize the neuron alignment during training (due to space constraints). In the main body of this paper, we validate our theorem using a binary classification task for two MNIST digits. Such training data do not satisfy Assumption 1 since every data vector is a grayscale image with non-negative entries, making the inner product between any pair of data non-negative, regardless of their labels. However, we can preprocess the training data by centering: $x_i \leftarrow x_i - \bar{x}$, where $\bar{x} = \sum_{i \in [n]} x_i / n$. The preprocessed data, then, approximately satisfies our assumption (see the left-most plot in Figure 5): a pair of data points is very likely to have a positive correlation if they have the same label and to have a negative correlation if they have

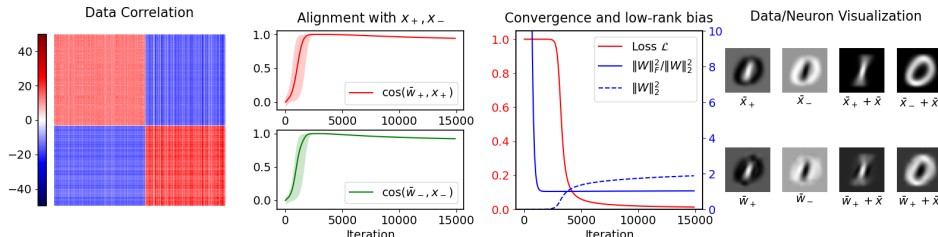

Figure 5: Training two-layer ReLU network under small initialization for binary classification on MNIST digits 0 and 1. (*First Plot*) Data correlation $[\langle x_i, x_j \rangle]_{ij}$ as a heatmap, where the data are reordered by their label (digit 1 first, then digit 0); (*Second Plot*) Alignment between neurons and the aggregate positive/negative data $x_+ = \sum_{i \in \mathcal{I}_+} x_i$, $x_- = \sum_{i \in \mathcal{I}_-} x_i$. (*Third Plot*) The loss $\mathcal{L}$, the stable rank and the squared spectral norm of $W$ during training; (*Fourth Plot*) Visualizing neuron centers $\bar{w}_+, \bar{w}_-$ and data centers $\bar{x}_+, \bar{x}_-$ (at iteration 15000).

different labels. Thus we expect our theorem to make reasonable predictions on the training dynamics with preprocessed data. For the remaining section, we use $x_i, i \in [n]$, to denote the preprocessed (centered) data and use $\bar{x}$ to denote the mean of the original data.

We build a two-layer ReLU network with $h = 50$ neurons and initialize all entries of the weights as $[W]_{ij} \overset{i.i.d.}{\sim} \mathcal{N}(0, \alpha^2), v_j \overset{i.i.d.}{\sim} \mathcal{N}(0, \alpha^2), \forall i \in [n], j \in [h]$ with $\alpha = 10^{-6}$. Then we run gradient descent on both $W$ and $v$ with step size $\eta = 2 \times 10^{-3}$. Notice that here the weights are not initialized to be balanced as in (3). The numerical results are shown in Figure 5.

**Alignment phase**: Without balancedness, one no longer has $\text{sign}(v_j(t)) = \text{sign}(v_j(0))$. With a little abuse of notation, we denote $\mathcal{V}_+(t) = \{j \in [h] : \text{sign}(v_j(t)) > 0\}$ and $\mathcal{V}_+(t) = \{j \in [h] : \text{sign}(v_j(t)) > 0\}$, and we expect that at the end of the alignment phase, neurons in $\mathcal{V}_+$ are aligned with $x_+ = \sum_{i \in \mathcal{I}_+} x_i$, and neurons in $\mathcal{V}_-$ with $x_- = \sum_{i \in \mathcal{I}_-} x_i$. The second plot in Figure 5 shows such an alignment between neurons and $x_+, x_-$. In the top part, the red solid line shows $\cos(\bar{w}_+, x_+)$ during training, where $\bar{w}_+ = \sum_{j \in \mathcal{V}_+} w_j / |\mathcal{V}_+|$, and the shaded region defines the range between $\min_{j \in \mathcal{V}_+} \cos(w_i, x_+)$ and $\max_{j \in \mathcal{V}_+} \cos(w_i, x_+)$. Similarly, in the bottom part, the green solid line shows $\cos(\bar{w}_-, x_-)$ during training, where $\bar{w}_- = \sum_{j \in \mathcal{V}_-} w_j / |\mathcal{V}_-|$, and the shaded region delineates the range between $\min_{j \in \mathcal{V}_-} \cos(w_i, x_-)$ and $\max_{j \in \mathcal{V}_-} \cos(w_i, x_-)$. Initially, every neuron is approximately orthogonal to $x_+, x_-$ due to random initialization. Then all neurons in $\mathcal{V}_+$ ($\mathcal{V}_-$) start to move towards $x_+$ ($x_-$) and achieve good alignment after $\sim$2000 iterations. When the loss starts to decrease, the alignment drops. We conjecture that because Assumption 1 is not exactly satisfied, neurons in $\mathcal{V}_+$ have to fit some negative data, for which $x_+$ is not the best direction.

**Final convergence**: After $\sim 3000$ iterations, the norm $\|W\|_2^2$ starts to grow and the loss decreases, as shown in the third plot in Figure 5. Moreover, the stable rank $\|W\|_F^2 / \|W\|_2^2$ decreases below 2. For this experiment, we almost have $\cos(x_+, x_-) \simeq -1$, thus the neurons in $\mathcal{V}_+$ (aligned with $x_+$) and those in $\mathcal{V}_-$ (aligned with $x_-$) are almost co-linear. Therefore, the stable rank $\|W\|_F^2 / \|W\|_2^2$ is almost 1, as seen from the plot. Finally, at iteration 15000, we visualize the mean neuron $\bar{w}_+ = \sum_{j \in \mathcal{V}_+} w_j / |\mathcal{V}_+|$, $\bar{w}_- = \sum_{j \in \mathcal{V}_-} w_j / |\mathcal{V}_-|$ as grayscale images, and compare them with $\bar{x}_+ = x_+ / |\mathcal{I}_+|$, $x_- = x_- / |\mathcal{I}_-|$, showing good alignment.

**Comparison with other training schemes**: For two-layer ReLU networks, there is another line of work (Brutzkus et al., 2018; Wang et al., 2019) that studies GD/SGD only on the first-layer weights $W$ and keeping the second-layer $v$ fixed throughout training. In Appendix A.5, we compare our training schemes to those in Brutzkus et al. (2018); Wang et al. (2019), and show that while both schemes achieve small training loss, the aforementioned two-phase training (alignment then final convergence) does no happen if only the first-layer in trained.

## 5 CONCLUSION

This paper studies the problem of training a binary classifier via gradient flow on two-layer ReLU networks under small initialization. We consider a training dataset with well-separated input vectors. A careful analysis of the neurons' directional dynamics allows us to provide an upper bound on the time it takes for all neurons to achieve good alignment with the input data. Numerical experiment on classifying two digits from the MNIST dataset correlates with our theoretical findings.

ACKNOWLEDGEMENT

The authors thank the support of the NSF-Simons Research Collaborations on the Mathematical and Scientific Foundations of Deep Learning (NSF grant 2031985), the NSF HDR TRIPODS Institute for the Foundations of Graph and Deep Learning (NSF grant 1934979), the ONR MURI Program (ONR grant 503405-78051), and the NSF CAREER Program (NSF grant 1752362). The authors thank Ziqing Xu and Salma Tarmoun for the insightful discussions.

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

# A    ADDITIONAL EXPERIMENTS

## A.1    ILLUSTRATIVE EXAMPLE

We illustrate our theorem using a toy example: we train a two-layer ReLU network with $h = 50$ neurons under a toy dataset in $\mathbb{R}^2$ (See Figure. 6) that satisfies our Assumption 1, and initialize all entries of the weights as $[W]_{ij} \overset{i.i.d.}{\sim} \mathcal{N}(0, \alpha^2), v_j \overset{i.i.d.}{\sim} \mathcal{N}(0, \alpha^2), \forall i \in [n], j \in [h]$ with $\alpha = 10^{-6}$. Then we run gradient descent on both $W$ and $v$ with step size $\eta = 2 \times 10^{-3}$. Our theorem well predicts the dynamics of neurons at the early stage of the training: aside from neurons that ended up in $\mathcal{S}_{\text{dead}}$, neurons in $\mathcal{V}_+$ reach $\mathcal{S}_+$ and achieve good alignment with $x_+$, and neurons in $\mathcal{V}_-$ are well aligned with $x_-$ in $\mathcal{S}_-$. Note that after alignment, the loss experiences two sharp decreases before it gets close to zero, which is studied and explained in Boursier et al. (2022).

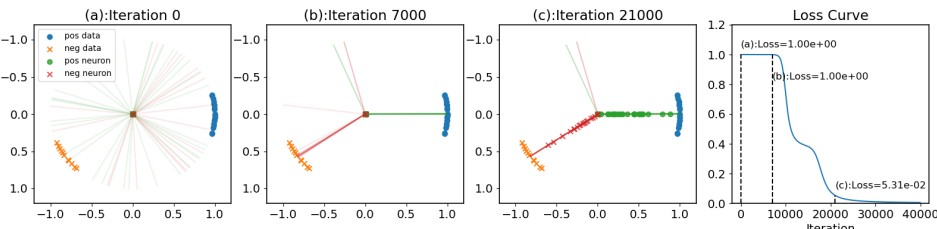

Figure 6: Illustration of gradient descent on two-layer ReLU network with small initialization. The marker represents either a data point or a neuron. Solid lines represent the directions of neurons. (a) at initialization, all neurons have small norm and are pointing in different directions; (b) around the end of the alignment phase, all neurons are in $\mathcal{S}_+, \mathcal{S}_-$, or $\mathcal{S}_{\text{dead}}$. Moreover, neurons in $\mathcal{S}_+$ ($\mathcal{S}_-$) are well aligned with $x_+$ ($x_-$); (c) With good alignment, neurons in $\mathcal{S}_-, \mathcal{S}_+$ start to grow in norm and the loss decreases. When the loss is close to zero, the resulting network has its first-layer weight approximately low-rank.

## A.2    EFFECT OF DATA SEPARABILITY $\mu$

This section investigates the effect of data separability $\mu$ on the time required to achieve the desired alignment as in Theorem 1, through a simple example. we consider a similar setting as in A.1, and explore the cases when data separability $\mu \ll 1$. We expect that as separability $\mu$ decreases, the time for neurons to achieve the desired alignment as in Theorem 1 increases, necessitating a smaller initialization scale. For simplicity, we consider a dataset with only two positive data $(x_1, y_1 = +1), (x_2, y_2 = +1)$.

In Figure 7, we first set $\mu = \langle x_1, x_2 \rangle = \sin(0.1)$, and the neuron alignment is consistent with Theorem 1: positive neurons (that are not dead) eventually enters $\mathcal{S}_+$, activating both data points, and then final convergence follows.

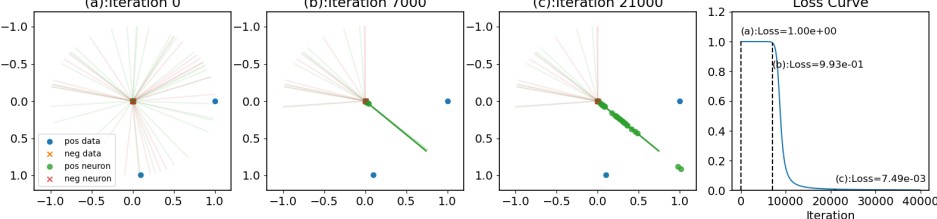

Figure 7: Neural alignment under two data points with small positive correlation: $x_1 = [1, 0], y_1 = +1, \mathbf{x_2} = [\sin(\mathbf{0.1}), \cos(\mathbf{0.1})], y_2 = +1$. The experimental setting is exactly the same as the illustrative example in Appendix A.1 (**initialization scale** $\alpha = 10^{-6}$). The marker represents either a data point or a neuron. Solid lines represent the directions of neurons. In the alignment phase, positive neurons are aligned with $x_+$, and then grow their norm for final convergence.

However, in Figure 8, as we decrease the separability $\mu$ to $\sin(0.001)$ (other settings remain unchanged), the neural alignment becomes slower: 1) at iteration 7000, there are still neurons (that are not dead) outside $\mathcal{S}_+$, namely those aligned with either $x_1$ or $x_2$, while in our previous setting ($\mu = \sin(0.1)$), all neurons (that are not dead) have reached $\mathcal{S}_+$; 2) In this particular instance of the experiment, we also see one neuron remains outside $\mathcal{S}_+$ at the late stage of the training (at iteration 21000). This clearly shows that as data separability $\mu$ decreases, the time needed for all neurons (that are not dead) to reach $\mathcal{S}_+$ increases, and if the initialization scale is not small enough for the alignment phase to hold for a long time, there will be neurons remains outside $\mathcal{S}_+$.

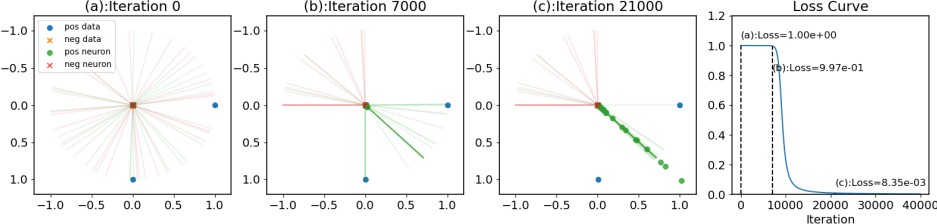

Figure 8: Neural alignment under two data points with tiny positive correlation: $x_1 = [1, 0], y_1 = +1$, $\mathbf{x_2} = [\sin(\mathbf{0.001}), \cos(\mathbf{0.001})], y_2 = +1$. The experimental setting is exactly the same as the illustrative example in Appendix A.1 (**initialization scale** $\alpha = 10^{-6}$). The marker represents either a data point or a neuron. Solid lines represent the directions of neurons. In the alignment phase, positive neurons are aligned with $x_+$, but the alignment is slower

### A.3 NEURON DYNAMICS UNDER ORTHOGONAL DATA

We have seen in the last section how a small $\mu$ affects the neuron dynamics. The orthogonal data assumption studied in Boursier et al. (2022) is precisely the extreme case of $\mu \to 0$, where the neuron behavior changes substantially. We follow exactly the same setting in Appendix A.2 and consider the case of $\mu = 0$.

In Figure 9, we see that $\mathcal{S}_+$ **is no longer the region that contains all (non-dead) positive neurons at the end of the alignment phase**. Depending on where each neuron is initialized, it could end up being in $\mathcal{S}_+$, aligned with $x_1$, or aligned with $x_2$. Moreover, for final convergence, only the neurons ended up in $\mathcal{S}_+$ grow their norms and fit the data, whose number is clearly less than that in the case of $\mu > 0$.

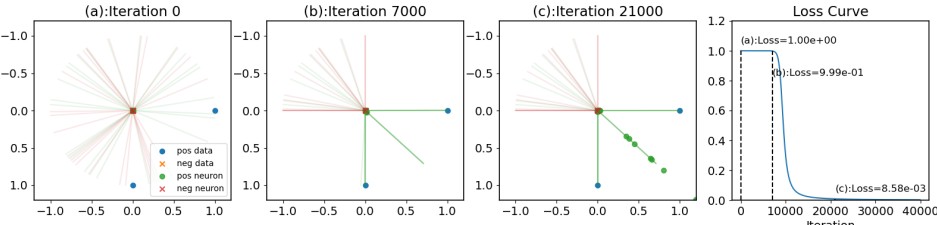

Figure 9: Neural alignment under two orthogonal data points: $x_1 = [1, 0], y_1 = +1, x_2 = [0, 1], y_2 = +1$. The experimental setting is exactly the same as the illustrative example in Appendix A.1. The marker represents either a data point or a neuron. Solid lines represent the directions of neurons. In the alignment phase, positive neurons are aligned with one of these directions: $x_1, x_2, x_+$, and only those aligned with $x_+$ grow their norm for final convergence.

This difference in neurons' dynamical behavior makes the analysis in Boursier et al. (2022) different than ours: First, Boursier et al. (2022) only studies the dynamics of the positive (negative) neurons that initially activate all positive (negative) data, which will end up in $\mathcal{S}_+$ ($\mathcal{S}_-$) and fit the data, and the analysis does not evolve the changes in their activation pattern. In our case, any positive (negative) neurons could potentially end up in $\mathcal{S}_+$ ($\mathcal{S}_-$), and in particular, it will if it initially activates at least

one positive (negative) data, thus it becomes necessary to track the evolution of the activation pattern of all these neurons (novelty in our analysis). Moreover, consider the case that neurons are being randomly initialized, Boursier et al. (2022) requires the set of positive (negative) neurons that initially activate all positive (negative) data being non-empty, which needs the number of neurons $h$ to scale exponentially in number training data $n$ (extremely overparametrized). In our case, we only require $h = \Theta(1)$ (See **Merits of overparametrization** after Theorem 1), a mild overparamerization.

In summary, while Boursier et al. (2022) also provides quantitative analysis on neural alignment under small initialization, it is done under the assumption that all data are orthogonal to each other, leading to a different neuron dynamical behavior than ours. Due to such differences, their analysis cannot be directly applied to the case of orthogonally separable data (ours), for which we develop novel analyses on the evolution of neuron activation patterns (See proof sketch in Section 3.3).

## A.4 ADDITIONAL EXPERIMENTS ON MNIST DATASET

We use exactly the same experimental setting as in the main paper and only use a different pair of digits. The results are as follows:

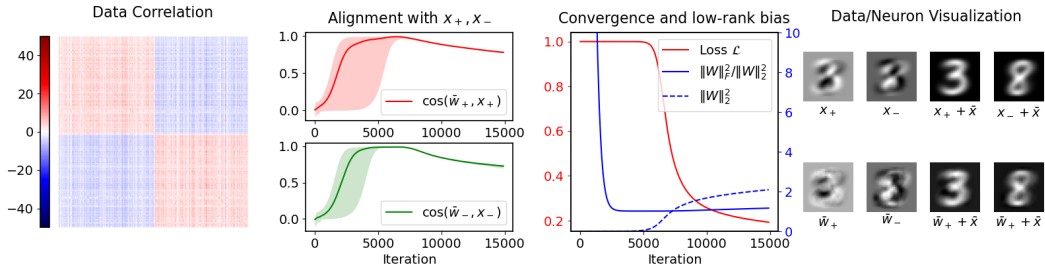

Figure 10: Binary classification on MNIST Digits 3 and 8.

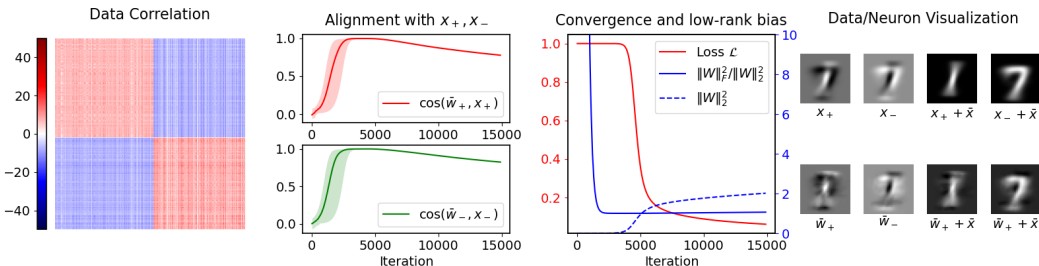

Figure 11: Binary classification on MNIST Digits 1 and 7.

## A.5 DISCUSSION ON THE TWO-PHASE CONVERGENCE

With the same two-digit MNIST dataset in Section 4, we further discuss the two-phase convergence under small initialization. We use a two-layer ReLU network with $h = 50$ neurons and initialize all entries of the weights as $[W]_{ij} \overset{i.i.d.}{\sim} \mathcal{N}\left(0, \alpha^2\right), v_j \overset{i.i.d.}{\sim} \mathcal{N}\left(0, \alpha^2\right), \forall i \in [n], j \in [h]$ with $\alpha = 10^{-6}$. Then we run stochastic gradient descent (SGD) with batch size 2000 on both $W$ and $v$ with step size $\eta = 2 \times 10^{-3}$. For comparison, we also consider the training schemes studied in Brutzkus et al. (2018); Wang et al. (2019), where only the first-layer weight $W$ is trained starting from a small initialization $[W]_{ij} \overset{i.i.d.}{\sim} \mathcal{N}\left(0, \alpha^2\right)$, and $v_j$ are chosen to be either $+1$ or $-1$ with equal probability, then fixed throughout training.

We consider the changes in neuron norms and directions separately. In particular, these quantities are defined as

$$\sum_i \frac{d}{dt} \|w_j\|^2 \bigg|_{\dot{w}_j = -\nabla_{w_j}\mathcal{L}} = \sum_i 2\left\langle -\nabla_{w_j}\mathcal{L}, w_j \right\rangle \qquad \text{(changes in neuron norms)}$$

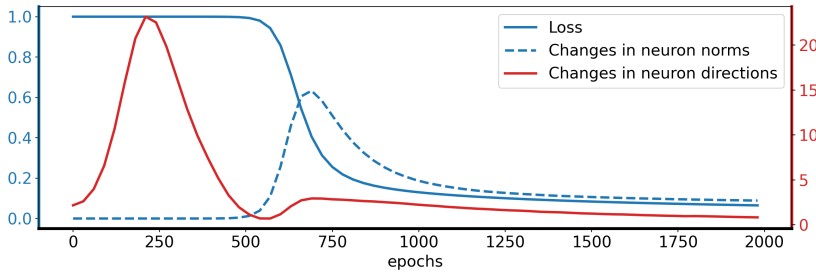

Figure 12: Two-phase training under small initialization (SGD with a batch size of 2000, step size of $2 \times 10^{-3}$). At the early phase of the training, only neuron directions are changing while neurons' norms do not grow. In the second stage, neurons start to grow their norms and loss starts to decrease. See Appendix A.5 for the precise definitions of "changes in neuron norms" and "changes in neuron directions"

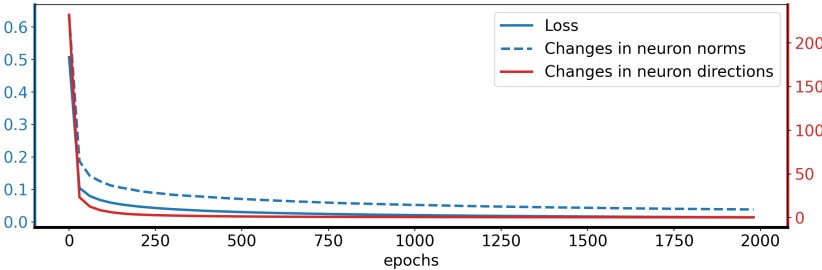

Figure 13: No two-phase training when only the first layer is trained (SGD with a batch size of 2000, step size of $2 \times 10^{-3}$).

$$\sum_i \left\| \frac{d}{dt} \frac{w_j}{\|w_j\|} \bigg|_{\dot{w}_j = -\nabla_{w_j}\mathcal{L}} \right\| = \sum_i \left\| \mathcal{P}_{w_j} \left( \frac{-\nabla_{w_j}\mathcal{L}}{\|w_j\|} \right) \right\| , \qquad \text{(changes in neuron directions)}$$

and they measure, at the end of every epoch, how much the neuron norms and directions will change if one uses a one-step full gradient descent with a small step size.

**Training both layers**: In Figure 12, we show the changes in neuron norms and directions over the training trajectory when we run stochastic gradient descent (SGD) on both first- and second-layer weights. The two-phase (alignment phase then final convergence) is clearly shown by comparing the relative scale of changes in neuron norms and directions in different phases of the training.

**Training only the first layer**: In Figure 13, we show the changes in neuron norms and directions over the training trajectory when we run stochastic gradient descent (SGD) on ONLY the first-layer weights (Brutzkus et al., 2018; Wang et al., 2019). The plot indicates that two-phase training does not happen in this case.

# B  PROOF OF LEMMA 1: NEURON DYNAMICS UNDER SMALL INITIALIZATION

The following property of $\ell$ (exponential loss $\ell(y, \hat{y}) = \exp(-y\hat{y})$ or logistic loss $\ell(y, \hat{y}) = 2\log(1 + \exp(-y\hat{y}))$) will be used throughout the Appendix for proofs of several results:

**Lemma 2.** *For $\ell$, we have*

$$| - \nabla_{\hat{y}} \ell(y, \hat{y}) - y| \leq 2|\hat{y}|, \forall y \in \{+1, -1\}, \quad \forall |\hat{y}| \leq 1. \tag{9}$$

*Proof.* **Exponential loss: when $\ell(y, \hat{y}) = \exp(-y\hat{y})$:**

$$
\begin{aligned}
| - \nabla_{\hat{y}} \ell(y, \hat{y}) - y| &= |y\exp(-y\hat{y}) - y| \\
&\leq |y||\exp(-y\hat{y}) - 1| \\
&\leq |\exp(-y\hat{y}) - 1| \leq 2|\hat{y}|,
\end{aligned}
$$

where the last inequality is due to the fact that $2x \geq \max\{1 - \exp(-x), \exp(x) - 1\}, \forall x \in [0, 1]$.
**Logistic loss: when $\ell(y, \hat{y}) = 2\log(1 + \exp(-y\hat{y}))$:**

$$
\begin{aligned}
| - \nabla_{\hat{y}} \ell(y, \hat{y}) - y| &= \left| 2y \frac{\exp(-y\hat{y})}{1 + \exp(-y\hat{y})} - y \right| \\
&= \left| \frac{y\exp(-y\hat{y}) - y}{1 + \exp(-y\hat{y})} \right| \\
&\leq |y||\exp(-y\hat{y}) - 1| \\
&\leq |\exp(-y\hat{y}) - 1| \leq 2|\hat{y}|,
\end{aligned}
$$

$\square$

**Remark 4.** *More generally, our subsequent results regarding neuron dynamics under small initialization hold for any loss function that satisfies the condition stated in Lemma 2, which includes the $l_2$ loss $\ell(y, \hat{y}) = \frac{1}{2}(y - \hat{y})^2$ studied in Boursier et al. (2022).*

## B.1  FORMAL STATEMENT

Our results for neuron direction dynamics during the early phase of the training will be stated for networks with any $\alpha$-leaky ReLU activation $\sigma(x) = \max\{x, \alpha x\}$ with $\alpha \in [0, 1]$. In particular, it is the ReLU activation when $\alpha = 0$, which is the activation function we considered in the main paper, and it is the linear activation when $\alpha = 1$.

Denote: $X_{\max} = \max_i \|x_i\|, W_{\max} = \max_j \|[W_0]_{:,j}\|$. The formal statement of Lemma 1 is as follow:

**Lemma 1.** *Let the activation function be an $\alpha$-leaky ReLU activation $\sigma(x) = \max\{x, \alpha x\}$. Given some initialization from (3), for any $\epsilon \leq \frac{1}{4\sqrt{h}X_{\max}W_{\max}^2}$, then any solution to the gradient flow dynamics (2) satisfies that $\forall t \leq T = \frac{1}{4nX_{\max}} \log \frac{1}{\sqrt{h}\epsilon}$,*

$$\max_j \left\| \frac{d}{dt} \frac{w_j(t)}{\|w_j(t)\|} - \text{sign}(v_j(0)) \mathcal{P}_{w_j(t)} x_a(w_j(t)) \right\| \leq 4\epsilon n\sqrt{h} X_{\max}^2 W_{\max}^2,$$

*where*

$$x_a(w_j) = \sum_{i=1}^n x_i y_i \sigma'(\langle x_i, w_j \rangle) = \sum_{i:\langle x_i, w_j \rangle > 0} x_i y_i + \alpha \sum_{i:\langle x_i, w_j \rangle \leq 0} x_i y_i.$$

With Lemma 1, and set $\alpha = 0$, we obtain the results stated in the main paper. Lemma 1 is a direct result of the following two lemmas.

**Lemma 3.** *Let the activation function be an $\alpha$-leaky ReLU activation $\sigma(x) = \max\{x, \alpha x\}$. Given some initialization in (3), then for any $\epsilon \leq \frac{1}{4\sqrt{h}X_{\max}W_{\max}^2}$, any solution to the gradient flow dynamics (2) satisfies*

$$\max_j \|w_j(t)\|^2 \leq \frac{2\epsilon W_{\max}^2}{\sqrt{h}}, \quad \max_i |f(x_i; W(t), v(t))| \leq 2\epsilon\sqrt{h} X_{\max} W_{\max}^2, \tag{10}$$

$\forall t \leq \frac{1}{4nX_{\max}} \log \frac{1}{\sqrt{h}\epsilon}$.

**Lemma 4.** *Let the activation function be an $\alpha$-leaky ReLU activation $\sigma(x) = \max\{x, \alpha x\}$. Consider any solution to the gradient flow dynamic* (2) *starting from initialization* (3). *Whenever* $\max_i |f(x_i; W, v)| \leq 1$, *we have,* $\forall i \in [n]$,

$$\left\| \frac{d}{dt} \frac{w_j}{\|w_j\|} - \operatorname{sign}(v_j(0)) \left( I - \frac{w_j w_j^\top}{\|w_j\|^2} \right) x_a(w_j) \right\| \leq 2n X_{\max} \max_i |f(x_i; W, v)|, \quad (11)$$

*where*

$$x_a(w_j) = \sum_{i=1}^n x_i y_i \sigma'(\langle x_i, w_j \rangle) = \sum_{i: \langle x_i, w_j \rangle > 0} x_i y_i + \alpha \sum_{i: \langle x_i, w_j \rangle \leq 0} x_i y_i.$$

**Remark 5.** *By stating our approximation results for neuron directional dynamics with any $\alpha$-leaky ReLU activation function, we highlight that even for some networks with other activation functions than ReLU, there is a similar notion of neuron alignment at the early stage of the training, and the analytical tools used in this paper can be applied to them. However, we note that our main results (Theorem 1) will not directly apply as the neuron directional dynamics have changed as we consider an activation function different than ReLU (see the general definition of $x_a(w_j)$), and additional efforts are required to establish the directional convergence for general leaky-ReLU functions.*

### B.2 PROOF OF LEMMA 3: BOUNDS ON NEURON NORMS

*Proof of Lemma 3.* Under gradient flow, we have

$$\frac{d}{dt} w_j = -\sum_{i=1}^n \mathbb{1}_{\langle x_i, w_j \rangle > 0} \nabla_{\hat{y}} \ell(y_i, f(x_i; W, v)) x_i v_j. \quad (12)$$

Balanced initialization enforces $v_j = \operatorname{sign}(v_j(0))\|w_j\|$, hence

$$\frac{d}{dt} w_j = -\sum_{i=1}^n \sigma'(\langle x_i, w_j \rangle) \nabla_{\hat{y}} \ell(y_i, f(x_i; W, v)) x_i \operatorname{sign}(v_j(0))\|w_j\|. \quad (13)$$

Let $T := \inf\{t : \max_i |f(x_i; W(t), v(t))| > 2\epsilon\sqrt{h} X_{\max} W_{\max}^2\}$, then $\forall t \leq T, j \in [h]$, we have

$$\begin{aligned}
\frac{d}{dt} \|w_j\|^2 &= \left\langle w_j, \frac{d}{dt} w_j \right\rangle \\
&= -2 \sum_{i=1}^n \sigma'(\langle x_i, w_j \rangle) \nabla_{\hat{y}} \ell(y_i, f(x_i; W, v)) \langle x_i, w_j \rangle \operatorname{sign}(v_j(0))\|w_j\| \\
&\leq 2 \sum_{i=1}^n |\nabla_{\hat{y}} \ell(y_i, f(x_i; W, v))| |\langle x_i, w_j \rangle| \|w_j\| \\
&\leq 2 \sum_{i=1}^n (|y_i| + 2|f(x_i; W, v)|) |\langle x_i, w_j \rangle| \|w_j\| && \text{(by Lemma 2)} \\
&\leq 2 \sum_{i=1}^n (1 + 4\epsilon\sqrt{h} X_{\max} W_{\max}^2) |\langle x_i, w_j \rangle| \|w_j\| && \text{(Since } t \leq T\text{)} \\
&\leq 2 \sum_{i=1}^n (1 + 4\epsilon\sqrt{h} X_{\max} W_{\max}^2) \|x_i\| \|w_j\|^2 \\
&\leq 2n(X_{\max} + 4\epsilon\sqrt{h} X_{\max}^2 W_{\max}^2) \|w_j\|^2. && (14)
\end{aligned}$$

Let $\tau_j := \inf\{t : \|w_j(t)\|^2 > \frac{2\epsilon W_{\max}^2}{\sqrt{h}}\}$, and let $j^* := \arg\min_j \tau_j$, then $\tau_{j^*} = \min_j \tau_j \leq T$ due to the fact that

$$|f(x_i; W, v)| = \left| \sum_{j \in [h]} \sigma'(\langle x_i, w_j \rangle) v_j \langle w_j, x_i \rangle \right| \leq \sum_{j \in [h]} \|w_j\|^2 \|x_i\| \leq h X_{\max} \max_{j \in [h]} \|w_j\|^2,$$

which implies "$|f(x_i; W(t), v(t))| > 2\epsilon\sqrt{h}X_{\max}W_{\max}^2 \Rightarrow \exists j, s.t. \|w_j(t)\|^2 > \frac{2\epsilon W_{\max}^2}{\sqrt{h}}$".

Then for $t \leq \tau_{j^*}$, we have

$$\frac{d}{dt}\|w_{j^*}\|^2 \leq 2n(X_{\max} + 4\epsilon\sqrt{h}X_{\max}^2 W_{\max}^2)\|w_{j^*}\|^2. \tag{15}$$

By Grönwall's inequality, we have $\forall t \leq \tau_{j^*}$

$$\begin{aligned}
\|w_{j^*}(t)\|^2 &\leq \exp\left(2n(X_{\max} + 4\epsilon\sqrt{h}X_{\max}^2 W_{\max}^2)t\right)\|w_{j^*}(0)\|^2, \\
&= \exp\left(2n(X_{\max} + 4\epsilon\sqrt{h}X_{\max}^2 W_{\max}^2)t\right)\epsilon^2\|[W_0]_{:,j^*}\|^2 \\
&\leq \exp\left(2n(X_{\max} + 4\epsilon\sqrt{h}X_{\max}^2 W_{\max}^2)t\right)\epsilon^2 W_{\max}^2.
\end{aligned}$$

Suppose $\tau_{j^*} < \frac{1}{4nX_{\max}}\log\left(\frac{1}{\sqrt{h}\epsilon}\right)$, then by the continuity of $\|w_{j^*}(t)\|^2$, we have

$$\begin{aligned}
\frac{2\epsilon W_{\max}^2}{\sqrt{h}} \leq \|w_{j^*}(\tau_{j^*})\|^2 &\leq \exp\left(2n(X_{\max} + 4\epsilon\sqrt{h}X_{\max}^2 W_{\max}^2)\tau_{j^*}\right)\epsilon^2 W_{\max}^2 \\
&\leq \exp\left(2n(X_{\max} + 4\epsilon\sqrt{h}X_{\max}^2 W_{\max}^2)\frac{1}{4nX_{\max}}\log\left(\frac{1}{\sqrt{h}\epsilon}\right)\right)\epsilon^2 W_{\max}^2 \\
&\leq \exp\left(\frac{1 + 4\epsilon\sqrt{h}X_{\max}W_{\max}^2}{2}\log\left(\frac{1}{\sqrt{h}\epsilon}\right)\right)\epsilon^2 W_{\max}^2 \\
&\leq \exp\left(\log\left(\frac{1}{\sqrt{h}\epsilon}\right)\right)\epsilon^2 W_{\max}^2 = \frac{\epsilon W_{\max}^2}{\sqrt{h}},
\end{aligned}$$

which leads to a contradiction $2\epsilon \leq \epsilon$. Therefore, one must have $T \geq \tau_{j^*} \geq \frac{1}{4nX_{\max}}\log\left(\frac{1}{\sqrt{h}\epsilon}\right)$. This finishes the proof. $\square$

### B.3 Proof of Lemma 4: Directional Dynamics of Neurons

*Proof of Lemma 4.* As we showed in the proof for Lemma 3, under balanced initialization,

$$\frac{d}{dt}w_j = -\sum_{i=1}^n \mathbb{1}_{\langle x_i, w_j\rangle > 0}\nabla_{\hat{y}}\ell(y_i, f(x_i; W, v))x_i \text{sign}(v_j(0))\|w_j\|. \tag{16}$$

Then for any $i \in [n]$,

$$\begin{aligned}
\frac{d}{dt}\frac{w_j}{\|w_j\|} &= -\text{sign}(v_j(0))\sum_{i=1}^n \mathbb{1}_{\langle x_i, w_j\rangle > 0}\nabla_{\hat{y}}\ell(y_i, f(x_i; W, v))\left(x_i - \frac{\langle x_i, w_j\rangle}{\|w_j\|^2}w_j\right) \\
&= -\text{sign}(v_j(0))\sum_{i:\langle x_i, w_j\rangle > 0}\nabla_{\hat{y}}\ell(y_i, f(x_i; W, v))\left(x_i - \frac{\langle x_i, w_j\rangle}{\|w_j\|^2}w_j\right) \\
&= -\text{sign}(v_j(0))\left(I - \frac{w_j w_j^\top}{\|w_j\|^2}\right)\left(\sum_{i=1}^n \sigma'(\langle x_i, w_j\rangle)\nabla_{\hat{y}}\ell(y_i, f(x_i; W, v))x_i\right).
\end{aligned}$$

Therefore, whenever $\max_i |f(x_i; W, v)| \leq 1$,

$$\begin{aligned}
&\left\|\frac{d}{dt}\frac{w_j}{\|w_j\|} - \text{sign}(v_j(0))\left(I - \frac{w_j w_j^\top}{\|w_j\|^2}\right)x_a(w_j)\right\| \\
&= \left\|\text{sign}(v_j(0))\left(\sum_{i=1}^n \sigma'(\langle x_i, w_j\rangle)\left(\nabla_{\hat{y}}\ell(y_i, f(x_i; W, v)) + y_i\right)x_i\right)\right\| \\
&\leq \sum_{i=1}^n |\nabla_{\hat{y}}\ell(y_i, f(x_i; W, v)) + y_i| \cdot \|x_i\| \\
&\leq \sum_{i=1}^n 2|f(x_i; W, v)| \cdot \|x_i\| \leq 2nM_x \max_i |f(x_i; W, v)|. \tag{17}
\end{aligned}$$

□

## C  PROOF FOR THEOREM 1: EARLY ALIGNMENT PHASE

We break the proof of Theorem 1 into two parts: In Appendix C we prove the first part regarding directional convergence. Then in Appendix D we prove the remaining statement on final convergence and low-rank bias.

### C.1  AUXILIARY LEMMAS

The first several Lemmas concern mostly some conic geometry given the data assumption:

Consider the following conic hull

$$K = \mathcal{CH}(\{x_i y_i, i \in [n]\}) = \left\{ \sum_{i=1}^{n} a_i x_i y_i : a_i \geq 0, i \in [n] \right\}. \tag{18}$$

It is clear that $x_i y_i \in K, \forall i$, and $x_a(w) \in K, \forall w$. The following lemma shows any pair of vectors in $K$ is $\mu$-coherent.

**Lemma 5.** $\cos(z_1, z_2) \geq \mu, \forall 0 \neq z_1, z_2 \in K$.

*Proof.* Since $z_1, z_2 \in K$, we let $z_1 = \sum_{i=1}^{n} x_i y_i a_{1i}$, and $z_2 = \sum_{j=1}^{n} x_j y_j a_{2j}$, where $a_{1i}, a_{2j} \geq 0$ but not all of them.

$$\cos(z_1, z_2) = \frac{1}{\|z_1\|\|z_2\|} \langle z_1, z_2 \rangle = \frac{1}{\|z_1\|\|z_2\|} \sum_{i,j \in [n]} a_{1i} a_{2j} \langle x_i y_i, x_j y_j \rangle$$

$$= \frac{\sum_{i,j \in [n]} \|x_i\|\|x_j\| a_{1i} a_{2j} \mu}{\|z_1\|\|z_2\|} \geq \mu,$$

where the last inequality is due to

$$\|z_1\|\|z_2\| \leq \left( \sum_{i=1}^{n} \|x_i\| a_{1i} \right) \left( \sum_{j=1}^{n} \|x_j\| a_{2j} \right) = \sum_{i,j \in [n]} \|x_i\|\|x_j\| a_{1i} a_{2j}.$$

$\square$

The following lemma is some basic results regarding $\mathcal{S}_+$ and $\mathcal{S}_-$:

**Lemma 6.** $\mathcal{S}_+$ and $\mathcal{S}_-$ are convex cones (excluding the origin).

*Proof.* Since $\mathbb{1}_{\langle x_i, z \rangle} = \mathbb{1}_{\langle x_i, az \rangle}, \forall i \in [n], a > 0$, $\mathcal{S}_+, \mathcal{S}_-$ are cones. Moreover, $\langle x_i, z_1 \rangle > 0$ and $\langle x_i, z_2 \rangle > 0$ implies $\langle x_i, a_1 z_1 + a_2 z_2 \rangle > 0, \forall a_1, a_2 > 0$, thus $\mathcal{S}_+, \mathcal{S}_-$ are convex cones. $\square$

Now we consider the complete metric space $\mathbb{S}^{D-1}$ (w.r.t. $\arccos(\langle \cdot, \cdot \rangle)$) and we are interested in its subsets $K \cap \mathbb{S}^{D-1}, \mathcal{S}_+ \cap \mathbb{S}^{D-1}$, and $\mathcal{S}_- \cap \mathbb{S}^{D-1}$. First, we have (we use $\mathrm{Int}(S)$ to denote the interior of $S$)

**Lemma 7.** $K \cap \mathbb{S}^{D-1} \subset \mathrm{Int}(\mathcal{S}_+ \cap \mathbb{S}^{D-1})$, and $-K \cap \mathbb{S}^{D-1} \subset \mathrm{Int}(\mathcal{S}_- \cap \mathbb{S}^{D-1})$

*Proof.* Consider any $x_c = \sum_{j=1}^{n} a_j x_j y_j \in K \cap \mathbb{S}^{D-1}$, For any $x_i, y_i, i \in [n]$, we have

$$\langle x_c, x_i \rangle = \sum_{i=j}^{n} a_j \|x_j\| \left\langle \frac{x_j y_j}{\|x_j\|}, \frac{x_i y_i}{\|x_i\|} \right\rangle \frac{\|x_i\|}{y_i}$$

$$\geq \mu y_i \|x_i\| \sum_{i=j}^{n} a_j \|x_j\| \begin{cases} \geq \mu X_{\min} > 0, & y_i > 0 \\ \leq -\mu X_{\min} < 0, & y_i < 0 \end{cases}.$$

Depending on the sign of $y_i$, we have either

$$\langle x_c, x_i \rangle = \sum_{i=j}^{n} a_j \|x_j\| \left\langle \frac{x_j y_j}{\|x_j\|}, \frac{x_i y_i}{\|x_i\|} \right\rangle \frac{\|x_i\|}{y_i} \geq \mu \frac{\|x_i\|}{y_i} \sum_{i=j}^{n} a_j \|x_j\| \geq \mu X_{\min} > 0, \ (y_i = +1)$$

or

$$\langle x_c, x_i \rangle = \sum_{i=j}^{n} a_j \|x_j\| \left\langle \frac{x_j y_j}{\|x_j\|}, \frac{x_i y_i}{\|x_i\|} \right\rangle \frac{\|x_i\|}{y_i} \leq \mu \frac{\|x_i\|}{y_i} \sum_{i=j}^{n} a_j \|x_j\| \leq -\mu X_{\min} < 0, \ (y_i = -1)$$

where we use the fact that $1 = \|x_c\| = \|\sum_{j=1}^{n} a_j x_j y_j\| \leq \sum_{j=1}^{n} a_j \|x_j\|$. This already tells us $x_c \in \mathcal{S}_+ \cap \mathbb{S}^{D-1}$.

Since $f_i(z) = \langle z, x_i \rangle$ is a continuous function of $z \in \mathbb{S}^{D-1}$. There exists an open ball $\mathcal{B}(x_c, \delta_i)$ centered at $x_c$ with some radius $\delta_i > 0$, such that $\forall z \in \mathcal{B}(x_c, \delta_i)$, one have $|f_i(z) - f_i(x_c)| \leq \frac{\mu X_{\min}}{2}$, which implies

$$\langle z, x_i \rangle \begin{cases} \geq \mu X_{\min}/2 > 0, & y_i > 0 \\ \leq -\mu X_{\min}/2 < 0, & y_i < 0 \end{cases}.$$

Hence $\cap_{i=1}^{n} \mathcal{B}\left(\frac{x_c}{\|x_c\|}, \delta_i\right) \in \mathcal{S}_+ \cap \mathbb{S}^{D-1}$. Therefore, $x_c \in \text{Int}(\mathcal{S}_+ \cap \mathbb{S}^{D-1})$. This suffices to show $K \cap \mathbb{S}^{D-1} \subset \text{Int}(\mathcal{S}_+ \cap \mathbb{S}^{D-1})$. The other statement $-K \cap \mathbb{S}^{D-1} \subset \text{Int}(\mathcal{S}_- \cap \mathbb{S}^{D-1})$ is proved similarly. $\square$

The following two lemmas are some direct results of Lemma 7.

**Lemma 8.** $\exists \zeta_1 > 0$ *such that*

$$\mathcal{S}_{x_+}^{\zeta_1} \subset \mathcal{S}_+, \qquad \mathcal{S}_{x_-}^{\zeta_1} \subset \mathcal{S}_-, \tag{19}$$

*where* $\mathcal{S}_x^{\zeta} := \{z \in \mathbb{R}^D : \cos(z, x) \geq \sqrt{1 - \zeta}\}$.

*Proof.* By Lemma 7, $\frac{x_+}{\|x_+\|} \in K \subset \text{Int}(S_+)$. Since $\mathbb{S}^{D-1}$ is a complete metric space (w.r.t $\arccos \langle \cdot, \cdot \rangle$), there exists a open ball centered at $\frac{x_+}{\|x_+\|}$ of some radius $\arccos(\sqrt{1 - \zeta_1})$ that is a subset of $\mathcal{S}_+$, from which one can show $\mathcal{S}_{x_+}^{\zeta_1} \subset \mathcal{S}_+$. The other statement $\mathcal{S}_{x_-}^{\zeta_1} \subset \mathcal{S}_-$ simply comes from the fact that $x_+ = -x_-$ and $\text{Int}(\mathcal{S}_+) = -\text{Int}(\mathcal{S}_-)$. $\square$

**Lemma 9.** $\exists \xi > 0$*, such that*

$$\sup_{x_1 \in K \cap \mathbb{S}^{D-1}, x_2 \in (\mathcal{S}_+ \cap \mathbb{S}^{D-1})^c \cap (\mathcal{S}_- \cap \mathbb{S}^{D-1})^c} |\cos(x_1, x_2)| \leq \sqrt{1 - \xi}. \tag{20}$$

*($S^c$ here is defined to be $\mathbb{S}^{D-1} - S$, the set complement w.r.t. complete space $\mathbb{S}^{D-1}$)*

*Proof.* Notice that

$$\sup_{x_1 \in K \cap \mathbb{S}^{D-1}, x_2 \in (\text{Int}(\mathcal{S}_+ \cap \mathbb{S}^{D-1}))^c} \langle x_1, x_2 \rangle = \inf_{x_1 \in K \cap \mathbb{S}^{D-1}, x_2 \in (\text{Int}(\mathcal{S}_+ \cap \mathbb{S}^{D-1}))^c} \arccos \langle x_1, x_2 \rangle.$$

Since $\mathbb{S}^{D-1}$ is a complete metric space (w.r.t $\arccos \langle \cdot, \cdot \rangle$) and $K \cap \mathbb{S}^{D-1}$ and $x_2 \in (\text{Int}(\mathcal{S}_+ \cap \mathbb{S}^{D-1}))^c$ are two of its compact subsets. Suppose

$$\inf_{x_1 \in K \cap \mathbb{S}^{D-1}, x_2 \in x_2 \in (\text{Int}(\mathcal{S}_+ \cap \mathbb{S}^{D-1}))^c} \arccos \langle x_1, x_2 \rangle = 0,$$

then $\exists x_1 \in K \cap \mathbb{S}^{D-1}, x_2 \in (\text{Int}(\mathcal{S}_+ \cap \mathbb{S}^{D-1}))^c$ such that $\arccos \langle x_1, x_2 \rangle = 0$, i.e., $x_1 = x_2$, which contradicts the fact that $K \cap \mathbb{S}^{D-1} \subseteq \text{Int}(\mathcal{S}_+ \cap \mathbb{S}^{D-1})$ (Lemma 7). Therefore, we have the infimum strictly larger than zero, then

$$\sup_{x_1 \in K \cap \mathbb{S}^{D-1}, x_2 \in (S_+ \cap \mathbb{S}^{D-1})^c} \langle x_1, x_2 \rangle \leq \sup_{x_1 \in K \cap \mathbb{S}^{D-1}, x_2 \in (\text{Int}(\mathcal{S}_+ \cap \mathbb{S}^{D-1}))^c} \langle x_1, x_2 \rangle < 1. \tag{21}$$

Similarly, one can show that

$$\sup_{x_1 \in -K \cap \mathbb{S}^{D-1}, x_2 \in (\mathcal{S}_- \cap \mathbb{S}^{D-1})^c} \langle x_1, x_2 \rangle < 1. \tag{22}$$

Finally, find $\xi < 1$ such that

$$\max \left\{ \sup_{x_1 \in K \cap \mathbb{S}^{D-1}, x_2 \in (\mathcal{S}_+ \cap \mathbb{S}^{D-1})^c} \langle x_1, x_2 \rangle, \sup_{x_1 \in -K \cap \mathbb{S}^{D-1}, x_2 \in (\mathcal{S}_- \cap \mathbb{S}^{D-1})^c} \langle x_1, x_2 \rangle \right\} = \sqrt{1 - \xi},$$

then for any $x_1 \in K \cap \mathbb{S}^{D-1}$ and $x_2 \in (\mathcal{S}_+ \cap \mathbb{S}^{D-1})^c \cap (\mathcal{S}_- \cap \mathbb{S}^{D-1})^c$, we have

$$-\sqrt{1-\xi} \leq \langle x_1, x_2 \rangle \leq \sqrt{1-\xi},$$

which is the desired result. $\qquad\square$

The remaining two lemmas are technical but extensively used in the main proof.

**Lemma 10.** *Consider any solution to the gradient flow dynamic* (2) *starting from initialization* (3). *Let $x_r \in \mathbb{S}^{n-1}$ be some reference direction, we define*

$$\psi_{rj} = \left\langle x_r, \frac{w_j}{\|w_j\|} \right\rangle, \ \psi_{ra} = \left\langle x_r, \frac{x_a(w_j)}{\|x_a(w_j)\|} \right\rangle, \ \psi_{aj} = \left\langle \frac{w_j}{\|w_j\|}, \frac{x_a(w_j)}{\|x_a(w_j)\|} \right\rangle, \quad (23)$$

*where $x_a(w_j) = \sum_{i:\langle x_i, w_j \rangle > 0} y_i x_i$.*

*Whenever $\max_i |f(x_i; W, v)| \leq 1$, we have*

$$\left| \frac{d}{dt} \psi_{rj} - \operatorname{sign}(v_j(0)) \left( \psi_{ra} - \psi_{rj}\psi_{aj} \right) \|x_a(w_j)\| \right| \leq 2n X_{\max} \max_i |f(x_i; W, v)|. \quad (24)$$

*Proof.* A simple application of Lemma 4, together with Cauchy-Schwartz:

$$\left| \frac{d}{dt} \psi_{rj} - \operatorname{sign}(v_j(0)) \left( \psi_{ra} - \psi_{rj}\psi_{aj} \right) \|x_a(w_j)\| \right|$$

$$= \left| x_r^\top \left( \frac{d}{dt} \frac{w_j}{\|w_j\|} - \operatorname{sign}(v_j(0)) \left( I - \frac{w_j w_j^\top}{\|w_j\|^2} \right) \left( \sum_{i:\langle x_i, w_j \rangle > 0} y_i x_i \right) \right) \right| \leq 2n X_{\max} \max_i |f(x_i; W, v)|.$$

$\qquad\square$

**Lemma 11.**

$$\|x_a(w)\| \geq \sqrt{\mu} n_a(w) X_{\min}, \quad (25)$$

*where $n_a(w) = |\{i \in [n] : \langle x_i, w \rangle > 0\}|$.*

*Proof.* Let $\mathcal{I}_a(w)$ denote $\{i \in [n] : \langle x_i, w \rangle > 0\}$, then

$$\|x_a(w)\| = \left\| \sum_{i:\langle x_i, w \rangle > 0} x_i y_i \right\| = \sqrt{ \sum_{i \in \mathcal{I}_a(w)} \|x_i\|^2 y_i^2 + \sum_{i,j \in \mathcal{I}_a(w), i<j} \|x_i\|\|x_j\| \left\langle \frac{x_i y_i}{\|x_i\|}, \frac{x_j y_j}{\|x_j\|} \right\rangle }$$

$$\geq \sqrt{ \sum_{i \in \mathcal{I}_a(w)} \|x_i\|^2 y_i^2 + \sum_{i,j \in \mathcal{I}_a(w), i<j} \|x_i\|\|x_j\||y_i||y_j|\mu }$$

$$\geq \sqrt{ n_a(w) X_{\min}^2 + \mu n_a(w)(n_a(w)-1) X_{\min}^2 }$$

$$\geq \sqrt{ n_a(w)(1 + \mu(n_a(w)-1)) } X_{\min}$$

$$\geq \sqrt{\mu} n_a(w) X_{\min}.$$

$\qquad\square$

## C.2 PROOF FOR EARLY ALIGNMENT PHASE

*Proof of Theorem 1: First Part.* Given some initialization in (3), by Assumption 2, $\exists \zeta_2 > 0$, such that

$$\max_{j \in \mathcal{V}_+} \cos(w_j(0), x_-) < \sqrt{1 - \zeta_2}, \quad \max_{j \in \mathcal{V}_-} \cos(w_j(0), x_+) < \sqrt{1 - \zeta_2}. \quad (26)$$

We define $\zeta := \max\{\zeta_1, \zeta_2\}$, where $\zeta_1$ is from Lemma 8. In addition, by Lemma 9, $\exists \xi > 0$, such that

$$\sup_{x_1 \in K \cap \mathbb{S}^{D-1}, x_2 \in \mathcal{S}_-^c \cap \mathcal{S}_+^c \cap \mathbb{S}^{D-1}} |\cos(x_1, x_2)| \leq \sqrt{1 - \xi}. \quad (27)$$

We pick a initialization scale $\epsilon$ that satisfies:

$$\epsilon \leq \min\left\{ \frac{\min\{\mu, \zeta, \xi\}\sqrt{\mu}X_{\min}}{4\sqrt{h}nX_{\max}^2W_{\max}^2}, \frac{1}{\sqrt{h}}\exp\left(-\frac{64nX_{\max}}{\min\{\zeta, \xi\}\sqrt{\mu}X_{\min}}\log n\right)\right\} \leq \frac{1}{4\sqrt{h}X_{\max}W_{\max}^2}\,. \tag{28}$$

By Lemma 3, $\forall t \leq T = \frac{1}{4nX_{\max}}\log\frac{1}{\sqrt{h}\epsilon}$, we have

$$\max_i |f(x_i; W, v)| \leq \frac{\min\{\mu, \zeta, \xi\}\sqrt{\mu}X_{\min}}{4nX_{\max}}\,, \tag{29}$$

which is the key to analyzing the alignment phase. For the sake of simplicity, we only discuss the analysis of neurons in $\mathcal{V}_+$ here, the proof for neurons in $\mathcal{V}_-$ is almost identical.

**Activation pattern evolution:** Pick any $w_j$ in $\mathcal{V}_+$ and pick $x_r = x_i y_i$ for some $i \in [n]$, and consider the case when $\langle w_j, x_i \rangle = 0$. From Lemma 10, we have

$$\left| \frac{d}{dt}\psi_{rj} - (\psi_{ra} - \psi_{rj}\psi_{aj})\|x_a(w_j)\| \right| \leq 2nX_{\max}\max_i |f(x_i; W, v)|\,.$$

$\langle w_j, x_i \rangle = 0$ implies $\psi_{rj} = \left\langle \frac{x_i y_i}{\|x_i\|}, \frac{w_j}{\|w_j\|} \right\rangle = 0$, thus we have

$$\left| \frac{d}{dt}\psi_{rj}|_{\langle w_j, x_i \rangle = 0} - \psi_{ra}\|x_a(w_j)\| \right| \leq 2nX_{\max}\max_i |f(x_i; W, v)|\,.$$

Then whenever $w_j \notin \mathcal{S}_{\text{dead}}$, we have

$$\begin{aligned}
\frac{d}{dt}\psi_{rj}|_{\langle w_j, x_i \rangle = 0} &\geq \psi_{ra}\|x_a(w_j)\| - 2nX_{\max}\max_i |f(x_i; W, v)| \\
&\geq \mu\|x_a(w_j)\| - 2nX_{\max}\max_i |f(x_i; W, v)| &\text{(by Lemma 5)} \\
&\geq \mu^{3/2}X_{\min} - 2nX_{\max}\max_i |f(x_i; W, v)| &\text{(by Lemma 11)} \\
&\geq \mu^{3/2}X_{\min}/2 > 0\,. &\text{(by (29))}
\end{aligned}$$

This is precisely (7) in Section 3.3.

**Bound on activation transitions and duration:** Next we show that if at time $t_0 < T$, $w_j(t_0) \notin \mathcal{S}_+ \cup \mathcal{S}_{\text{dead}}$, and the activation pattern of $w_j$ is $\mathbb{1}_{\langle x_i, w_j(t_0) \rangle > 0}$, then $\mathbb{1}_{\langle x_i, w_j(t_0 + \Delta t) \rangle > 0} \neq \mathbb{1}_{\langle x_i, w_j(t_0) \rangle > 0}$, where $\Delta t = \frac{4}{\min\{\zeta, \xi\}\sqrt{\mu}X_{\min}n_a(w_j(t_0))}$ and $n_a(w_j(t_0))$ is defined in Lemma 11 as long as $t_0 + \Delta t < T$ as well. That is, during the alignment phase $[0, T]$, $w_j$ must change its activation pattern within $\Delta t$ time. There are two cases:

- The first case is when $w_j(t_0) \in \mathcal{S}_+^c \cap \mathcal{S}_-^c \cap \mathcal{S}_{\text{dead}}^c$. In this case, suppose that $\mathbb{1}_{\langle x_i, w_j(t_0 + \tau) \rangle > 0} = \mathbb{1}_{\langle x_i, w_j(t_0) \rangle > 0}, \forall 0 \leq \tau \leq \Delta t$, i.e. $w_j$ fixes its activation during $[t_0, t_0 + \Delta t]$, then we have $x_a(w_j(t_0 + \tau)) = x_a(w_j(t_0)), \forall 0 \leq \tau \leq \Delta t$. Let us pick $x_r = x_a(w_j(t_0))$, then Lemma 10 leads to

$$\left| \frac{d}{dt}\cos(w_j, x_a(w_j)) - \left(1 - \cos^2(w_j, x_a(w_j))\right)\|x_a(w_j)\| \right| \leq 2nX_{\max}\max_i |f(x_i; W, v)|\,.$$

Since $x_a(w_j)$ is fixed, we have $\forall t \in [t_0, t_0 + \Delta t]$,

$$\left| \frac{d}{dt}\cos(w_j, x_a(w_j(t_0))) - \left(1 - \cos^2(w_j, x_a(w_j(t_0)))\right)\|x_a(w_j(t_0))\| \right| \leq 2nX_{\max}\max_i |f(x_i; W, v)|\,,$$

$$\begin{aligned}
\frac{d}{dt}\cos(w_j, x_a(w_j(t_0))) &\geq \left(1 - \cos^2(w_j, x_a(w_j(t_0)))\right)\|x_a(w_j(t_0))\| \\
&\quad - 2nX_{\max}\max_i |f(x_i; W, v)| \\
&\geq \xi\|x_a(w_j(t_0))\| - 2nX_{\max}\max_i |f(x_i; W, v)| &\text{(by (27))} \\
&\geq \xi\sqrt{\mu}n_a(w_j(t_0))X_{\min} - 2nX_{\max}\max_i |f(x_i; W, v)| &\text{(by Lemma 11)} \\
&\geq \xi\sqrt{\mu}n_a(w_j(t_0))X_{\min}/2\,. &\text{(by (29))} \\
&\geq \min\{\xi, \zeta\}\sqrt{\mu}n_a(w_j(t_0))X_{\min}/2\,,
\end{aligned}$$

which implies that, by the Fundamental Theorem of Calculus,

$$\cos(w_j(t_0 + \Delta t), x_a(w_j(t_0)))$$

$$= \cos(w_j(t_0), x_a(w_j(t_0))) + \int_0^{\Delta t} \frac{d}{dt} \cos(w_j(t_0 + \tau), x_a(w_j(t_0)))d\tau$$

$$\geq \cos(w_j(t_0), x_a(w_j(t_0))) + \Delta t \cdot \min\{\xi, \zeta\}\sqrt{\mu}n_a(w_j(t_0))X_{\min}/2$$

$$= \cos(w_j(t_0), x_a(w_j(t_0))) + 2 \geq 1,$$

which leads to $\cos(w_j(t_0 + \Delta t), x_a(w_j(t_0))) = 1$. This would imply $w_j(t_0 + \Delta t) \in \mathcal{S}_+$ because $x_a(w_j(t_0)) \in \mathcal{S}_+$, which contradicts our original assumption that $w_j$ fixes the activation pattern. Therefore, $\exists 0 < \tau_0 \leq \Delta t$ such that $\mathbb{1}_{\langle x_i, w_j(t_0 + \tau_0))\rangle} \neq \mathbb{1}_{\langle x_i, w_j(t_0)\rangle > 0}$, due to the restriction on how $w_j$ can change its activation pattern, it cannot return to its previous activation pattern, then one must have $\mathbb{1}_{\langle x_i, w_j(t_0 + \Delta t))\rangle} \neq \mathbb{1}_{\langle x_i, w_j(t_0)\rangle > 0}$.

- The other case is when $w_j(t_0) \in \mathcal{S}_-$. For this case, we need first show that $w_j(t_0 + \tau) \notin \mathcal{S}_{x_-}^{\zeta}, \forall 0 \leq \tau \leq \Delta t$, or more generally, $\mathcal{S}_{x_-}^{\zeta}$ does not contain any $w_j$ in $\mathcal{V}_+$ during $[0, T]$. To see this, let us pick $x_r = x_-$, then Lemma 10 suggests that

$$\left| \frac{d}{dt}\psi_{rj} - (\psi_{ra} - \psi_{rj}\psi_{aj}) \|x_a(w_j)\| \right| \leq 2nX_{\max} \max_i |f(x_i; W, v)|.$$

Consider the case when $\cos(w_j, x_-) = \sqrt{1 - \zeta}$, i.e. $w_j$ is at the boundary of $\mathcal{S}_{x_-}^{\zeta}$. We know that in this case, $w_j \in \mathcal{S}_{x_-}^{\zeta} \subseteq \mathcal{S}_-$ thus $x_a(w_j) = -x_-$, and

$$\left| \frac{d}{dt}\cos(w_j, x_-) \right|_{\cos(w_j, x_-) = \sqrt{1-\zeta}} + \left(1 - \cos^2(w_j, x_-)\right) \|x_-\| \right| \leq 2nX_{\max} \max_i |f(x_i; W, v)|,$$

which is

$$\left| \frac{d}{dt}\cos(w_j, x_-) \right|_{\cos(w_j, x_-) = \sqrt{1-\zeta}} + \zeta\|x_-\| \right| \leq 2nX_{\max} \max_i |f(x_i; W, v)|$$

$$\Rightarrow \frac{d}{dt}\cos(w_j, x_-) \Big|_{\cos(w_j, x_-) = \sqrt{1-\zeta}}$$

$$\leq -\zeta\|x_-\| + 2nX_{\max} \max_i |f(x_i; W, v)|$$

$$\leq -\zeta\sqrt{\mu}X_{\min} + 2nX_{\max} \max_i |f(x_i; W, v)| \qquad \text{(by Lemma 11)}$$

$$\leq -\zeta\sqrt{\mu}X_{\min}/2 < 0. \qquad \text{(by (29))}$$

Therefore, during $[0, T]$, neuron $w_j$ in $\mathcal{V}_+$ cannot enter $\mathcal{S}_{x_-}^{\zeta}$ if at initialization, $w_j(0) \notin \mathcal{S}_{x_-}^{\zeta}$, which is guaranteed by (26).

With the argument above, we know that $w_j(t_0 + \tau) \notin \mathcal{S}_{x_-}^{\zeta}, \forall 0 \leq \tau \leq \Delta t$. Again we suppose that $w_j(t) \in \mathcal{S}_- - \mathcal{S}_{x_-}^{\zeta}, \forall t \in [t_0, t_0 + \Delta t]$, i.e., $w_j$ fixes its activation during $[t_0, t_0 + \Delta t]$. Let us pick $x_r = x_-$, then Lemma 10 suggests that

$$\left| \frac{d}{dt}\cos(w_j, x_-) + \left(1 - \cos^2(w_j, x_-)\right) \|x_-\| \right| \leq 2nX_{\max} \max_i |f(x_i; W, v)|,$$

which leads to $\forall t \in [t_0, t_0 + \Delta t]$,

$$\frac{d}{dt}\cos(w_j, x_-) \leq -\left(1 - \cos^2(w_j, x_-)\right) \|x_-\| + 2nX_{\max} \max_i |f(x_i; W, v)|$$

$$\leq -\zeta\|x_-\| + 2nX_{\max} \max_i |f(x_i; W, v)| \qquad (w_j \notin \mathcal{S}_{x_-}^{\zeta})$$

$$\leq -\zeta\sqrt{\mu}n_a(w_j(t_0))X_{\min} + 2nX_{\max} \max_i |f(x_i; W, v)| \qquad \text{(by Lemma 11)}$$

$$\leq -\zeta\sqrt{\mu}n_a(w_j(t_0))X_{\min}/2. \qquad \text{(by (29))}$$

$$\leq -\min\{\xi, \zeta\}\sqrt{\mu}n_a(w_j(t_0))X_{\min}/2,$$

Similarly, by FTC, we have

$$\cos(w_j(t_0 + \Delta t), x_-) \leq -1 .$$

This would imply $w_j(t_0 + \Delta t) \in \mathcal{S}_+$ because $-x_- = x_a(w_j(t_0)) \in \mathcal{S}_+$, which contradicts our original assumption that $w_j$ fixes its activation pattern. Therefore, one must have $\mathbb{1}_{\langle x_i, w_j(t_0 + \Delta t)) \rangle} \neq \mathbb{1}_{\langle x_i, w_j(t_0) \rangle > 0}$.

In summary, we have shown that, during $[0, T]$, a neuron in $\mathcal{V}_+$ can not keep a fixed activation pattern for a time longer than $\Delta t = \frac{4}{\min\{\zeta, \xi\}\sqrt{\mu}X_{\min}n_a}$, where $n_a$ is the number of data points that activate $w_j$ under the fixed activation pattern.

**Bound on total travel time until directional convergence** As we have discussed in Section 3.3 and also formally proved here, during alignment phase $[0, T]$, a neuron in $\mathcal{V}_+$ must change its activation pattern within $\Delta t = \frac{4}{\min\{\zeta, \xi\}\sqrt{\mu}X_{\min}n_a}$ time unless it is in either $\mathcal{S}_+$ or $\mathcal{S}_{\text{dead}}$. And the new activation it is transitioning into must contain no new activation on negative data points and must keep all existing activation on positive data points, together it shows that a neuron must reach either $\mathcal{S}_+$ or $\mathcal{S}_{\text{dead}}$ within a fixed amount of time, which is the remaining thing we need to formally show here.

For simplicity of the argument, we first assume $T = \infty$, i.e., the alignment phase lasts indefinitely, and we show that a neuron in $\mathcal{V}_+$ must reach $\mathcal{S}_+$ or $\mathcal{S}_{\text{dead}}$ before $t_1 = \frac{16 \log n}{\min\{\zeta, \xi\}\sqrt{\mu}X_{\min}}$. Lastly, such directional convergence can be achieved if $t_1 \leq T$, which is guaranteed by our choice of $\epsilon$ in (28).

- For a neuron in $\mathcal{V}_+$ that reaches $\mathcal{S}_{\text{dead}}$, the analysis is easy: It must start with no activation on positive data and then lose activation on negative data one by one until losing all of its activation. Therefore, it must reach $\mathcal{S}_{\text{dead}}$ before

$$\sum_{k=1}^{n_a(w_j(0))} \frac{4}{\min\{\zeta, \xi\}\sqrt{\mu}X_{\min}k} \leq \frac{4}{\min\{\zeta, \xi\}\sqrt{\mu}X_{\min}} \left( \sum_{k=1}^{n} \frac{1}{k} \right) \leq \frac{16 \log n}{\min\{\zeta, \xi\}\sqrt{\mu}X_{\min}} = t_1 .$$

- For a neuron in $\mathcal{V}_+$ that reaches $\mathcal{S}_+$, there is no difference conceptually, but it can switch its activation pattern in many ways before reaching $\mathcal{S}_+$, so it is not straightforward to see its travel time until $\mathcal{S}_+$ is upper bounded by $t_1$.

  To formally show the upper bound on the travel time, we need some definition of a path that keeps a record of the activation patterns of a neuron $w_j(t)$ before it reaches $\mathcal{S}_+$.

  Let $n_+ = |\mathcal{I}_+|$, $n_- = |\mathcal{I}_-|$ be the number of positive, negative data respectively, then we call $\mathcal{P}_{(k^{(0)}, k^{(1)}, \cdots, k^{(L)})}$ a *path* of length-$L$, if

  1. $\forall 0 \leq l \leq L$, we have $k^{(l)} = (k_+^{(l)}, k_-^{(l)}) \in \mathbb{N} \times \mathbb{N}$ with $0 \leq k_+^{(l)} \leq n_+, 0 \leq k_-^{(l)} \leq n_-$;
  2. For $k^{(l_1)}, k^{(l_2)}$ with $l_1 < l_2$, we have either $k_+^{(l_1)} > k_+^{(l_2)}$ or $k_-^{(l_1)} < k_-^{(l_2)}$;
  3. $k^{(L)} = (n_+, 0)$;
  4. $k^{(l)} \neq (0, 0), \forall 0 \leq l \leq L$.

  Given all our analysis on how a neuron $w_j(t)$ can switch its activation pattern in previous parts, we know that for any $w_j(t)$ that reaches $\mathcal{S}_+$, there is an associated $\mathcal{P}_{(k^{(0)}, k^{(1)}, \cdots, k^{(L)})}$ that keeps an ordered record of encountered values of

  $$\left( |\{i \in \mathcal{I}_+ : \langle x_i, w_j(t) \rangle > 0\}|, \ |\{i \in \mathcal{I}_- : \langle x_i, w_j(t) \rangle > 0\}| \right) ,$$

  before $w_j$ reaches $\mathcal{S}_+$. That is, a neuron $w_j$ starts with some activation pattern that activates $k_+(0)$ positive data and $k_-(0)$ negative data, then switch its activation pattern (by either losing negative data or gaining positive data) to one that activates $k_+(1)$ positive data and $k_-(1)$ negative data. By keep doing so, it reaches $\mathcal{S}_+$ that activates $k_+(L) = n_+$ positive data and $k_-(L) = 0$ negative data. Please see Figure 14 for an illustration of a path.

  Given a path $\mathcal{P}_{(k^{(0)}, k^{(1)}, \cdots, k^{(L)})}$ of neuron $w_j$, we define the *travel time* of this path as

  $$T(\mathcal{P}_{(k^{(0)}, k^{(1)}, \cdots, k^{(L)})}) = \sum_{l=0}^{L-1} \frac{4}{\min\{\zeta, \xi\}\sqrt{\mu}X_{\min}(k_+^{(l)} + k_-^{(l)})} ,$$

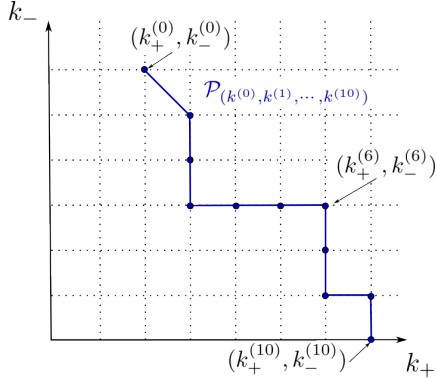
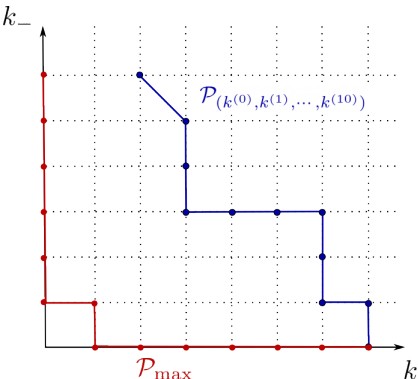

Figure 14: Illustration of a path of length-10. Each dot on the grid represents one $k^{(l)}$.

Figure 15: Illustration of a path and the maximal path

which is exactly the traveling time from $k^{(0)}$ to $k^{(L)}$ if one spends $\frac{4}{\min\{\zeta,\xi\}\sqrt{\mu}X_{\min}(k_+^{(l)}+k_-^{(l)})}$ on the edge between $k^{(l)}$ and $k^{(l+1)}$.

Our analysis shows that if $w_j$ reaches $\mathcal{S}_+$, then

$$\inf\{t : w_j(t) \in \mathcal{S}_+\} \leq T(\mathcal{P}_{(k^{(0)},k^{(1)},\cdots,k^{(L)})})\,.$$

Now we define the maximal path $\mathcal{P}_{\max}$ as a path that has the maximum length $n = n_+ + n_-$, which is uniquely determined by the following trajectory of $k^{(l)}$

$$(0,n_-), (0,n_--1), (0,n_--2), \cdots, (0,1), (1,1), (1,0), \cdots, (n_+-1,0), (n_+,0)\,.$$

Please see Figure 15 for an illustration.

The traveling time for $\mathcal{P}_{\max}$ is

$$
\begin{aligned}
T(\mathcal{P}_{\max}) =\ & \frac{4}{\min\{\zeta,\xi\}\sqrt{\mu}X_{\min}} \left( \sum_{k=1}^{n_-} \frac{1}{k} + \frac{1}{2} + \sum_{k=1}^{n_+-1} \frac{1}{k} \right) \\
\leq\ & \frac{4}{\min\{\zeta,\xi\}\sqrt{\mu}X_{\min}} \left( 2\sum_{k=1}^{n} \frac{1}{k} + \frac{1}{2} \right) \\
\leq\ & \frac{16\log n}{\min\{\zeta,\xi\}\sqrt{\mu}X_{\min}} = t_1\,.
\end{aligned}
$$

The proof is complete by the fact that any path satisfies

$$T(\mathcal{P}_{(k^{(0)},k^{(1)},\cdots,k^{(L)})}) \leq T(\mathcal{P}_{\max})\,.$$

This is because there is a one-to-one correspondence between the edges $(k^{(l)}, k^{(l+1)})$ in $\mathcal{P}_{(k^{(0)},k^{(1)},\cdots,k^{(L)})}$ and a subset of edges in $\mathcal{P}_{\max}$, and the travel time from of edge $(k^{(l)}, k^{(l+1)})$ is shorter than the corresponding edge in $\mathcal{P}_{\max}$. Formally stating such correspondence is tedious and a visual illustration in Figure 16 and 17 is more effective (Putting all correspondence makes a clustered plot thus we split them into two figures):

Therefore, if $w_j$ reaches $\mathcal{S}_+$, then it reaches $\mathcal{S}_+$ within $t_1$:

$$\inf\{t : w_j(t) \in \mathcal{S}_+\} \leq T(\mathcal{P}_{(k^{(0)},k^{(1)},\cdots,k^{(L)})}) \leq T(\mathcal{P}_{\max}) \leq t_1\,.$$

So far we have shown when the alignment phase lasts long enough, i.e., $T$ large enough, the directional convergence is achieved by $t_1$. We simply pick $\epsilon$ such that

$$T = \frac{1}{4nX_{\max}} \log \frac{1}{\sqrt{h}\epsilon} \geq t_1 = \frac{16\log n}{\min\{\zeta,\xi\}\sqrt{\mu}X_{\min}}\,,$$

and (28) suffices. $\qquad\square$

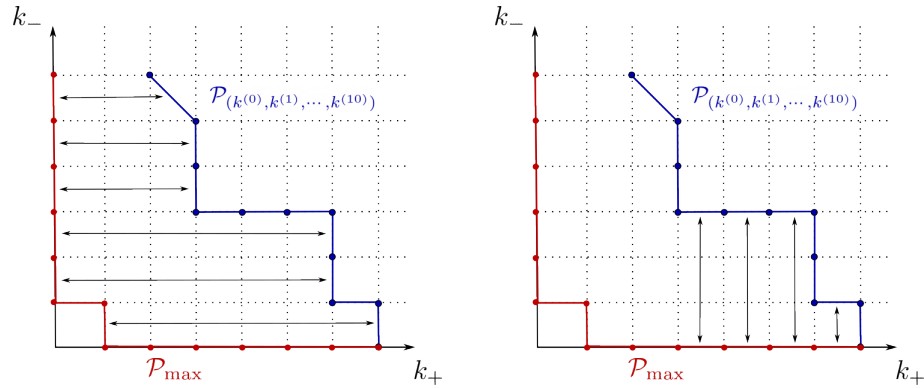

Figure 16: Correspondence between edges in $\mathcal{P}_{(k^{(0)},k^{(1)},\cdots,k^{(L)})}$ and $\mathcal{P}_{\max}$. (Part 1)

Figure 17: Correspondence between edges in $\mathcal{P}_{(k^{(0)},k^{(1)},\cdots,k^{(L)})}$ and $\mathcal{P}_{\max}$. (Part 2)

## D  PROOF FOR THEOREM 1: FINAL CONVERGENCE

Since we have proved the first part of Theorem 1 in Section C, we will use it as a fact, then prove the remaining part of Theorem 1.

### D.1  AUXILIARY LEMMAS

First, we show that $\mathcal{S}_+, \mathcal{S}_-, \mathcal{S}_{\text{dead}}$ are trapping regions.

**Lemma 12.** *Consider any solution to the gradient flow dynamic* (2), *we have the following:*

- *If at some time $t_1 \geq 0$, we have $w_j(t_1) \in \mathcal{S}_{dead}$, then $w_j(t_1 + \tau) \in \mathcal{S}_{dead}$, $\forall \tau \geq 0$;*

- *If at some time $t_1 \geq 0$, we have $w_j(t_1) \in \mathcal{S}_+$ for some $j \in \mathcal{V}_+$, then $w_j(t_1 + \tau) \in \mathcal{S}_+$, $\forall \tau \geq 0$;*

- *If at some time $t_1 \geq 0$, we have $w_j(t_1) \in \mathcal{S}_-$ for some $j \in \mathcal{V}_-$, then $w_j(t_1 + \tau) \in \mathcal{S}_-$, $\forall \tau \geq 0$;*

*Proof.* The first statement is simple, if $w_j \in \mathcal{S}_{\text{dead}}$, then one have $\dot{w}_j = 0$, thus $w_j$ remains in $\mathcal{S}_{\text{dead}}$. For the second statement, we have, since $j \in \mathcal{V}_+$,

$$\frac{d}{dt}w_j = -\sum_{i=1}^{n} \mathbb{1}_{\langle x_i, w_j \rangle > 0} \nabla_{\hat{y}} \ell(y_i, f(x_i; W, v)) x_i \|w_j\|.$$

When $\ell$ is the exponential loss, by the Fundamental Theorem of Calculus, one writes, $\forall \tau \geq 0$,

$$
\begin{aligned}
w_j(t_1 + \tau) &= w_j(t_1) + \int_0^\tau \frac{d}{dt} w_j d\tau \\
&= w_j(t_1) + \int_0^\tau -\sum_{i=1}^{n} \mathbb{1}_{\langle x_i, w_j \rangle > 0} \nabla_{\hat{y}} \ell(y_i, f(x_i; W, v)) x_i \|w_j\| d\tau \\
&= w_j(t_1) + \int_0^\tau \sum_{i=1}^{n} \mathbb{1}_{\langle x_i, w_j \rangle > 0} y_i \exp(-y_i f(x_i; W, v)) x_i \|w_j\| d\tau \\
&= w_j(t_1) + \underbrace{\sum_{i \in \mathcal{I}_+} \left( \int_0^\tau \exp(-y_i f(x_i; W, v)) \|w_j\| d\tau \right) x_i}_{:=\tilde{x}_+} .
\end{aligned}
$$

Here $w_j(t_1) \in \mathcal{S}_+$ by our assumption, $\tilde{x}_+ \in K \subseteq \mathcal{S}_+$ because $\tilde{x}_+$ is a conical combination of $x_i, i \in \mathcal{I}_+$. Since $\mathcal{S}_+$ is a convex cone, we have $w_j(t_1 + \tau) \in \mathcal{S}_+$ as well.

When $\ell$ is the logistic loss, we have, similarly,

$$w_j(t_1 + \tau) = w_j(t_1) + \int_0^\tau \sum_{i=1}^n \mathbb{1}_{\langle x_i, w_j \rangle > 0} y_i \frac{2\exp(-y_i f(x_i; W, v))}{1 + \exp(-y_i f(x_i; W, v))} x_i \|w_j\| d\tau$$

$$= w_j(t_1) + \underbrace{\sum_{i \in \mathcal{I}_+} \left( \int_0^\tau \frac{2\exp(-y_i f(x_i; W, v))}{1 + \exp(-y_i f(x_i; W, v))} \|w_j\| d\tau \right) x_i \in \mathcal{S}_+}_{:= \tilde{x}_+}.$$

The proof of the third statement is almost identical (we only show the case of exponential loss here): when $j \in \mathcal{V}_-$, we have

$$\frac{d}{dt} w_j = \sum_{i=1}^n \mathbb{1}_{\langle x_i, w_j \rangle > 0} \nabla_{\hat{y}} \ell(y_i, f(x_i; W, v)) x_i \|w_j\|,$$

and

$$w_j(t_1 + \tau) = w_j(t_1) + \underbrace{\sum_{i \in \mathcal{I}_-} \left( \int_0^\tau \exp(-y_i f(x_i; W, v)) \|w_j\| d\tau \right) x_i}_{:= \tilde{x}_-}.$$

Again, here $w_j(t_1) \in \mathcal{S}_-$ by our assumption, $\tilde{x}_- \in -K \subseteq \mathcal{S}_-$ because $\tilde{x}_-$ is a conical combination of $x_i, i \in \mathcal{I}_-$. Since $\mathcal{S}_-$ is a convex cone, we have $w_j(t_1 + \tau) \in \mathcal{S}_+$ as well. $\qquad \square$

Then the following Lemma provides a lower bound on neuron norms upon $t_1$.

**Lemma 13.** *Consider any solution to the gradient flow dynamic* (2) *starting from initialization* (3). *Let $t_1$ be the time when directional convergence is achieved, as defined in Theorem 1, and we define $\tilde{\mathcal{V}}_+ : \{j : w_j(t_1) \in \mathcal{S}_+\}$ and $\tilde{\mathcal{V}}_- : \{j : w_j(t_1) \in \mathcal{S}_-\}$. If both $\tilde{\mathcal{V}}_+$ and $\tilde{\mathcal{V}}_-$ are non-empty, we have*

$$\sum_{j \in \tilde{\mathcal{V}}_+} \|w_j(t_1)\|^2 \geq \exp(-4nX_{\max} t_1) \sum_{j \in \tilde{\mathcal{V}}_+} \|w_j(0)\|^2,$$

$$\sum_{j \in \tilde{\mathcal{V}}_-} \|w_j(t_1)\|^2 \geq \exp(-4nX_{\max} t_1) \sum_{j \in \tilde{\mathcal{V}}_-} \|w_j(0)\|^2,$$

*Proof.* We have shown that

$$\frac{d}{dt} \|w_j\|^2 = -2 \sum_{i=1}^n \mathbb{1}_{\langle x_i, w_j \rangle > 0} \nabla_{\hat{y}} \ell(y_i, f(x_i; W, v)) \langle x_i, w_j \rangle \operatorname{sign}(v_j(0)) \|w_j\|.$$

Then before $t_1$, we have $\forall j \in [h]$

$$\frac{d}{dt} \|w_j\|^2 = -2 \sum_{i=1}^n \mathbb{1}_{\langle x_i, w_j \rangle > 0} \nabla_{\hat{y}} \ell(y_i, f(x_i; W, v)) \langle x_i, w_j \rangle \operatorname{sign}(v_j(0)) \|w_j\|$$

$$\geq -2 \sum_{i=1}^n (|y_i| + 2 \max_i |f(x_i; W, v)|) \|x_i\| \|w_j\|^2$$

$$\geq -4 \sum_{i=1}^n \|x_i\| \|w_j\|^2 \geq -4nX_{\max} \|w_j\|^2,$$

where the second last inequality is because $\max_i |f(x_i; W, v)| \leq \frac{1}{2}$ before $t_1$. Summing over $j \in \tilde{\mathcal{V}}_+$, we have

$$\frac{d}{dt} \sum_{j \in \tilde{\mathcal{V}}_+} \|w_j\|^2 \geq -4nX_{\max} \sum_{j \in \tilde{\mathcal{V}}_+} \|w_j\|^2.$$

Therefore, we have the following bound:

$$\sum_{j \in \tilde{\mathcal{V}}_+} \|w_j(t_1)\|^2 \geq \exp(-4nX_{\max} t_1) \sum_{j \in \tilde{\mathcal{V}}_+} \|w_j(0)\|^2.$$

$\qquad \square$

Moreover, after $t_1$, the neuron norms are non-decreasing, as suggested by

**Lemma 14.** *Consider any solution to the gradient flow dynamic* (2) *starting from initialization* (3). *Let $t_1$ be the time when directional convergence is achieved, as defined in Theorem 1, and we define $\tilde{\mathcal{V}}_+ : \{j : w_j(t_1) \in \mathcal{S}_+\}$ and $\tilde{\mathcal{V}}_- : \{j : w_j(t_1) \in \mathcal{S}_-\}$. If both $\tilde{\mathcal{V}}_+$ and $\tilde{\mathcal{V}}_-$ are non-empty, we have $\forall \tau \geq 0$ and $t_2 \geq t_1$,*

$$\sum_{j \in \tilde{\mathcal{V}}_+} \|w_j(t_2 + \tau)\|^2 \geq \sum_{j \in \tilde{\mathcal{V}}_+} \|w_j(t_2)\|, \qquad \sum_{j \in \tilde{\mathcal{V}}_-} \|w_j(t_2 + \tau)\|^2 \geq \sum_{j \in \tilde{\mathcal{V}}_-} \|w_j(t_2)\| \qquad (30)$$

*Proof.* It suffices to show that after $t_1$, the following derivatives:

$$\frac{d}{dt} \sum_{j \in \tilde{\mathcal{V}}_+} \|w_j(t)\|^2, \quad \frac{d}{dt} \sum_{j \in \tilde{\mathcal{V}}_-} \|w_j(t)\|^2 \,,$$

are non-negative.

For $j \in \tilde{\mathcal{V}}_+$, $w_j$ stays in $\mathcal{S}_+$ by Lemma 12, and we have

$$
\begin{aligned}
\frac{d}{dt} \|w_j\|^2 &= -2 \sum_{i \in \mathcal{I}_+} \nabla_{\hat{y}} \ell(y_i, f(x_i; W, v)) \langle x_i, w_j \rangle \|w_j\| \,. \\
&= \begin{cases} 2 \sum_{i \in \mathcal{I}_+} y_i \exp(-y_i f(x_i; W, v)) \langle x_i, w_j \rangle \|w_j\| & (\ell \text{ is exponential}) \\ 2 \sum_{i \in \mathcal{I}_+} y_i \frac{2 \exp(-y_i f(x_i; W, v))}{1 + \exp(-y_i f(x_i; W, v))} \langle x_i, w_j \rangle \|w_j\| & (\ell \text{ is logistic}) \end{cases} \\
&\geq 0 \,.
\end{aligned}
$$

Summing over $j \in \tilde{\mathcal{V}}_+$, we have $\frac{d}{dt} \sum_{j \in \tilde{\mathcal{V}}_+} \|w_j(t)\|^2 \geq 0$. Similarly one has $\frac{d}{dt} \sum_{j \in \tilde{\mathcal{V}}_-} \|w_j(t)\|^2 \geq 0$. $\square$

Finally, the following lemma is used for deriving the final convergence.

**Lemma 15.** *Consider the following loss function*

$$\mathcal{L}_{lin}(W, v) = \sum_{i=1}^n \ell\left(y_i, v^\top W^\top x_i\right) \,,$$

*if $\{x_i, y_i\}, i \in [n]$ are linearly separable, i.e., $\exists \gamma > 0$ and $z \in \mathbb{S}^{D-1}$ such that $y_i \langle z, x_i \rangle \geq \gamma, \forall i \in [n]$, then under the gradient flow on $\mathcal{L}_{lin}(W, v)$, whenever $y_i v^\top W^\top x_i \geq 0, \forall i$, we have*

$$\dot{\mathcal{L}}_{lin} \leq -\frac{1}{4} \|v\|^2 \mathcal{L}^2 \gamma^2 \,. \qquad (31)$$

*Proof.* For $\ell$ being exponential loss, we have:

$$
\begin{aligned}
\dot{\mathcal{L}} &= -\|\nabla_W \mathcal{L}\|_F^2 - \|\nabla_v \mathcal{L}\|_F^2 \leq -\|\nabla_W \mathcal{L}\|_F^2 \\
&= -\left\| \sum_{i=1}^n y_i \ell(y_i, v^\top W^\top x_i) x_i v^\top \right\|_F^2 \\
&= -\|v\|^2 \left\| \sum_{i=1}^n y_i \ell(y_i, v^\top W^\top x_i) x_i \right\|^2 \\
&\leq -\|v\|^2 \left| \left\langle z, \sum_{i=1}^n y_i \ell(y_i, v^\top W^\top x_i) x_i \right\rangle \right|^2 \\
&\leq -\|v\|^2 \left| \sum_{i=1}^n \ell(y_i, v^\top W^\top x_i) \gamma \right|^2 \\
&\leq -\|v\|^2 \mathcal{L}^2 \gamma^2 \leq -\frac{1}{4} \|v\|^2 \mathcal{L}^2 \gamma^2 \,.
\end{aligned}
$$

For $\ell$ being logistic loss, we have:

$$\dot{\mathcal{L}} = -\|\nabla_W \mathcal{L}\|_F^2 - \|\nabla_v \mathcal{L}\|_F^2 \leq -\|\nabla_W \mathcal{L}\|_F^2$$

$$= -\left\| \sum_{i=1}^n y_i \frac{2\exp(-y_i v^\top W^\top x_i)}{1+\exp(-y_i v^\top W^\top x_i)} x_i v^\top \right\|_F^2$$

$$= -\|v\|^2 \left\| \sum_{i=1}^n y_i \frac{2\exp(-y_i v^\top W^\top x_i)}{1+\exp(-y_i v^\top W^\top x_i)} x_i \right\|^2$$

$$\leq -\|v\|^2 \left| \left\langle z, \sum_{i=1}^n y_i \frac{2\exp(-y_i v^\top W^\top x_i)}{1+\exp(-y_i v^\top W^\top x_i)} x_i \right\rangle \right|^2$$

$$\leq -\|v\|^2 \left| \sum_{i=1}^n \frac{2\exp(-y_i v^\top W^\top x_i)}{1+\exp(-y_i v^\top W^\top x_i)} \gamma \right|^2$$

$$= -\|v\|^2 \gamma^2 \left| \sum_{i=1}^n \frac{2\exp(-y_i v^\top W^\top x_i)}{1+\exp(-y_i v^\top W^\top x_i)} \right|^2$$

$$\leq -\|v\|^2 \gamma^2 \left| \sum_{i=1}^n \log(1+\exp(-y_i v^\top W^\top x_i)) \right|^2$$

$$= -\frac{1}{4}\|v\|^2 \mathcal{L}^2 \gamma^2,$$

where the last inequality uses the fact that $2\frac{z}{1+z} \geq \log(1+z)$ when $z \in [0,1]$. $\qquad\square$

### D.2 PROOF OF FINAL CONVERGENCE

*Proof of Theorem 1: Second Part.* By Lemma 12, we know that after $t_1$, neurons in $\mathcal{S}_+$ ($\mathcal{S}_-$) stays in $\mathcal{S}_+$ ($\mathcal{S}_-$). Thus the loss can be decomposed as

$$\mathcal{L} = \underbrace{\sum_{i \in \mathcal{I}_+} \ell\left(y_i, \sum_{j \in \tilde{\mathcal{V}}_+} v_j \langle w_j, x_i \rangle \right)}_{\mathcal{L}_+} + \underbrace{\sum_{i \in \mathcal{I}_-} \ell\left(y_i, \sum_{j \in \tilde{\mathcal{V}}_-} v_j \langle w_j, x_i \rangle \right)}_{\mathcal{L}_-}, \tag{32}$$

where $\tilde{\mathcal{V}}_+ : \{j : w_j(t_1) \in \mathcal{S}_+\}$ and $\tilde{\mathcal{V}}_- : \{j : w_j(t_1) \in \mathcal{S}_-\}$. Therefore, the training after $t_1$ is decoupled into 1) using neurons in $\tilde{\mathcal{V}}_+$ to fit positive data in $\mathcal{I}_+$ and 2) using neurons in $\tilde{\mathcal{V}}_-$ to fit positive data in $\mathcal{I}_-$.

We define $f_+(x_i; W, v) = \sum_{j \in \tilde{\mathcal{V}}_+} v_j \langle w_j, x_i \rangle$ and let $t_2^+ = \inf\{t : \max_{i \in \mathcal{I}_+} |f_+(x_i; W, v)| > \frac{1}{4}\}$. Similarly, we also define $f_-(x_i; W, v) = \sum_{j \in \tilde{\mathcal{V}}_+} v_j \langle w_j, x_i \rangle$ and let $t_2^- = \inf\{t : \max_{i \in \mathcal{I}_-} |f_-(x_i; W, v)| > \frac{1}{4}\}$. Then $t_1 \leq \min\{t_2^+, t_2^-\}$, by Lemma 3.

$\mathcal{O}(1/t)$ **convergence after** $t_2$: We first show that when both $t_2^+, t_2^-$ are finite, then it implies $\mathcal{O}(1/t)$ convergence on the loss. Then we show that they are indeed finite and $t_2 := \max\{t_2^+, t_2^-\} = \mathcal{O}(\frac{1}{n}\log\frac{1}{\epsilon})$.

At $t_2 = \max\{t_2^+, t_2^-\}$, by definition, $\exists i_+ \in \mathcal{I}_+$ such that

$$\frac{1}{4} \leq f_+(x_{i_+}; W, v) \leq \sum_{j \in \tilde{\mathcal{V}}_+} v_j \langle w_j, x_{i_+} \rangle \leq \sum_{j \in \tilde{\mathcal{V}}_+} \|w_j\|^2 \|x_{i_+}\|, \tag{33}$$

which implies, by Lemma 14, $\forall t \geq t_2$

$$\sum_{j \in \tilde{\mathcal{V}}_+} \|w_j(t)\|^2 \geq \sum_{j \in \tilde{\mathcal{V}}_+} \|w_j(t_2)\|^2 \geq \frac{1}{4\|x_{i_+}\|} \geq \frac{1}{4X_{\max}}. \tag{34}$$

Similarly, we have $\forall t \geq t_2$,

$$\sum_{j \in \tilde{\mathcal{V}}_-} \|w_j(t)\|^2 \geq \frac{1}{4X_{\max}} \,. \tag{35}$$

Under the gradient flow dynamics (2), we apply Lemma 15 to the decomposed loss (32)

$$4\dot{\mathcal{L}} \leq -\left(\sum_{j \in \tilde{\mathcal{V}}_+} v_j^2\right) \cdot \mathcal{L}_+^2 \cdot (\mu X_{\min})^2 - \left(\sum_{j \in \tilde{\mathcal{V}}_+} v_j^2\right) \cdot \mathcal{L}_-^2 \cdot (\mu X_{\min})^2 \,.$$

Here, we can pick the same $\gamma = \mu X_{\min}$ for both $\mathcal{L}_+$ and $\mathcal{L}_-$ because $\{x_i, y_i\}, i \in \mathcal{I}_+$ is linearly separable with $z = \frac{y_1 x_1}{\|x_1\|}$: $\langle z, x_i y_i \rangle \geq \mu\|x_i\| \geq \mu X_{\min}$ by Assumption 1. And similarly, $\{x_i, y_i\}, i \in \mathcal{I}_-$ is linearly separable with $\langle z, x_i y_i \rangle \geq \mu\|x_i\| \geq \mu X_{\min}$. Replace $v_i^2$ by $\|w_j\|^2$ from balancedness, together with (34)(35), we have

$$4\dot{\mathcal{L}} \leq -\left(\sum_{j \in \tilde{\mathcal{V}}_+} \|w_j\|^2\right) \cdot \mathcal{L}_+^2 \cdot (\mu X_{\min})^2 - \left(\sum_{j \in \tilde{\mathcal{V}}_+} \|w_j\|^2\right) \cdot \mathcal{L}_-^2 \cdot (\mu X_{\min})^2$$

$$\leq -\frac{(\mu X_{\min})^2}{4X_{\max}}(\mathcal{L}_+^2 + \mathcal{L}_-^2) \leq -\frac{(\mu X_{\min})^2}{8X_{\max}}(\mathcal{L}_+ + \mathcal{L}_-)^2 = -\frac{(\mu X_{\min})^2}{8X_{\max}}\mathcal{L}^2 \,,$$

which is

$$\frac{1}{\mathcal{L}^2}\dot{\mathcal{L}} \leq -\frac{(\mu X_{\min})^2}{32X_{\max}} \,.$$

Integrating both side from $t_2$ to any $t \geq t_2$, we have

$$\left.\frac{1}{\mathcal{L}}\right|_{t_2}^{\top} \leq -\frac{(\mu X_{\min})^2}{32X_{\max}}(t - t_2) \,,$$

which leads to

$$\mathcal{L}(t) \leq \frac{\mathcal{L}(t_2)}{\mathcal{L}(t_2)\alpha(t - t_2) + 1} \,, \text{ where } \alpha = \frac{(\mu X_{\min})^2}{32X_{\max}} \,.$$

**Showing $t_2 = \mathcal{O}(\frac{1}{n}\log\frac{1}{\epsilon})$:** The remaining thing is to show $t_2$ is $\mathcal{O}(\frac{1}{n}\log\frac{1}{\epsilon})$.

Since after $t_1$, the gradient dynamics are fully decoupled into two gradient flow dynamics (on $\mathcal{L}_+$ and on $\mathcal{L}_-$), it suffices to show $t_2^+ = \mathcal{O}(\frac{1}{n}\log\frac{1}{\epsilon})$ and $t_2^- = \mathcal{O}(\frac{1}{n}\log\frac{1}{\epsilon})$ separately, then combine them to show $t_2 = \max\{t_2^+, t_2^-\} = \mathcal{O}(\frac{1}{n}\log\frac{1}{\epsilon})$. The proof is almost identical for $\mathcal{L}_+$ and $\mathcal{L}_-$, thus we only prove $t_2^+ = \mathcal{O}(\frac{1}{n}\log\frac{1}{\epsilon})$ here.

Suppose

$$t_2 \geq t_1 + \frac{6}{\sqrt{\mu}n_+X_{\min}} + \frac{4}{\sqrt{\mu}n_+X_{\min}}\left(\log\frac{2}{\epsilon^2\sqrt{\mu}X_{\min}W_{\min}^2} + 4nX_{\max}t_1\right) \,, \tag{36}$$

where $n_+ = |\mathcal{I}_+|$. It takes two steps to show a contradiction: First, we show that for some $t_a \geq 0$, a refined alignment $\cos(w_j(t_1 + t_a), x_+) \geq \frac{1}{4}, \forall j \in \tilde{\mathcal{V}}_+$ is achieved, and such refined alignment is maintained until at least $t_2^+$: $\cos(w_j(t), x_+) \geq \frac{1}{4}, \forall j \in \tilde{\mathcal{V}}_+$ for all $t_1 + t_a \leq t \leq t_2^+$. Then, keeping this refined alignment leads to a contradiction.

- For $j \in \tilde{\mathcal{V}}_+$, we have

$$\frac{d}{dt}\frac{w_j}{\|w_j\|} = \left(I - \frac{w_j w_j^\top}{\|w_j\|^2}\right)\underbrace{\left(\sum_{i \in \mathcal{I}_+} -\nabla_{\hat{y}}\ell(y_i, f_+(x_i; W, v))x_i\right)}_{:=\tilde{x}_a} \,.$$

Then

$$\frac{d}{dt}\cos(x_+, w_j) = (\cos(x_+, \tilde{x}_a) - \cos(x_+, w_j)\cos(\tilde{x}_a, w_j))\|\tilde{x}_a\|$$

$$\geq (\cos(x_+, \tilde{x}_a) - \cos(x_+, w_j))\|\tilde{x}_a\| \,.$$

We can show that $\cos(x_+, \tilde{x}_a) \geq \frac{1}{3}$ and $\|\tilde{x}_a\| \geq \sqrt{\mu}n_+ X_{\min}/2$ when $t_1 \leq t \leq t_2^+$ (we defer the proof to the end as it breaks the flow), thus within $[t_1, t_2^+]$, we have

$$\frac{d}{dt} \cos(x_+, w_j) \geq \left( \frac{1}{3} - \cos(x_+, w_j) \right) \sqrt{\mu}n_+ X_{\min}/2 \,. \tag{37}$$

We use (37) in two ways: First, since

$$\frac{d}{dt} \cos(x_+, w_j) \Big|_{\cos(x_+, w_j) = \frac{1}{4}} \geq \frac{\sqrt{\mu}n_+ X_{\min}}{24} > 0 \,,$$

$\cos(x_+, w_j) \geq \frac{1}{4}$ is a trapping region for $w_j$ during $[t_1, t_2^+]$. Define $t_a := \inf\{t \geq t_1 : \min_{j \in \tilde{\mathcal{V}}_+} \cos(x_+, w_j(t)) \geq \frac{1}{4}\}$, then clearly, if $t_a \leq t_2^+$, then $\cos(w_j(t), x_+) \geq \frac{1}{4}, \forall j \in \tilde{\mathcal{V}}_+$ for all $t_1 + t_a \leq t \leq t_2^+$.

Now we use (37) again to show that $t_a \leq t_1 + \frac{6}{\sqrt{\mu}n_+ X_{\min}}$: Suppose that $t_a \geq t_1 + \frac{6}{\sqrt{\mu}n_+ X_{\min}}$, then $\exists j^*$ such that $\cos(x_+, w_{j^*}(t)) < \frac{1}{4}, \forall t \in [t_1, t_1 + \frac{6}{\sqrt{\mu}n_+ X_{\min}}]$, and we have

$$\frac{d}{dt} \cos(x_+, w_{j^*}) \geq \left( \frac{1}{3} - \cos(x_+, w_j) \right) \sqrt{\mu}n_+ X_{\min}/2 \geq \frac{\sqrt{\mu}n_+ X_{\min}}{24} \,. \tag{38}$$

This shows

$$\cos(x_+, w_{j^*}(t_1 + 1)) \geq \cos(x_+, w_{j^*}(t_1)) + \frac{1}{4} \geq \frac{1}{4} \,,$$

which contradicts that $\cos(x_+, w_{j^*}(t)) < \frac{1}{4}$. Hence we know $t_a \leq t_1 + \frac{6}{\sqrt{\mu}n_+ X_{\min}}$.

In summary, we have $\cos(w_j(t), x_+) \geq \frac{1}{4}, \forall j \in \tilde{\mathcal{V}}_+$ for all $t_1 + \frac{6}{\sqrt{\mu}n_+ X_{\min}} \leq t \leq t_2^+$.

- Now we check the dynamics of $\sum_{j \in \tilde{\mathcal{V}}_+} \|w_j(t)\|^2$ during $t_1 + \frac{6}{\sqrt{\mu}n_+ X_{\min}} \leq t \leq t_2^+$. For simplicity, we denote $t_1 + \frac{6}{\sqrt{\mu}n_+ X_{\min}} := t_1'$.

For $j \in \tilde{\mathcal{V}}_+$, we have, for $t_1' \leq t \leq t_2^+$,

$$\begin{aligned}
\frac{d}{dt}\|w_j\|^2 &= 2 \sum_{i \in \mathcal{I}_+} -\nabla_{\hat{y}} \ell(y_i, f(x_i; W, v)) \langle x_i, w_j \rangle \|w_j\| \\
&\geq \sum_{i \in \mathcal{I}_+} \langle x_i, w_j \rangle \|w_j\| && \text{(by (40))} \\
&= \langle x_+, w_j \rangle \|w_j\| \\
&= \|x_+\| \|w_j\|^2 \cos(x_+, w_j) \\
&\geq \frac{1}{4} \|x_+\| \|w_j\|^2 && \text{(Since } t \geq t_1') \\
&\geq \frac{\sqrt{\mu}n_+ X_{\min}}{4} \|w_j\|^2 \,, && \text{(by Lemma 11)}
\end{aligned}$$

which leads to (summing over $j \in \tilde{\mathcal{V}}_+$)

$$\frac{d}{dt} \sum_{j \in \tilde{\mathcal{V}}_+} \|w_j\|^2 \geq \frac{\sqrt{\mu}n_+ X_{\min}}{4} \sum_{j \in \tilde{\mathcal{V}}_+} \|w_j\|^2 \,.$$

By Gronwall's inequality, we have

$$
\sum_{j \in \tilde{\mathcal{V}}_+} \|w_j(t_2^+)\|^2
$$

$$
\geq \exp\left(\frac{\sqrt{\mu} n_+ X_{\min}}{4}(t_2^+ - t_1')\right) \sum_{j \in \tilde{\mathcal{V}}_+} \|w_j(t_1')\|^2
$$

$$
\geq \exp\left(\frac{\sqrt{\mu} n_+ X_{\min}}{4}(t_2^+ - t_1')\right) \sum_{j \in \tilde{\mathcal{V}}_+} \|w_j(t_1)\|^2 \qquad \text{(By Lemma 14)}
$$

$$
\geq \exp\left(\frac{\sqrt{\mu} n_+ X_{\min}}{4}(t_2^+ - t_1')\right) \exp\left(-4n X_{\max} t_1\right) \sum_{j \in \tilde{\mathcal{V}}_+} \|w_j(0)\|^2 \qquad \text{(By Lemma 13)}
$$

$$
\geq \exp\left(\frac{\sqrt{\mu} n_+ X_{\min}}{4}(t_2^+ - t_1')\right) \exp\left(-4n X_{\max} t_1\right) \epsilon^2 W_{\min}^2 \geq \frac{2}{\sqrt{\mu} X_{\min}} . \qquad \text{(by (36))}
$$

However, at $t_2^+$, we have

$$
\frac{1}{4} \geq \frac{1}{n_+} \sum_{i \in \mathcal{I}_+} f_+(x_i; W, v) = \frac{1}{n_+} \sum_{i \in \mathcal{I}_+} \sum_{j \in \tilde{\mathcal{V}}_+} v_j \langle w_j, x_i \rangle
$$

$$
= \frac{1}{n_+} \sum_{j \in \tilde{\mathcal{V}}_+} v_j \langle w_j, x_+ \rangle *
$$

$$
= \frac{1}{n_+} \sum_{j \in \tilde{\mathcal{V}}_+} \|w_j\|^2 \cos(w_j, x_+) \|x_+\|
$$

$$
\geq \frac{1}{4n_+} \sum_{j \in \tilde{\mathcal{V}}_+} \|w_j\|^2 \|x_+\| \qquad \text{(Since } t \geq t_1')
$$

$$
\geq \frac{1}{4} \sum_{j \in \tilde{\mathcal{V}}_+} \|w_j\|^2 \sqrt{\mu} X_{\min} , \qquad \text{(by Lemma 11)}
$$

which suggests $\sum_{j \in \tilde{\mathcal{V}}_+} \|w_j\|^2 \leq \frac{1}{\sqrt{\mu} X_{\min}}$. A contradiction.

Therefore, we must have

$$
t_2^+ \leq t_1 + \frac{6}{\sqrt{\mu} n_+ X_{\min}} + \frac{4}{\sqrt{\mu} n_+ X_{\min}} \left( \log \frac{2}{\epsilon^2 \sqrt{\mu} X_{\min} W_{\min}^2} + 4n X_{\max} t_1 \right) . \qquad (39)
$$

Since the dominant term here is $\frac{4}{\sqrt{\mu} n_+ X_{\min}} \log \frac{2}{\epsilon^2 \sqrt{\mu} X_{\min} W_{\min}^2}$, we have $t_2^+ = \mathcal{O}(\frac{1}{n} \log \frac{1}{\epsilon})$. A similar analysis shows $t_2^- = \mathcal{O}(\frac{1}{n} \log \frac{1}{\epsilon})$. Therefore $t_2 = \max\{t_2^+, t_2^-\} = \mathcal{O}(\frac{1}{n} \log \frac{1}{\epsilon})$

**Complete the missing pieces** We have two claims remaining to be proved. The first is $\cos(x_+, \tilde{x}_a) \geq \frac{1}{2}$ when $t_1 \leq t \leq t_2^+$. Since $x_+ = \sum_{i \in \mathcal{I}_+} x_i$ and $\tilde{x}_a = \sum_{i \in \mathcal{I}_+} -\nabla_{\hat{y}} \ell(y_i, f_+(x_i; W, v)) x_i$. We simply use the fact that before $t_2^+$, we have, by Lemma 2,

$$
\frac{1}{2} \leq -\nabla_{\hat{y}} \ell(y_i, f_+(x_i; W, v)) = \leq \frac{3}{2} , \qquad (40)
$$

to show the following

$$\cos(x_+, \tilde{x}_a) = \frac{\langle x_+, \tilde{x}_a \rangle}{\|x_+\|\|\tilde{x}_a\|}$$

$$= \frac{\sum_{i,j \in \mathcal{I}_+} (-\nabla_{\hat{y}} \ell(y_i, f_+(x_i; W, v))) \langle x_i, x_j \rangle}{\sqrt{\sum_{i,j \in \mathcal{I}_+} \langle x_i, x_j \rangle} \sqrt{\sum_{i,j \in \mathcal{I}_+} (-\nabla_{\hat{y}} \ell(y_i, f_+(x_i; W, v)))^2 \langle x_i, x_j \rangle}}$$

$$\geq \frac{\frac{1}{2} \sum_{i,j \in \mathcal{I}_+} \langle x_i, x_j \rangle}{\sqrt{\sum_{i,j \in \mathcal{I}_+} \langle x_i, x_j \rangle} \sqrt{\sum_{i,j \in \mathcal{I}_+} (-\nabla_{\hat{y}} \ell(y_i, f_+(x_i; W, v)))^2 \langle x_i, x_j \rangle}}$$

$$\geq \frac{\frac{1}{2} \sum_{i,j \in \mathcal{I}_+} \langle x_i, x_j \rangle}{\sqrt{\sum_{i,j \in \mathcal{I}_+} \langle x_i, x_j \rangle} \sqrt{\sum_{i,j \in \mathcal{I}_+} (\frac{3}{2})^2 \langle x_i, x_j \rangle}} \geq \frac{1}{3},$$

since all $\langle x_i, x_j \rangle, i, j \in \mathcal{I}_+$ are non-negative.

The second claim is $\|\tilde{x}_a\| \geq \sqrt{\mu} n_+ X_{\min}/2$ is due to that

$$\|\tilde{x}_a\| = \sqrt{\sum_{i,j \in \mathcal{I}_+} (-\nabla_{\hat{y}} \ell(y_i, f_+(x_i; W, v)))^2 \langle x_i, x_j \rangle} \geq \frac{1}{2} \sqrt{\sum_{i,j \in \mathcal{I}_+} \langle x_i, x_j \rangle} = \frac{\|x_+\|}{2} \geq \frac{\sqrt{\mu} n_+ X_{\min}}{2},$$

where the last inequality is from Lemma 11. $\qquad \square$

### D.3 Proof of low-rank bias

So far we have proved the directional convergence at the early alignment phase and final $\mathcal{O}(1/t)$ convergence of the loss in the later stage. The only thing that remains to be shown is the low-rank bias. The proof is quite straightforward but we need some additional notations.

As we proved above, after $t_1$, neurons in $\mathcal{S}_+$ ($\mathcal{S}_-$) stays in $\mathcal{S}_+$ ($\mathcal{S}_-$). Thus the loss can be decomposed as

$$\mathcal{L} = \underbrace{\sum_{i \in \mathcal{I}_+} \ell \left( y_i, \sum_{j \in \tilde{\mathcal{V}}_+} v_j \langle w_j, x_i \rangle \right)}_{\mathcal{L}_+} + \underbrace{\sum_{i \in \mathcal{I}_-} \ell \left( y_i, \sum_{j \in \tilde{\mathcal{V}}_-} v_j \langle w_j, x_i \rangle \right)}_{\mathcal{L}_-},$$

where $\tilde{\mathcal{V}}_+ : \{j : w_j(t_1) \in \mathcal{S}_+\}$ and $\tilde{\mathcal{V}}_- : \{j : w_j(t_1) \in \mathcal{S}_-\}$. Therefore, the training after $t_1$ is decoupled into 1) using neurons in $\tilde{\mathcal{V}}_+$ to fit positive data in $\mathcal{I}_+$ and 2) using neurons in $\tilde{\mathcal{V}}_-$ to fit positive data in $\mathcal{I}_-$. We use

$$W_+ = [W]_{:, \tilde{\mathcal{V}}_+}, \quad W_- = [W]_{:, \tilde{\mathcal{V}}_-}$$

to denote submatrices of $W$ by picking only columns in $\tilde{\mathcal{V}}_+$ and $\tilde{\mathcal{V}}_-$, respectively. Similarly, we define

$$v_+ = [v]_{\tilde{\mathcal{V}}_+}, \quad v_- = [v]_{\tilde{\mathcal{V}}_-}$$

for the second layer weight $v$. Lastly, we also define

$$W_{\text{dead}} = [W]_{:, \tilde{\mathcal{V}}_{\text{dead}}}, v_{\text{dead}} = [v]_{\tilde{\mathcal{V}}_{\text{dead}}},$$

where $\tilde{\mathcal{V}}_{\text{dead}} := \{j : w_j(t_1) \in \mathcal{S}_{\text{dead}}\}$. Given these notations, after $t_1$ the loss is decomposed as

$$\mathcal{L} = \underbrace{\sum_{i \in \mathcal{I}_+} \ell \left( y_i, x_i^\top W_+ v_+ \right)}_{\mathcal{L}_+} + \underbrace{\sum_{i \in \mathcal{I}_-} \ell \left( y_i, x_i^\top W_- v_- \right)}_{\mathcal{L}_-},$$

and the GF on $\mathcal{L}$ is equivalent to GF on $\mathcal{L}_+$ and $\mathcal{L}_-$ separately. It suffices to study one of them. For GF on $\mathcal{L}_+$, we have the following important invariance Arora et al. (2018b) $\forall t \geq t_1$:

$$W_+^\top(t) W_+(t) - v_+(t) v_+^\top(t) = W_+^\top(t_1) W_+(t_1) - v_+(t_1) v_+^\top(t_1),$$

from which one has

$$
\begin{aligned}
\|W_+^\top(t)W_+(t) - v_+(t)v_+^\top(t)\|_2 =\ & \|W_+^\top(t_1)W_+(t_1) - v_+(t_1)v_+^\top(t_1)\|_2 \\
\leq\ & \|W_+^\top(t_1)W_+(t_1)\|_2 - \|v_+(t_1)v_+^\top(t_1)\|_2 \\
\leq\ & \operatorname{tr}(W_+^\top(t_1)W_+(t_1)) + \|v_+(t_1)\|^2 \\
=\ & 2\sum_{j\in\tilde{\mathcal{V}}_+}\|w_j(t_1)\|^2 \leq \frac{4\epsilon W_{\max}^2}{\sqrt{h}}|\tilde{\mathcal{V}}_+|\,,
\end{aligned}
$$

where the last inequality is by Lemma 3. Then one can immediately get

$$
\|v_+(t)v_+^\top(t)\|_2 - \|W_+^\top(t)W_+(t)\|_2 \leq \|W_+^\top(t)W_+(t) - v_+(t)v_+^\top(t)\|_2 \leq \frac{4\epsilon W_{\max}^2}{\sqrt{h}}|\tilde{\mathcal{V}}_+|\,,
$$

which is precisely

$$
\|W_+(t)\|_F^2 \leq \|W_+(t)\|_2^2 + \frac{4\epsilon W_{\max}^2}{\sqrt{h}}|\tilde{\mathcal{V}}_+|\,. \tag{41}
$$

Similarly, we have

$$
\|W_-(t)\|_F^2 \leq \|W_-(t)\|_2^2 + \frac{4\epsilon W_{\max}^2}{\sqrt{h}}|\tilde{\mathcal{V}}_-|\,. \tag{42}
$$

Lastly, one has

$$
\|W_{\text{dead}}\|_F^2 = \sum_{j\in\tilde{\mathcal{V}}_{\text{dead}}}\|w_j(t_1)\|^2 \leq \frac{4\epsilon W_{\max}^2}{\sqrt{h}}|\tilde{\mathcal{V}}_{\text{dead}}| \tag{43}
$$

Adding (41)(42)(43) together, we have

$$
\begin{aligned}
\|W(t)\|_F^2 =\ & \|W_+(t)\|_F^2 + \|W_-(t)\|_F^2 + \|W_{\text{dead}}\|_F^2 \\
\leq\ & \|W_+(t)\|_2^2 + \|W_-(t)\|_2^2 + \frac{4\sqrt{h}\epsilon W_{\max}^2}{\sqrt{h}} \leq 2\|W(t)\|_2^2 + 4\sqrt{h}\epsilon W_{\max}^2\,.
\end{aligned}
$$

Finally, since we have shown $\mathcal{L}\to 0$ as $t\to\infty$, then $\forall i\in[n]$, we have $\ell(y_i, f(x_i;W,v))\to 0$. This implies

$$
f(x_i;W,v) = -\frac{1}{y_i}\log\ell(y_i, f(x_i;W,v)) \to \infty\,.
$$

Because we have shown that

$$
f(x_i;W,v) \leq \sum_{j\in[h]}\|w_j\|^2\|x_i\| \leq \|W\|_F^2 X_{\max}\,,
$$

$f(x_i;W,v)\to\infty$ enforces $\|W\|_F^2\to\infty$ as $t\to\infty$, thus $\|W\|_2^2\to\infty$ as well. This gets us

$$
\limsup_{t\to\infty}\frac{\|W\|_F^2}{\|W\|_2^2} = 2\,.
$$

# E   EXISTENCE OF CARATHEODORY SOLUTION UNDER FIXED SUBGRADIENT $\sigma'(x) = \mathbb{1}_{x>0}$

In this Appendix, we first introduce the notion of solution we are interested in for the GF (2): Caratheodory solutions that satisfy (2) for almost all time $t$. Next, in Appendix E.2, we show that if we fix the ReLU subgradient as $\sigma'(x) = \mathbb{1}_{x>0}$, then global Caratheodory solutions exists for (2) under Assumption 1. Finally, we use simple examples to illustrate two points: 1) Caratheodory solutions cease to exist when ReLU subgradient at zero is chosen to be a fixed non-zero value, highlighting the importance of choosing the right subgradient for analysis; 2) Caratheodory solutions are potentially non-unique, the neurons' dynamical behavior could become somewhat irregular if certain solutions are not excluded, justifying the introduction of regular solutions (Definition 1).

## E.1   CARATHEODORY SOLUTIONS

Given an differential equation

$$\dot{\theta} = F(\theta), \theta(0) = \theta_0 \,, \tag{44}$$

with $F$ potentially be discontinuous, $\theta(t)$ is said to be a Caratheodory solution of (44) if it satisfies the following integral equation

$$\theta(t) = \theta_0 + \int_0^t F(\theta(\tau))d\tau \,, \tag{45}$$

for all $t \in [0, a)$, where $a \in \mathbb{R}_{\geq 0} \cup \infty$. In this section, we are interested in global Caratheodory solutions: $\theta(t)$ that satisfies (45) for all time $t \geq 0$.

## E.2   PROOF OF EXISTENCE OF REGULAR CARATHEODORY SOLUTIONS UNDER ASSUMPTION 1

In this section, we show the existence of global regular (Definition 1) Caratheodory solutions to $\dot{\theta} = F(\theta), \theta(0) = \theta_0$, where $\theta := \{W, v\}$ and $F := \nabla_{W,v}\mathcal{L}$ defined from a fixed choice of ReLU subgradient $\sigma'(x) = \mathbb{1}_{x>0}$, under Assumption 1. For the sake of a clear presentation, we first discuss the case of $\mathcal{S}_{\text{dead}} = \emptyset$, where all solutions are regular. then discuss the modifications one needs to make when $\mathcal{S}_{\text{dead}} \neq \emptyset$.

**Existence of Caratheodory solutions when $\mathcal{S}_{\text{dead}} = \emptyset$:** First of all, notice that $\nabla_{W,v}\mathcal{L}$ is continuous almost everywhere except for a zero measure set $\mathcal{A} = \{W, v : \exists i \in [n], j \in [h] \ s.t. \langle x_i, w_j \rangle = 0\}$, since discontinuity only happens when one has to evaluate $\sigma'(\langle x_i, w_j \rangle)$ at $\langle x_i, w_j \rangle = 0$ for some $i, j$. Being a finite union of hyperplanes, $\mathcal{A}$ has zero measure.

For points outside $\mathcal{A}$, the existence of a local solution is guaranteed by the generalized Caratheodory existence theorem in Persson (1975) (We refer readers to Appendix E.5 for the construction of such a local solution). The local solution can be extended to a global solution, as long as it does not encounter any point in $\mathcal{A}$ (the set where the flow is discontinuous). Whenever a point in $\mathcal{A}$ is reached, one requires extra certificates to extend the solution beyond that point. Simply speaking, the existence of a local solution around a point in $\mathcal{A}$ requires that the flow around this point does not push trajectories towards $\mathcal{A}$ from both sides of the zero measure set, causing an infinite number of crossings of $\mathcal{A}$, called Zeno behavior (van der Schaft & Schumacher, 2000; Maennel et al., 2018). See Figure 18 and 19 for an illustration. In Appendix E.5, we formally show that if there is no Zeno behavior, then a solution can be extended until reaching discontinuity in $\mathcal{A}$, and gets extended by leaving $\mathcal{A}$ immediately.[2]

One sufficient condition for avoiding Zeno behavior is to show: For each hyperplane $\mathcal{A}_{ij} := \{\langle x_i, w_j \rangle = 0\}$, all points in a neighborhood around this hyperplane $\mathcal{A}_{ij}$ must satisfy that the inner products between the normal vector of $\mathcal{A}_{ij}$ and the flow $F$ have the same sign. Formally speaking, we need that there exists $\delta > 0$, such that for all pair of $\theta_k, \theta_l \in \{\theta = (W, v) : 0 < |\langle x_i, w_j \rangle| < \delta\}$, we have $\langle \mathcal{N}_{\mathcal{A}_{ij}}, F(\theta_k) \rangle \langle \mathcal{N}_{\mathcal{A}_{ij}}, F(\theta_l) \rangle > 0$, here $\mathcal{N}_{\mathcal{A}_{ij}}$ should be a fixed choice of the normal vector of hyperplane $\mathcal{A}_{ij}$.

---

[2]Strictly speaking, Appendix E.5 is part of the proof but discussing the technical part right now disrupts the presentation.

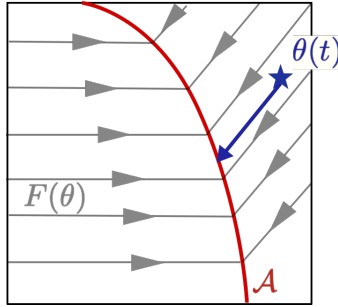 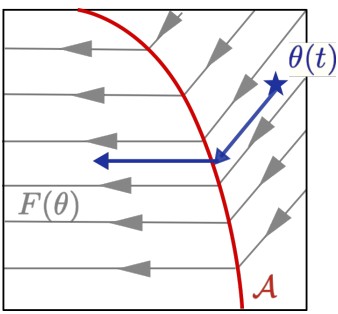

Figure 18: Non-existence of Caratheodory solution around points of discontinuity (Does not happen under Assumption 1). Since flow $F(\theta)$ is continuous The solution $\theta(t)$ can be extended until it reaches the points of discontinuity in $\mathcal{A}$, after which the solution is forced to stay within $\mathcal{A}$ (often referred as Zeno behavior: the solution is crossing $\mathcal{A}$ infinitely many times), and Caratheodory solution ceases to exist.

Figure 19: Existence of Caratheodory solution around points of discontinuity (Guaranteed by Assumption 1). When the solution $\theta(t)$ reaches $\mathcal{A}$, it immediately leaves $\mathcal{A}$ since the flow on the opposite side is flowing outward. This is a valid Caratheodory solution.

This inner product $\langle \mathcal{N}_{\mathcal{A}_{ij}}, F(\theta_k) \rangle$ between the normal vector and the flow is exactly computed as $\langle x_i, \nabla_{w_j} \mathcal{L} \rangle$. Under Assumption 1, we have a much stronger result than what is required in the last paragraph: we can show that on the entire parameter space, we have (shown in Appendix E.5)

$$y_i \text{sign}(v_j) \langle x_i, \nabla_{w_j} \mathcal{L} \rangle > 0 \,, \tag{46}$$

As such, since $v_j(t)$ does not change sign, Assumption 1 prevents Zeno behavior and ensures the existence of local solution around points in $\mathcal{A}$.

In summary, from any initialization, the Caratheodory solution can be extended (Persson, 1975) until the trajectory encounters points of discontinuity in $\mathcal{A}$, then the existence of a local solution is guaranteed by ensuring that the flow forces the solution to leave $\mathcal{A}$ immediately. Moreover, (46) ensures that $\mathcal{A}$ can only be crossed a finite number of times (every hyperplane can only be crossed once), after which no discontinuity is encountered and the solution can be extended to $t = \infty$. Therefore a global Caratheodory solution always exists.

**Existence of Caratheodory solutions when $\mathcal{S}_{\text{dead}} \neq \emptyset$:** Notice that when $\mathcal{S}_{\text{dead}} \neq \emptyset$. $\mathcal{A}$ contains boundary of $\mathcal{S}_{\text{dead}}$. If the solution gets extended to $\mathcal{A}$ where one neuron lands on the boundary of $\mathcal{S}_{\text{dead}}$, then this neuron stays at the boundary of $\mathcal{S}_{\text{dead}}$, i.e. the solution stays at $\mathcal{A}$. Therefore, the previous argument about existence does not apply.

However, one only needs very a minor modification: If at time $t_0$, the solution enters $\mathcal{A}$ by having one neuron (say $w_j(t)$) land on the boundary of $\mathcal{S}_{\text{dead}}$, set $w_j(t) \equiv w_j(t_0)$ and $v_j(t) \equiv v_j(t_0)$ for $t \geq t_0$, then exclude $\{w_j, v_j\}$ from the parameter space and continue constructing and extending local solutions for other parameters via the previous argument. This shows the existence of the Caratheodory solution under non-empty $\mathcal{S}_{\text{dead}}$, and by our construction, the solution is regular.

### E.3 NON-EXISTENCE OF CARATHEODORY SOLUTION UNDER OTHER FIXED SUBGRADIENT

Consider the following simple example: The training data consists of a single data point $x = [1, 0]^\top$, $y = -1$, and the network consists of a single neuron $\{w, v\}$ initialized at $\{w(0) = [0, 1]^\top, v(0) = 1\}$. See Figure 20 for an illustration.

When the ReLU subgradient is chosen to be $\sigma'(x) = \mathbb{1}_{x>0}$, the Caratheodory solution $\{w(t) \equiv [0, 1], v(t) \equiv 1\}$ exists, i.e. the neuron stays at the boundary of $\mathcal{S}_{\text{dead}} := \{w : \langle x, w \rangle \leq 0\}$.

If the ReLU subgradient is chosen to be $\sigma'(0) = a > 0$, then the Caratheodory solution ceases to exist: the neuron cannot stay at the boundary $\langle x, w \rangle = 0$ of $\mathcal{S}_{\text{dead}}$, because the non-zero $\sigma'(0)$ pushes

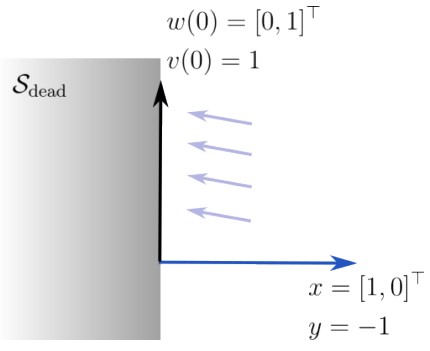 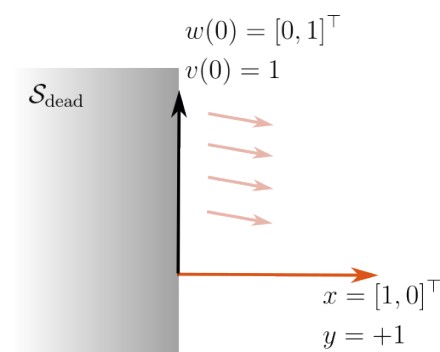

Figure 20: Non-existence of Caratheodory solution under fixed ReLU subgradient $\sigma'(0) = a > 0$. For any $t > 0$, $w(t)$ cannot stay at $[0, 1]^\top$, because the subgradient $\sigma'(0) = a > 0$ is positive, leading to non-zero $\dot{w}$. At the same time, it cannot enter the interior of $\mathcal{S}_{\text{dead}}$, because there is no flow in the interior.

Figure 21: Non-uniqueness of Caratheodory solutions. One solution is that $w(t) \equiv w(0), v(t) \equiv v(0)$, i.e. the neuron stays within $\mathcal{S}_{\text{dead}}$. However, the neuron can leave $\mathcal{S}_{\text{dead}}$ at any time $t_0 > 0$ then follow the flow outside $\mathcal{S}_{\text{dead}}$, which is also a valid Caratheodory solution.

it towards the interior of $\mathcal{S}_{\text{dead}}$. However, the neuron cannot enter the interior of $\mathcal{S}_{\text{dead}}$ because the flow is all zero within the interior of $\mathcal{S}_{\text{dead}}$.

To see this formally, suppose $\{w(t) = w(0), v(t) = v(0)\}$ for $t \in [0, t_0]$ (neuron stay at $\{w(0), v(0)\}$), then by definition of Caratheodory solution, we have

$$\int_0^{t_0} \nabla_{w,v}\mathcal{L}(w(0), v(0))dt = t_0\nabla_{w,v}\mathcal{L}(w(0), v(0)) = 0\,,$$

suggesting $\nabla_{w,v}\mathcal{L}(w(0), v(0)) = 0$, which is not true when $\sigma'(0) > 0$, thus a contradiction. Now suppose $w(t_0) \in \text{Int}(\mathcal{S}_{\text{dead}})$ for some $t_0$, then it must be that $w(t) \in \text{Int}(\mathcal{S}_{\text{dead}}), \forall 0 < t \leq t_0$, otherwise it leads to the same contradiction as in previous paragraph. By definition of Caratheodory solution, we have

$$\int_0^{t_0} \nabla_{w,v}\mathcal{L}(w(t), v(t))dt = w(t_0) - w(0)\,.$$

The left-hand side is zero because $w(t) \in \text{Int}(\mathcal{S}_{\text{dead}}) \Rightarrow \nabla_{w,v}\mathcal{L}(w(t), v(t)) = 0, \forall 0 < t \leq t_0$. The right-hand side is non-zero because $w(t_0) \in \text{Int}(\mathcal{S}_{\text{dead}})$, thus a contradiction. Similarly, $w(t)$ cannot enter $\mathcal{S}_{\text{dead}}^c$. Therefore, the Caratheodory solution $\{w(t), v(t)\}$ does not exist for any $t > 0$.

### E.4 NON-UNIQUENESS OF CARATHEODORY SOLUTIONS

Consider the following simple example: The training data consists of a single data point $x = [1, 0]^\top$, $y = 1$, and the network consists of a single neuron $(w, v)$ initialized at $w(0) = [0, 1]^\top, v(0) = 1$. See Figure 21 for an illustration.

We consider the case when the ReLU subgradient is chosen to be $\sigma'(x) = \mathbb{1}_{x>0}$. There exists one Caratheodory solution $w(t) \equiv [0, 1]^\top$, $v(t) \equiv 1$, i.e. the neuron stays at the boundary of $\mathcal{S}_{\text{dead}} := \{w : \langle x, w \rangle \leq 0\}$. However, consider $\tilde{w}(t), \tilde{v}(t)$ being the solution to the following ode (the one that neuron follows once enters the positive orthant):

$$\dot{\tilde{w}} = y\exp(-y\tilde{v}\langle x, \tilde{w}\rangle)\tilde{v}x, \quad \dot{\tilde{v}} = y\exp(-y\tilde{v}\langle x, \tilde{w}\rangle)\langle x, \tilde{w}\rangle, \tilde{w}(0) = w(0), \tilde{v}(0) = v(0)\,. \quad (47)$$

Then for any $t_0 \geq 0$,

$$w(t) = \mathbb{1}_{t<t_0}w(0) + \mathbb{1}_{t\geq t_0}\tilde{w}(t - t_0), v(t) = \mathbb{1}_{t<t_0}v(0) + \mathbb{1}_{t\geq t_0}\tilde{v}(t - t_0)$$

is a Caratheodory solution. This example shows that the Caratheodory solution could be non-unique.

This is somewhat troublesome for our analysis, one would like that all neurons in $\mathcal{S}_{\text{dead}}$ stay within $\mathcal{S}_{\text{dead}}$, but Caratheodory solutions do not have this property, and in fact, as long as the neuron is on

the boundary of $\mathcal{S}_{\text{dead}}$, and the flow outside $\mathcal{S}_{\text{dead}}$ is pointing away from the boundary, the neuron can leave $\mathcal{S}_{\text{dead}}$ at any time and it does not violate the definition of a Caratheodory solution. Therefore, for our main theorem, we added an additional regularity condition (Definition 1) on the solution, forcing neurons to stay within $\mathcal{S}_{\text{dead}}$.

**Remark 6.** *This issue of having irregular solutions is not specific to our choice of the notion of solutions. Even if one considers more generally the Filippov solution Filippov (1971) of the differential inclusion in* (2)*, the same issue of non-uniqueness persists and needs attention when analyzing neuron dynamics.*

**Remark 7.** *Although irregular solutions are not desired for analyzing neuron behaviors, as we see in this example, they are rare cases under very specific initialization of the neurons and thus can be avoided by randomly initializing the weights.*

### E.5 Constructing Global Caratheodory Solution

In this section, we formally show that if there is no Zeno behavior, then a solution can be extended until reaching discontinuity in $\mathcal{A}$, and gets extended by leaving $\mathcal{A}$ immediately, leading to a construction of global Caratheodory solution. The only ingredient that is needed is the existence theorem in Persson (1975, Theorem 2.3), showing that if $F(\theta)$ is continuous and $\forall \theta$

$$\|F(\theta)\|_F \leq M(1 + \|\theta\|_F), \tag{48}$$

for some $M > 0$, then global solution of $\dot{\theta} = F(\theta)$ exists. Obviously, this result cannot be applied directly for two reasons: a) it requires continuity of the flow; b) it requires linear growth of $\|F(\theta)\|_F$ w.r.t. $\|\theta\|_F$. The key idea is constructing a local solution by restricting the flow to a neighborhood of initial conditions where a) and b) are satisfied, and then extending this solution to a global one.

As we discussed in Appendix E.2, we can assume $\mathcal{S}_{\text{dead}} = \emptyset$ without loss of generality. Moreover, it suffices to show that starting from an initialization $\theta(0) = \{W(0), v(0)\}$ outside $\mathcal{A}^3$, we can construct either: 1) a global solution without encountering any point in $\mathcal{A}$; or 2) a local solution that lands on $\mathcal{A}$ at some $t_0$ then leave $\mathcal{A}$ immediately. Because if 2) happens, we take the end of this local solution as a new initial condition and repeat this argument. Importantly, 2) cannot happen infinitely many times because we have shown in Appendix E.2 that $\mathcal{A}$ can only be crossed finitely many times, thus 1) must happen, resulting in a global solution.

**Construct local solution from initial condition**: Now given an initial condition $\theta(0) = \{W(0), v(0)\}$, define the following two sets (Notation-wise, we drop the dependency on $\{W(0), v(0)\}$ for simplicity):

$$\Theta_0 := \{\theta = (W, v) : \mathcal{L}(W, v) \leq \mathcal{L}(W(0), v(0)), \text{sign}(v_j) = \text{sign}(v_j(0)), \forall j \in [h]\},$$
$$\Theta_1 := \{\theta = (W, v) : \forall i \in [n], j \in [h], \langle x_i, w_j \rangle \langle x_i, w_j(0) \rangle > 0\},$$

$\Theta_1$ is the positive invariant set of $\{W(0), v(0)\}$: all solutions from $\{W(0), v(0)\}$ never leaves $\Theta_1$, so it suffices to study the flow within $\Theta_1$ for the existence of solutions. Moreover, $\Theta_0$ is the intersection of a closed set $\{v : \text{sign}(v_j) = \text{sign}(v_j(0))\}$ and the pre-image of a continuous function $\mathcal{L}$ on the range $[0, \mathcal{L}(W(0), v(0))]$ thus closed. $\Theta_2$ is the largest connected set that contains $\{W(0), v(0)\}$ without point of discontinuity.

Consider the following set

$$\tilde{\Theta}_1 := \Theta_0 \cap \text{cl}(\Theta_1). \tag{49}$$

Then $\tilde{\Theta}_1$ is closed. Consider a new flow $F_1^{\text{cl}}$ on $\tilde{\Theta}_1$ such that $F_1^{\text{cl}} = F = \nabla_{W,v}\mathcal{L}$ for all $\theta \in \text{Int}(\tilde{\Theta}_1)$, and $F_1^{\text{cl}}(\theta) = \lim_{k \to \infty} F(\theta_k)$ for all $\theta \in \tilde{\Theta} \setminus \text{Int}(\tilde{\Theta}_1)$, where $\theta_k \in \text{Int}(\tilde{\Theta}_1), k = 1, 2, \cdots$ is a convergent sequence to $\theta$.

$F_1^{\text{cl}}|_{\tilde{\Theta}_1}$ is continuous by construction, and we can show that (at the end of this section)

$$\|F_1^{\text{cl}}(\theta)\|_F \leq C\|\theta\|_F, \forall \theta \in \tilde{\Theta}_1. \tag{50}$$

By a generalized version of the Tietze extension theorem (Ercan, 1997), there exists continuous $\tilde{F}_1$ on the entire parameter space, such that

$$F_1^{\text{cl}}(\theta) = \tilde{F}_1(\theta), \forall \theta \in \tilde{\Theta}_1, \tag{51}$$

---

[3] initial condition within $\mathcal{A}$ is taken care of by 2).

and
$$\|\tilde{F}_1(\theta)\|_F \le C\|\theta\|_F, \forall \theta\,. \tag{52}$$

Because now $F = F_1^{\text{cl}} = \tilde{F}_1$ on $\text{Int}(\tilde{\Theta}_1)$, any solution $\tilde{\theta}_1(t)$ of $\dot{\tilde{\theta}}_1 = \tilde{F}_1(\tilde{\theta}_1), \tilde{\theta}_1(0) = \theta(0)$ (existence guaranteed by Persson (1975)) gives a local solution of $\dot{\theta} = F(\theta), \theta(0) = \theta(0)$, for $t \le t_0$, where $t_0 := \inf\{t : \tilde{\theta}_1(t) \notin \tilde{\Theta}_1\}$.

If $t_0 = \infty$, one has a global solution and the construction is finished. If $t_0 < \infty$, it must be that $\tilde{\theta}_1(t_0) \in \mathcal{A}$ (since $\tilde{\theta}_1$ must leave $\tilde{\Theta}$ via the boundary of $\Theta_2$). Now we need to construct a solution that leaves $\mathcal{A}$ immediately.

**Construct local solution that leaves $\mathcal{A}$**: As we discussed, $\mathcal{A}$ is a union of hyperplanes. For simplicity, let us assume $\tilde{\theta}(t_0)$ is not at the intersection of two hyperplanes (the treatment is similar but tedious, we will make remarks in the end).

Now $\tilde{\theta}(t_0)$ lands on a single hyperplane, let it be $\{\theta : \langle x_{i^*}, w_{j^*} \rangle = 0\}$, we define
$$\Theta_2 := \{\theta = (W, v) : \forall i \ne i^*, j \ne j^*, \langle x_i, w_j \rangle \langle x_i, w_j(0) \rangle > 0,$$
$$\text{and } \langle x_{i^*}, w_{j^*} \rangle \langle x_{i^*}, w_{j^*}(0) \rangle < 0\}\,,$$

and we let
$$\tilde{\Theta}_2 := \Theta_0 \cap \text{cl}(\Theta_2)\,, \tag{53}$$

It is clear that, from the definition of $\Theta_2$, any solution we construct that leaves $\mathcal{A}$ immediately after $t_0$ must enter $\text{Int}(\tilde{\Theta}_2)$. To construct the solution, we just need to repeat the first part, but now for $\tilde{\Theta}_2$: We construct $F_2^{\text{cl}}$ that is continuous on $\tilde{\Theta}_2$ and agrees with $F$ on the interior, then extends $F_2^{\text{cl}}$ to $\tilde{F}$ on the entire parameter space. Consider the solution $\tilde{\theta}_2(t)$ of
$$\dot{\tilde{\theta}}_2 = F(\tilde{\theta}_2), \tilde{\theta}_2(0) = \tilde{\theta}_1(t_0)\,, \tag{54}$$

gives a local solution of
$$\dot{\theta} = F(\theta), \theta(0) = \tilde{\theta}_1(t_0)\,. \tag{55}$$

Because we have shown that Zeno behavior does not happen, $\tilde{\theta}_2(t)$ leaves $\mathcal{A}$ immediately and enters $\text{Int}(\tilde{\Theta}_2)$. We just pick any $\tau_0 > 0$ such that $\tilde{\theta}_2(\tau_0) \in \text{Int}(\tilde{\Theta}_2)$ then
$$\theta(t) = \mathbb{1}_{t \le t_0} \tilde{\theta}_1(t) + \mathbb{1}_{t_0 < t \le t_0 + \tau_0} \tilde{\theta}_2(t - t_0)\,, \tag{56}$$

is a Caratheodory solution to $\dot{\theta} = F(\theta), \theta(0) = \theta(0)$ for $t \le t_0 + \tau_0$. This is exactly what we intended to show.

**Remark 8.** *When $\tilde{\theta}(t_0)$ lands at the intersection of two (or more) hyperplanes, the only difference is that now there could be more regions to escape to. But under Assumption 1, (46) suggests that the solution must cross all hyperplanes after $t_0$, leaving one unique region similar to $\Theta_2$. Then one constructs the local solution following previous procedures.*

**Complete the missing pieces** To complete the proof, there are two statements ((46) and (50)) left to be shown.

To show (46), we start from the derivative
$$\nabla_{w_j}\mathcal{L} = -\sum_{k=1}^n \mathbb{1}_{\langle x_k, w_j \rangle > 0} \nabla_{\hat{y}}\ell(y_k, f(x_k; W, v)) x_k \text{sign}(v_j(0)) \|w_j\|\,,$$
$$= \sum_{k=1}^n \mathbb{1}_{\langle x_k, w_j \rangle > 0} y_k \exp(-y_k f(x_k; W, v)) x_k \text{sign}(v_j(0)) \|w_j\|\,,$$

and we have
$$y_i \text{sign}(v_j) \langle x_i, \nabla_{w_j}\mathcal{L} \rangle = \sum_{k=1}^n \mathbb{1}_{\langle x_k, w_j \rangle > 0} \exp(-y_k f(x_k; W, v)) \langle y_i x_i, y_k x_k \rangle \|w_j\|$$
$$\ge \sum_{k=1}^n \mathbb{1}_{\langle x_k, w_j \rangle > 0} \exp(-y_k f(x_k; W, v)) \mu \|x_k\| \|x_i\| \|w_j\| > 0\,,$$

since there is at least one summand ($\mathcal{S}_{\text{dead}} = \emptyset$), the summation is always positive.

To show $(50)^4$, we first consider $\theta \in \text{Int}(\tilde{\Theta}_1)$, and we have

$$
\begin{aligned}
\sum_{j=1}^{h} \|\nabla_{w_j} \mathcal{L}\|^2 &= \sum_{j=1}^{h} \left\| \sum_{k=1}^{n} \mathbb{1}_{\langle x_k, w_j \rangle > 0} y_k \exp(-y_k f(x_k; W, v)) x_k v_j \right\|^2 \\
&\leq \sum_{j=1}^{h} \left( \sum_{k=1}^{n} \exp(-y_k f(x_k; W, v)) \|x_k\| |v_j| \right)^2 \\
&\leq \sum_{j=1}^{h} |v_j|^2 \cdot \left( X_{\max} \sum_{k=1}^{n} \exp(-y_k f(x_k; W, v)) \right)^2 \\
&= \sum_{j=1}^{h} |v_j|^2 \cdot (X_{\max} \mathcal{L}(W, v))^2 \\
&\leq \sum_{j=1}^{h} |v_j|^2 \cdot (X_{\max} \mathcal{L}(W(0), v(0)))^2 = X_{\max}^2 \mathcal{L}^2(W(0), v(0)) \|v\|^2,
\end{aligned}
$$

similarly, we also have

$$
\begin{aligned}
\sum_{j=1}^{h} \|\nabla_{v_j} \mathcal{L}\|^2 &= \sum_{j=1}^{h} \left\| \sum_{k=1}^{n} \mathbb{1}_{\langle x_k, w_j \rangle > 0} y_k \exp(-y_k f(x_k; W, v)) \langle x_k, w_j \rangle \right\|^2 \\
&\leq \sum_{j=1}^{h} \left( \sum_{k=1}^{n} \exp(-y_k f(x_k; W, v)) \|x_k\| \|w_j\| \right)^2 \\
&\leq \sum_{j=1}^{h} \|w_j\|^2 \cdot \left( X_{\max} \sum_{k=1}^{n} \exp(-y_k f(x_k; W, v)) \right)^2 \\
&= \sum_{j=1}^{h} \|w_j\|^2 \cdot (X_{\max} \mathcal{L}(W, v))^2 \\
&\leq \sum_{j=1}^{h} \|w_j\| \cdot (X_{\max} \mathcal{L}(W(0), v(0)))^2 = X_{\max}^2 \mathcal{L}^2(W(0), v(0)) \|W\|_F^2.
\end{aligned}
$$

Therefore, we have $\forall \theta \in \text{Int}(\tilde{\Theta}_1)$

$$
\begin{aligned}
\|F_1^{\text{cl}}(\theta)\|_F^2 &= \|F(\theta)\|_F^2 = \sum_{j=1}^{h} (\|\nabla_{w_j} \mathcal{L}\|^2 + \|\nabla_{v_j} \mathcal{L}\|^2) \\
&\leq X_{\max}^2 \mathcal{L}^2(W(0), v(0)) (\|W\|_F^2 + \|v\|^2) = X_{\max}^2 \mathcal{L}^2(W(0), v(0)) \|\theta\|_F^2,
\end{aligned}
$$

which gives (50) with $C = X_{\max} \mathcal{L}(W(0), v(0))$.

Then for $\theta \in \tilde{\Theta} \setminus \text{Int}(\tilde{\Theta}_1)$, $\|F_1^{\text{cl}}(\theta)\| = \lim_{k \to \infty} \|F(\theta_k)\| \leq C \lim_{k \to \infty} \|\theta_k\| = C\|\theta\|$, given some Cauchy sequence $\theta_k \in \text{Int}(\tilde{\Theta}_1), k = 1, 2, \cdots$ convergent to $\theta$. This finishes proving (50).

---

[4]We show it for exponential loss, the case of logistic loss is similar

## F    EXTEND MAIN RESULTS TO SOLUTIONS TO DIFFERENTIAL INCLUSION

For Filippov (1971) solutions (regular according to Definition 1) to the differential inclusion (2), our Theorem 1 remains the same. The only difference is that the notion of $x_a(w)$ in (4) is no longer a singleton, but rather an element from a set:

$$x_a(w) \in \left\{ \sum_i \sigma'(\langle x_i, w \rangle) y_i x_i \right\} , \tag{57}$$

where $\sigma'(\langle x_i, w \rangle)$ is a subgradient of ReLU activation $\sigma(z)$ at $z = \langle x_i, w \rangle$. Therefore, the proof of Theorem 1 shall be modified (which can be done) to consider all possible choices of $x_a(w)$.

In the case of $\sigma'(z)|_{z=0} = 0$, $x_a(w)$ become a singleton $\sum_{i:\langle x_i, w \rangle > 0} y_i x_i$, which simplifies our discussions. This is the main reason we opt to fix this subgradient $\sigma'(z)$ in the main paper.

