# OpenReview forum: "Early Neuron Alignment in Two-layer ReLU Networks with Small Initialization"
_ICLR.cc/2024/Conference — ICLR 2024 poster_

### Official Review · Reviewer_Qg2K · 2023-10-26

**Soundness:** 2 fair
**Presentation:** 3 good
**Contribution:** 2 fair
**Rating:** 5
**Confidence:** 3

**Summary:**

This paper studies the learning dynamics in the special case of two-layer ReLU neural network with small initialization.

**Strengths:**

The results of this paper extend prior results from infinitesimal initialization to finitely small initialization.

The relation and difference with prior results are clearly presented. The paper is clearly written and easy to follow. The figures are helpful for understanding the argument.

**Weaknesses:**

The setting of the neural network is unconventional. It requires the second layer weights to depend on the first layer weights in a way as shown in Eq.(3), instead of independently initialized.  This setting is used neither in practice nor in most theoretical analysis. According to the analysis,  I doubt that the results of this paper hold without this restriction on the second layer weights.

> The paper has some discussion on this setting. However, it does not justify the validity of this setting. That it is commonly assumed in other papers does not directly justify. I would like to see some analysis, or at least some intuition, on why the results would hold on the natural setting.

The assumption on the data (Assumption 1) is strong, as it is not met by almost all real data.

The significance of the results is limited, as it is an extension of similar results from $\epsilon \to 0$ to the finite but small $\epsilon$.

**Questions:**

In Eq.(5), why the R.H.S. is independent of the network output $f(x_i)$? Or, why there is no such term $y_i-f(x_i)$ which usually appears in the expression of gradients.

---

> ### Author Response · Authors · 2023-11-21
> **Thank you for your review**
>
> We thank the reviewer for the comments, please see our response to your questions/concerns:
>
> **Clarification on balanced initialization**: We would like to point out that our balanced initialization assumption is made for theoretical analysis, and we believe it is not necessary for the dynamics behavior described by our Theorem 1 to happen in practice: All experiments in our manuscript use Gaussian initialization with small variance $\alpha^2$ with some $\alpha<<1$ (please see highlighted text on Page 9 for the details), and the numerical results can be well explained by our Theorem 1, despite not having a balanced initialization.
>
> **What happens if initialization is not balanced**: Having random Gaussian initialization (not balanced) for both layers only introduces additional technical challenges for the analysis, and we would like to discuss them here in detail: Even if the initialization is not balanced, the imbalance is preserved, i.e., one has $v_j^2(t)-\Vert w_j(t)\Vert^2=v_j^2(0)-\Vert w_j(0)\Vert^2=\mathcal{O}(\epsilon^2)$, thus $v_j(t)=sign(v_j(t))\Vert w_j(t)\Vert+\mathcal{O}(\epsilon^2)$. The last equation is ALL we use in the analysis regarding the last layer. In the balanced case, there is no $\mathcal{O}(\epsilon^2)$ term, and we also have $sign(v_j(t))=sign(v_j(0))$ fixed. If the initialization is not balanced, there is an additional $\mathcal{O}(\epsilon^2)$ error term in all our analyses, but it can be well controlled when $\epsilon$ is sufficiently small. The only nontrivial part is that the $sign(v_j(t))$ is no longer fixed. Thus, one needs to keep track of the changes in $sign(v_j(t))$ over the GF trajectory, which is an interesting future research direction. Although there are additional technical challenges, NONE of them substantially change the GF dynamics: one still has the two-phase training (first alignment, then final convergence), and the neuron's directional dynamics still follow equation (4), with $sign(v_j(0))$ replaced by $sign(v_j(t))$. Therefore, our theoretical results would still hold without a balanced initialization assumption, and this is exactly what we show in the numerical experiments.
>
> **Assumption on the data**: Although we assumed separable data for our analysis, we would like to point out that our numerical experiments do not enforce this assumption: For the two-MNIST-digits experiment, we show that centered data approximately satisfies our Assumption 1 (please see the left-most plot in Figure 5). Despite not strictly satisfying Assumption 1, the numerical results can be well explained by our Theorem 1. That being said, we are working diligently to extend our results to more general/relaxed assumptions in our future work. Moreover, we would like to refer the reviewer to the global response we posted for a detailed discussion about our assumption on the data.

---

> > ### Author Response · Authors · 2023-11-21
> > **Part 2**
> >
> > **significance of our results**: Our results are NOT some simple extension to prior results that study $\epsilon\rightarrow 0$. First of all, $\epsilon\rightarrow 0$ is impractical because all-zero initialization keeps the weights fixed at zero throughout training (or, equivalently, the origin is a saddle point of the GF dynamics). For non-zero $\epsilon$, if one makes the initialization scale, $\epsilon$, smaller and smaller, the GF dynamics will further slow down. This can be theoretically shown by our Lemma 3 in Appendix B.1: Given an $\epsilon$-small initialization, all weights are close to zero for at least $T=\Theta(\log\frac{1}{\epsilon})$ amount of time. When $\epsilon\rightarrow 0$, we have $T\rightarrow \infty$, meaning the weights do not move away from zero for an infinite amount of time! Therefore, any analysis with $\epsilon\rightarrow 0$ can not be implemented in practice. Our results work for a finite $\epsilon$, a choice that can be used in practice (We note that our analysis is for GF; there needs to be a choice of step size when applying our results to gradient descent, which can be obtained from [Elkabetz'21]). Secondly, it requires much more technical analyses for deriving quantitative bounds under a finite $\epsilon$, as we explained in "Weakness in prior analyses" on Page 4 and our proof sketch on Page 7, and our results make significant technical contributions to understanding the GF dynamics with small initialization.
> >
> > **RHS of equation (5)**: During the alignment phase (first $\Theta(\log\frac{1}{\epsilon})$ time of the GF trajectory), we can formally show that the network's output is small for all input $x_i$, i.e. $\max_i|f(x_i)|\leq \mathcal{O}(\epsilon)$ (please see our Lemma 3 in Appendix B.1). Therefore, in the approximation we have in equation (4)-(5), we can approximate $y_i-f(x_i)$ by $y_i$, with an error of $\mathcal{O}(\epsilon)$. One minor point is that one would get $y_i-f(x_i)$ in the gradient if the $l_2$ loss is used. Here, we are using exponential/logistic loss, the gradient term has $y_i\exp(y_if(x_i))$, and we approximate it by $y_i$.
> >
> > We hope our response addresses your concerns and the reviewer can re-evaluate our manuscript based on our response. If you have additional questions/concerns, please feel free to post another comment. We will respond to them before the discussion period ends.
> >
> > Reference:
> >
> > [Elkabetz'21] Omer Elkabetz, and Nadav Cohen. "Continuous vs. discrete optimization of deep neural networks.", NeurIPS, 2021

---

### Official Review · Reviewer_5S56 · 2023-10-30

**Soundness:** 3 good
**Presentation:** 4 excellent
**Contribution:** 3 good
**Rating:** 8
**Confidence:** 4

**Summary:**

The paper improves on previous theoretical analysis of the early alignement phase of the neurons of a shllow neural network initialized with small weights. This allows them to prove quantitative bounds in terms of the initialization scale, time, number of neurons and number of datapoints required to guarantee convergence.

**Strengths:**

The paper is easy to follow and explains well the previous issues and how they are solved. It is nice that the results apply to deterministic initialization of the weights, and do not require a random initialization (though they can of course be applied to this case).

**Weaknesses:**

The assumption of positively correlated labels and balancedness are very strong, and usually are not true in practice.

**Questions:**

The description of Assumption 2 before the statement of the assumption does not match the statement of the assumption.

---

> ### Author Response · Authors · 2023-11-21
> **Thank you for your positive review**
>
> We thank the reviewer for the positive review. Please see our response to your questions/concerns:
>
> **Assumption on the data and initialization**: We agree with the reviewer that our assumptions that data is positively correlated and initialization is balanced are often unrealistic. However, we believe they are not necessary for the dynamics behavior described by our Theorem 1 to happen in practice. We would like to point out that our numerical experiments enforce NONE of these assumptions: For the two-MNIST-digits experiment, we show that centered data approximately satisfies our Assumption 1 (Please see the left-most plot in Figure 5), and we initialize the weights by Gaussian initialization with small variance (not balanced). Despite not strictly satisfying these assumptions, the numerical results can be well explained by our Theorem 1. That being said, we are working diligently to extend our results to more general/relaxed assumptions in our future work. Moreover, we would like to refer the reviewer to the global response we posted for a detailed discussion about our assumption on the data.
>
> We hope our response addresses your concerns. If you have additional questions/concerns, please feel free to post another comment. We will respond to them before the discussion period ends.

---

### Official Review · Reviewer_osUg · 2023-11-01

**Soundness:** 3 good
**Presentation:** 4 excellent
**Contribution:** 1 poor
**Rating:** 5
**Confidence:** 3

**Summary:**

The paper studies the problem of training a two-layer ReLU network for binary
classification using gradient flow with small initialization on well-separated input datasets, i.e. datasets $(x_i, y_i)_{i \in [n]}$ with $ \min \frac{\langle x_iy_i, x_jy_j  \rangle}{\Vert x_i \Vert_2 \Vert x_j \Vert_2 } \geq \mu$.
They show that in time $O(\frac{\log(n)}{\sqrt{\mu}})$ all neurons are well aligned with the input data, which means that positive neurons show in the same direction as the positively labeled points (or have a negative scalar prouct with all vectors of the dataset) (and an equivalent result for negative neurons).
Furhter they show that after the early alignment phase, the loss converges to zero at a $O(1/t)$ rate, and the weight
matrix on the first layer is approximately low-rank. Numerical experiments are provided.

**Strengths:**

All claims are proven and illustrations are supporting the explanations.

**Weaknesses:**

The assumption that the dataset is well seperated is very strong. I don't see why one would use a neural network on such a dataset rather than linear regression.

**Questions:**

-

---

> ### Author Response · Authors · 2023-11-21
> **Thank you for your review**
>
> We thank the reviewer for acknowledging the strengths of our paper. However, we strongly (yet respectfully) disagree with the reviewer's comment on the weaknesses. Please see our response to your questions/concerns:
>
> **Goal/Contribution of this paper**: We believe the reviewer's assessment of the contribution of our paper is inaccurate. The well-separated dataset studied in this paper is indeed easy and a linear classifier can achieve perfect accuracy. However, the point of this paper is NOT to find a classifier that fits the dataset, but rather to understand (a) how one can successfully train a ReLU network to fit the data, without getting stuck at some local minimum (as Reviewer Qhf5 pointed out, this can happen when training ReLU on linearly separable data); (b) why using a small initialization in the training of a two-layer ReLU network will lead to a classifier with special property (in our case, low-rank), while a network trained using a large initialization scale does not have such property [Chizat'19]. Through a complete, non-asymptotic, quantitative analysis of the GF on two-layer ReLU networks with small initialization, our results show that one can achieve the global minimum of the loss (with a characterization of the convergence rate), and provide a detailed analysis of how network weights evolve during training, along with its connection to the low-rank property of the weight matrix. As such, our paper makes a significant contribution toward understanding the implicit bias of small initialization in training a ReLU network.
>
> **Assumption on the data**: Although we assumed separable data for our analysis, we believe it is not necessary for the dynamics behavior described by our Theorem 1 to happen in practice. We would like to point out that our numerical experiments do not enforce this assumption: For the two-MNIST-digits experiment, we show that centered data only approximately satisfies our Assumption 1 (Please see the left-most plot in Figure 5). Despite not strictly satisfying Assumption 1, the numerical results can be well explained by our Theorem 1. That being said, we are working diligently to extend our results to more general/relaxed assumptions in our future work. Moreover, we would like to refer the reviewer to the global response we posted for a detailed discussion about our assumption on the data.
>
> We hope our response addresses your concerns and the reviewer can re-evaluate our manuscript based on our response. If you have additional questions/concerns, please feel free to post another comment. We will respond to them before the discussion period ends.
>
> Reference:
>
> [Chizat'19] Chizat, L., Oyallon, E., and Bach, F. On lazy training in differentiable programming. NeurIPS, 2019.

---

> > ### Comment · Reviewer_osUg · 2023-12-01
> >
> > Thanks for addressing my concerns.
> >
> > My point was not about that the paper is about to find a classifier that fits the dataset and I am aware of the fact that the goal is to understand the training process of ReLU networks. The reason I mentioned linear regression is because it is a very similar model, in fact replacing the ReLU activation function by the identy gives us a linear classifier. For this reason I would prefer an assumption that helps to understand why Two ReLU networks work well on data that is approximately seperable not just seperable data.
> >
> > I remain my score as I still think that the assumption is too strong. If the results are extended to more general/relaxed assumptions I would agree that the paper should be accepted.

---

### Official Review · Reviewer_Qhf5 · 2023-11-03

**Soundness:** 3 good
**Presentation:** 3 good
**Contribution:** 3 good
**Rating:** 8
**Confidence:** 4

**Summary:**

This paper studies the problem of training a two-layer ReLU network classifier via gradient flow under small initialization. Training dataset assumes well-separated input vectors. Analysis of the neurons’ directional dynamics establishes an upper bound on the time it takes for all neurons to achieve good alignment with the input data. Numerical experiment on the MNIST dataset validate the theoretical findings.

**Strengths:**

+ Interesting alignment behavior of the gradient flow for training two-layer ReLU networks under small initialization and separable data
+ Rate analysis for the after-alignment-phase convergence

**Weaknesses:**

- The results only hold for correlated data.
- Only ReLU activation functions are analyzed.
- Some of the results have been previously known or observed, e.g., the solution of (stochastic) gradient flow finds in training two-layer ReLU networks on separable data is (almost) low-rank.
- How small the initialization shall be to ensure the two-phase convergence is not qualitiatively discussed?

**Questions:**

1) Do the results/analysis extend to other activation functions?
2) How small the initialization shall be to ensure the two-phase convergence? In general, since the training is nonconvex even with correlated/separable data due to the nonlinear relu activation function in the los. SGD/GD converges to a local minimum and indeed, this has been observed and numerically validated in the literature; see also [R1] Brutzkus et al. 2018. SGD learns over-parameterized networks that provably generalize on linearly separable data. ICLR. [R2] Wang et al. 2019. Learning ReLU networks on linearly separable data: Algorithm, optimality, and generalization. IEEE TSP. In [R1], SDG for two-layer ReLU networks under separable data converges to local minimum; yet for leaky ReLU networks, it finds global minimum. In [R2], it also shows that plain SGD on ReLU networks using separable data converges to local minimum numerically. Yet, a bit modification on the SGD helps SGD converge to a global minimum but with random initialization. It would be great if a comparison can be made between the approach in [R2] and the small-initialization SGD for training two-layer ReLU networks on e.g., MINIST. Moreover, is there any transition for such initialization value to go from the two-phase to single-phase gradient flow convergence to local minimum?
3) It would be great if more numerical tests are provided to demonstrate the two-phase convergence and provide the plots.
4) If the second-layer weights are initialized not in a balanced manner (although not initialized according to (3)), I understand it would also work and guess that the imbalance between positive v_j and negative v_j values only influences the time it takes for the two-phases. More balanced initalization, faster convergence. It would be interesting to numerically validate if this is the case.
5) There are some grammar issues and typos. Please correct.

---

> ### Author Response · Authors · 2023-11-21
> **Thank you for your review**
>
> We thank the reviewer for the constructive comments, please see our response to your questions/concerns:
>
> **Do the results extend to other activation functions**: Yes. Please find a generalized version of Lemma 1 in Appendix B, where we extend the analysis of neuron directional dynamics to any two-layer network with a positive homogenous activation function of degree 1 (including all $\alpha$-leaky ReLU activations, $\alpha\in[0,1]$). However, we note that our main results (Theorem 1) will not directly apply as the neuron directional dynamics have changed as we consider an activation function different than ReLU, and additional technical analysis is required to establish the directional convergence for general leaky-ReLU functions.
>
> **How small $\epsilon$ should be for two-phase convergence**: On page 8, we have a remark "choice of $\epsilon$" answering this question (we highlighted the text in the revision). In short, the initialization scale $\epsilon$ needs to be made small so that (a) the approximation of neuron angular dynamics in equation (4) is good enough (having specific error bound), and (b) the alignment phase lasts long enough so that directional convergence described in Theorem 1 can be achieved. The quantitative analysis in the appendices shows that (a) requires $\epsilon=\mathcal{O}(\frac{\sqrt{\mu}}{\sqrt{h}n})$ and (b) requires $\epsilon=\mathcal{O}(\frac{1}{\sqrt{h}}\exp(-\frac{n}{\sqrt{\mu}}\log n))$. For two-phase convergence, one needs $\epsilon$ to be sufficiently small to satisfy both.
>
> **Comparison with some prior work using SGD**: We thank the reviewer for the references and added them to our comparison with prior work. However, we first would like to point out the key difference in the problem settings: In [R1, R2], the second-layer $v$ is **fixed** and not updated throughout the training, whereas in our manuscript, both the second-layer $v$ and the first-layer $W$ are updated by gradient flow dynamics. We took the reviewer's suggestion and did additional numerical experiments on our two-MNIST-digits classification problem (Please check the highlight section in Appendix A.5) by plotting the quantitative changes in neuron norms and directions separately. We showed that two-phase training (align then fit) happens when weights in both layers are trained, starting from a small initialization (the scenario studied in our manuscript). In contrast, there is no two-phase training when only the first-layer weights are trained (the scenario in [R1, R2]), even from a small initialization.
>
> **Assumption on the data**: We refer the reviewers to the global response we posted for a detailed discussion about our assumption on the data.
>
> **Clarification on balanced initialization**: We would like to point out that our balanced initialization assumption is made for theoretical analysis, and we believe it is not necessary for the dynamics behavior described by our Theorem 1 to happen in practice: All experiments in our manuscript use Gaussian initialization with small variance $\alpha^2$ with some $\alpha<<1$ (Please see highlighted text on Page 9 for the details), and the numerical results can be well explained by our Theorem 1, despite not having a balanced initialization. Using random Gaussian initialization for both layers only introduces additional challenges for the analysis. For example, an important one among these technical challenges is that the $sign(v_j(t))$ is no longer fixed if the initialization is not balanced, thus one needs to keep track of the changes in $sign(v_j(t))$ over the GF trajectory, which is an interesting future research direction.
>
> We hope our response addresses your concerns (we will also proofread our manuscript to minimize grammar issues and typos). If you think there are additional experiments we can do, or clarifications we can make, please feel free to post another comment. We will respond to them before the discussion period ends.
>
> Reference:
>
> [R1] Brutzkus et al. 2018. SGD learns over-parameterized networks that provably generalize on linearly separable data. ICLR
>
> [R2] Wang et al. 2019. Learning ReLU networks on linearly separable data: Algorithm, optimality, and generalization. IEEE TSP

---

> > ### Comment · Reviewer_Qhf5 · 2023-12-01
> > **Nice response**
> >
> > Thanks for addressing the comments and providing additional convincing simulations. I have read the other comments and response, which I found satisfactory. Therefore, I am increasing the score.

---

### Author Response · Authors · 2023-11-21
**Regarding our data assumption**

We first address a concern raised by all reviewers: that our data assumption (Assumption 1) is too strong and unrealistic. Through this response, we hope to convince the reviewers that studying datasets satisfying Assumption 1 is an important and non-trivial first step that constitutes a significant contribution beyond prior works.

More specifically, **The goal** of this paper, and that of other prior works [Stoger'21, Maennel'18, Phuong'21, Boursier'22], is to fully understand the implicit bias of small initialization in training two-layer neural networks. The hope is that once a rigorous and complete analysis for two-layer networks has been developed, it can be generalized to deeper networks.

However, achieving this goal is non-trivial due to the complexity of analyzing the directional dynamics of the network weights during the early phase of training. As we discussed in equation (4)-(5) and formally showed in Lemma 1, the neurons' directional dynamics are conceptually simple but their convergence analysis depends critically on the structure of the data. Specifically, each neuron $w_j$ chases a quantity $x_a(w_j)$ that is a time-varying and non-smooth function of the data points. Therefore, the more complicated the dataset, the more difficult it is to establish convergence.

With this challenge in mind, our strategy is to first analyze simple data structures (e.g., separable datasets under Assumption 1) and understand them fully so that we can then extend the intuitions we obtain and the technical tools we develop to more complex cases. We argue that this strategy is necessary because even for simple data structures, existing analyses [Maennel'18, Phuong'21] have many weaknesses, as we discussed in the remark "Weakness in prior analyses". Therefore, prior to our work, even the simplest case of separable datasets was NOT well studied and understood.

In this regard, our paper makes a significant **contribution** by developing the first complete (from alignment to convergence), non-asymptotic (finite initialization scale $\epsilon$), and quantitative (bounds on lengths of different phases) convergence analysis for orthogonally-separable datasets. We note that despite its simplicity, novel analytical tools are needed for deriving a complete quantitative analysis (see our discussion in Section 3.3). Furthermore, we show that, in our numerical experiments, even for a dataset (two-MNIST-digits dataset) that only approximately satisfies Assumption 1, the experimental results are well explained/predicted by our Theorem 1. Generalizing our analysis to more general datasets is subject to current research that, because of this work, builds upon a solid foundation.

References:

[Stöger'21] Dominik Stöger, et al.. Small random initialization is akin to spectral learning: Optimization and generalization guarantees for overparameterized low-rank matrix reconstruction. NeurIPS, 2021.

[Maennel'18] Hartmut Maennel, et al.. Gradient descent quantizes relu network features. arXiv preprint arXiv:1803.08367, 2018.

[Phuong'21] Mary Phuong, et al.. The inductive bias of relu networks on orthogonally separable data. ICLR, 2021.

[Boursier'22] Etienne Boursier, et al.. Gradient flow dynamics of shallow relu networks for square loss and orthogonal inputs. NeurIPS, 2022.

---

### Meta-Review · Area_Chair_nXr6 · 2023-12-12

**Metareview:**

Summary: The article pursues theoretical analysis of training a binary classifier via gradient flow on two-layer ReLU network under small initialization obtaining results depending on several factors.

Strengths: Referees found the article well written and the results interesting. Particularly, the article gives a discussion of alignment and convergence with deterministic initialization and non asymptotic quantitative results.

Weaknesses: A concern were the assumptions needed for the results, particularly an assumption of positive correlated labels.The limitations from this assumption were weighed differently by different reviewers. Another concern was about the settings of the network which involve certain dependencies between weights. Lastly, there were some concerns with the significance, since the result is an extension of a previous results with infinitesimal initialization. The referees found the difference to prior works was clearly presented. Adding a reference / comment on the related work of Frei et al. on Implicit Bias in Leaky ReLU Networks Trained on High-Dimensional Data would seem adequate.

At the end of the discussion period the article had mixed ratings with two referees regarding it as being marginally below the acceptance threshold and two others regarding it as good. In spite of the limitations, I find that the presented results are a non-trivial and worthwhile advance in a relevant and challenging subject. Thus I am recommending accept. I ask that the authors pay special attention to taking into account the reviewers feedback and making sure the limitations are clearly spelled out in the final version of the article. I may add a detail that the work uses results of Boursier et al. to justify the use of the naive subgradient flow, but this seems to be justified only if it is assumed that the data is orthogonal. This probably doesn't affect the results too much, but it needs to be fixed adequately.

**Justification For Why Not Higher Score:**

The work makes a valuable contribution yet this is not without limitations. Thus while I am recommending accept, I am not recommending an award.

**Justification For Why Not Lower Score:**

The work makes a worthwhile contribution to a relevant and challenging topic. The provided analysis can serve to make further progress on a broader program.

---

### Decision · Program_Chairs · 2024-01-16

Accept (poster)